# FROM GEOMETRY TO DYNAMICS: LEARNING OVERDAMPED LANGEVIN DYNAMICS FROM SPARSE OBSERVATIONS WITH GEOMETRIC CONSTRAINTS

## ABSTRACT

How can we learn the laws underlying the dynamics of stochastic systems when their trajectories are sampled sparsely in time? Existing methods either require temporally resolved high-frequency observations, or rely on geometric arguments that apply only to conservative systems, limiting the range of dynamics they can recover. Here, we present a new framework that reconciles these two perspectives by reformulating inference as a stochastic control problem. Our method uses geometry-driven path augmentation, guided by the geometry in the system's invariant density to reconstruct likely trajectories and infer the underlying dynamics without assuming specific parametric models. Applied to overdamped Langevin systems, our approach accurately recovers stochastic dynamics even from extremely undersampled data, outperforming existing methods in synthetic benchmarks. This work demonstrates the effectiveness of incorporating geometric inductive biases into stochastic system identification methods.

## 1 INTRODUCTION

How can we discover the underlying driving forces that govern the behaviour of complex, stochastic systems when we only measure their state at discrete time points? From pollen motion in a liquid medium (Einstein, 1905) and chemical reactions (Li, 2020) to population dynamics (Silva-Dias and López-Castillo, 2018; Fisher and Mehta, 2014) and cell growth (Alonso et al., 2014), many natural processes evolve following stochastic dynamics, best described by Langevin or stochastic differential equations (SDEs) of the form

$$\mathrm{d}\mathbf{X}_t = \mathbf{f}(\mathbf{X}_t)\,\mathrm{d}t + \boldsymbol{\sigma}\,\mathrm{d}\mathbf{W}_t. \tag{1}$$

Under this formalism, the deterministic part of the equation $\mathbf{f}(\cdot) : \mathcal{R}^d \to \mathcal{R}^d$, the *drift* function, captures the long-term evolution of the state variables, while the stochastic part $\boldsymbol{\sigma} : \mathcal{R}^d \times \mathcal{R}^d$, the *diffusion*, accounts for the contribution of unresolved degrees of freedom. In practice, however, we rarely observe these systems at the fine time scales required by existing inference methods.

Recent advances in dynamical system inference have delivered valuable tools for identifying continuous-time *deterministic* systems from observations (Cremers and Hübler, 1987; Brunton et al., 2016; Daniels and Nemenman, 2015; McGoff et al., 2015; Kantz and Schreiber, 2004; Schmidt and Lipson, 2009). **Data-driven** (or **nonparametric**, or **equation-free**) approaches seek to reconstruct the governing equations of observed systems directly from state observations, without imposing explicit assumptions or inductive biases about the underlying dynamical models. They rely on function approximation to infer the system's structure from observations, such as basis functions (Acosta, 1995; Small and Tse, 2002; Judd and Mees, 1995; Small and Judd, 1998; Brückner et al., 2020; Frishman and Ronceray, 2020), symbolic regression (Kaiser et al., 2018; Brunton et al., 2016; Bongard and Lipson, 2007; Daniels and Nemenman, 2015), spectral approximations (Kevrekidis et al., 2003; Theodoropoulos et al., 2000), Gaussian processes (Alvarez et al., 2009; Sanguinetti et al., 2006; Särkkä, 2019), or neural networks (Teng, 2018; Bhattoo et al., 2022; Jüngling et al., 2019). However, extending these methods to *stochastic* systems remains difficult. In this setting, inference must disentangle the influence of underlying deterministic forces from random fluctuations, a task that is particularly difficult when sampling rates are low.

**Two dominant perspectives for stochastic inference.** Data-driven system identification for stochastic systems largely follows two tracks. *Temporal methods* (Fig. 1**A.**) rely on the **tempo-**

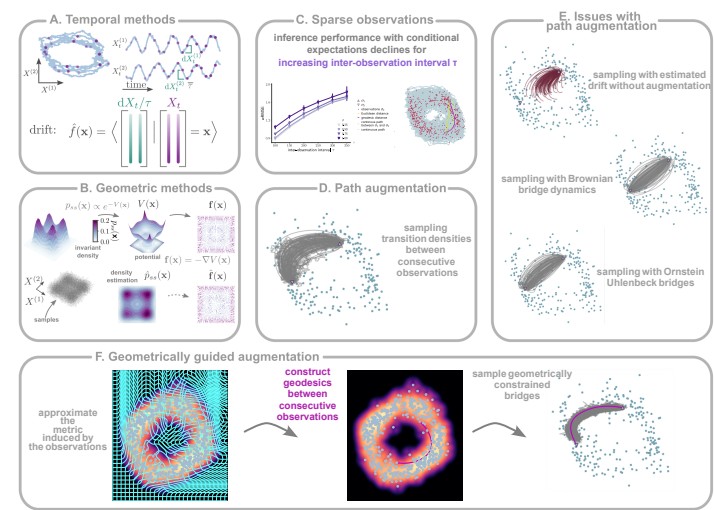

**Figure 1**

**Temporal and geometric perspectives for discovering stochastic dynamics and proposed inference with geometrically guided augmentation.** (**A.**) Temporal methods consider the time-ordering of observations $\{\mathcal{O}_k\}_{k=1}^K$ (*purple dots*) to approximate the drift $\mathbf{f}(\mathbf{x})$ with conditional rescaled state increments $\hat{\mathbf{f}}(\mathbf{x}) = \langle \frac{d\mathbf{X}_t}{\tau} | \mathbf{X}_t = \mathbf{x} \rangle$. (**B.**) Geometric methods assume a conservative drift $\mathbf{f}(\mathbf{x}) = -\nabla V(\mathbf{x})$ as the gradient of a potential. (**C.**) With increasing inter-observation interval $\tau$ performance of temporal methods degrades because Euclidean distances ignore the curvature of the latent continuous path between consecutive observations. (**D.**) Path augmentation alternates between state estimation - by sampling diffusion bridges for each inter-observation interval - and drift inference. (**E.**) Commonly used path augmentation methods employ Brownian or Ornstein-Uhlenbeck bridges that increasingly deviate from the unobserved path as $\tau$ grows. (lower) Illustration of the ground truth (*neon green*) and geodesic (*magenta*) continuous path between two observations and of that assumed during inference with Gaussian likelihood (*yellow line*). (**F.**) Geometrically guided augmentation approximates first the metric induced by the invariant density, constructs geodesics connecting consecutive observations, and samples geometrically constrained diffusion bridges.

**ral ordering** of measurements, regressing state increments against states to estimate the drift, which works when the inter-observation interval ($\tau$) is small (Batz et al., 2018; Friedrich and Peinke, 1997; Ragwitz and Kantz, 2001). *Geometric methods* on the other hand, approximate the **invariant density** (Batz et al., 2016; Gu et al., 2021) or eigenstructure of the infinitesimal generator of the diffusion process (Singer and Coifman, 2008; Nüske et al., 2021; Ionides et al., 2006; Talmon and Coifman, 2015; Dsilva et al., 2016; Berry and Harlim, 2018)) (Fig. 1**B.**), but are nevertheless limited to systems with conservative forces (Berry and Harlim, 2015; Batz et al., 2016) or decoupled state variables (Singer and Coifman, 2008). Each perspective has limitations: temporal approaches deteriorate with increasing inter-observation intervals (Fig. 1**C.**), whereas geometric methods are restricted to conservative flows.

> **A unifying perspective: reconcile temporal and geometric methods by constraining with most probable paths extracted from the invariant density.** Here, we recast inference into a stochastic control problem and introduce **geometry-aware path augmentation**. Our method follows a simple premise that incorporates **geometric inductive biases** informed by the system's *invariant density* into dynamical inference: we postulate that the augmented paths should lie **in the vicinity** of **geodesic curves** (Fig. 1**F.** middle, magenta line) that connect consecutive measurements on the **empirical manifold** induced by the observations. To achieve this, **(i)** we approximate the Riemannian metric induced by the observations (Fig. 1**F.**) without the need to predefine the dimensionality of the empirical manifold, **(ii)** compute geodesics between consecutive observations through nonparametric approximation of shortest path distances between consecutive observations according to the approximated metric, and **(iii)** estimate the unobserved path between consecutive observations by generating **geometrically constrained diffusion bridges** that both respect temporal order and are guided toward identified geodesics (Fig. 1 **F.**). Nonparametric estimation of the drift function based on the augmented paths within an Expectation Maximisation framework (**E.M.**) (Dempster et al., 1977) results in accurate approximations of the underlying

stochastic dynamics. Extensive numerical experiments demonstrate the effectiveness of our proposed method in recovering the true stochastic dynamics, even in challenging scenarios where existing approaches fail.

## 2 SETUP AND BACKGROUND

**Setting.** We consider a system whose state evolves according to Eq. 1. Here, $\mathbf{X}_t \in \mathcal{R}^d$ denotes the state of the system, $\mathbf{f}(\cdot) : \mathcal{R}^d \to \mathcal{R}^d$ is the drift function, $\boldsymbol{\sigma}$ stands for the diffusion constant or matrix, and $\mathbf{W}_t \in \mathcal{R}^d$ is a $d-$dimensional Wiener process representing random noise input or unresolved degrees of freedom.

**Data.** We observe the system state at discrete time points $t_k = k\tau$ at **inter-observation intervals** of $\tau$ time units, obtaining a time-ordered set of observations $\{\mathcal{O}_k \doteq \mathbf{X}_{t_k}\}_{k=1}^K$.

**Goal.** Our goal is to estimate the drift function $\mathbf{f}(\cdot)$ representing the deterministic forces acting on the system of interest from the discrete state observations $\{\mathcal{O}_k\}_{k=1}^K$.

**Background.** Common inference methods for this setting consider observations from the system path $\mathbf{X}_{0:T}$ in (nearly) continuous time (Batz et al., 2018; Friedrich and Peinke, 1997). Under such conditions, the infinitesimal transition probability of the SDE between observations $\mathbf{X}_t$ and $\mathbf{X}_{t+dt}$ is Gaussian

$$\mathbf{P}_f(\mathbf{X}_{0:T} \mid \mathbf{f}) \propto \exp\left(-\frac{1}{2\,\mathrm{d}t}\sum_t \|\mathbf{X}_{t+\mathrm{d}t} - \mathbf{X}_t - \mathbf{f}(\mathbf{X}_t)\mathrm{d}t\|_D^2\right), \tag{2}$$

where $\|\mathbf{u}\|_D \doteq \mathbf{u}^\top \cdot \mathbf{D}^{-1} \cdot \mathbf{u}$, denotes the weighted norm with $\mathbf{D} \doteq \boldsymbol{\sigma}\boldsymbol{\sigma}^\top$ indicating the noise covariance. The likelihood for the drift $\mathbf{f}$ given the path $\mathbf{X}_{0:T}$ observed during $[0, T]$, results from the Radon-Nykodym derivative (likelihood ratio) between $\mathbf{P}_f(\mathbf{X}_{0:T}|f)$ and the transition probability of a Wiener path $\mathbf{P}_\mathcal{W}(\mathbf{X}_{0:T}) = \exp\left(-\frac{1}{2\mathrm{d}t}\sum_t \|\mathbf{X}_{t+\mathrm{d}t} - \mathbf{X}_t\|_D^2\right)$ as (Liptser and Shiryaev, 2013)

$$\mathcal{L}(\mathbf{X}_{0:T} \mid \mathbf{f}) = \exp\left(-\frac{1}{2}\sum_t \|\mathbf{f}(\mathbf{X}_t)\|_D^2 \mathrm{d}t + \sum_t \langle \mathbf{f}(\mathbf{X}_t), \mathbf{X}_{t+\mathrm{d}t} - \mathbf{X}_t \rangle_D\right). \tag{3}$$

This likelihood has a quadratic form in terms of the drift function. This makes **Gaussian process** priors a natural and widely employed approach for modelling $\mathbf{f}$ (Ruttor et al., 2013; Hostettler et al., 2018; Zhao et al., 2020).

However, these approaches rely on *small* inter-observation intervals $\tau$ (Batz et al., 2018). As $\tau$ increases, the EuM approximation becomes inaccurate: transition densities are not Gaussian, and higher-order remainder terms related to the curvature of the flow field become important (see further theoretical analysis in Sec. G.3 and c.f. Fig. 6). Attempts to mitigate this problem by introducing bridge sampling to infer the unobserved path between observations (Batz et al., 2018; Sermaidis et al., 2013) provide small improvements, because these methods rely on linearised or otherwise simplified bridge dynamics that do not match the true transition densities (c.f. Sec. E).

Here, we target this large inter-observation interval setting by merging insights from both temporal and geometric perspectives. Specifically, our approach combines **nonlinear** bridge sampling with **a geometric approximation of the system's invariant density** as detailed in the following.

## 3 METHODOLOGY

**Core idea.** The invariant density of the observed system imposes a low-dimensional structure on the state space, within which the observations are confined. We propose that this low-dimensional structure is well approximated by a Riemannian manifold $\mathcal{M}_\infty \in \mathcal{R}^{m \le d}$ in the ambient space, and that the ensemble of observations $\{\mathcal{O}_k\}_{k=1}^K$ offers a reliable discrete approximation to $\mathcal{M}_\infty$ (Sec. **??**). We term this observation-based approximation the *empirical manifold* $\mathcal{M}$. The central premise of our approach is that **unobserved paths between successive observations will be lying either *on* or *in the vicinity* of the empirical manifold** $\mathcal{M}$. In particular, we postulate that unobserved paths should lie **in the vicinity of geodesics that connect consecutive observations** on $\mathcal{M}$.

However, while this view of a lower dimensional manifold embedded in a higher dimensional ambient space helps to build intuition, for practical purposes we adopt a complementary view of the low dimensional manifold inspired by (Fröhlich et al., 2021). According to this view, we consider

the entire observation space $\mathcal{R}^d$ as a smooth Riemannian manifold, $\mathcal{M} \doteq \mathcal{R}^d$, characterised by a Riemannian metric $\mathfrak{h}$. The effect of the nonlinear geometry of the observations is then captured by the metric $\mathfrak{h}$. Thus to approximate the geometric structure of the system's invariant density, we learn the Riemannian metric tensor $H : \mathcal{R}^d \to \mathcal{R}^{d \times d}$ and compute the geodesics between consecutive observations according to the learned metric. Intuitively according to this view the observations $\{\mathcal{O}_k\}_{k=1}^K$ introduce distortions in the way we compute distances on the state space. The advantage of this approach is that we do not have to estimate the dimensionality of the empirical manifold, which would have been difficult due to the presence of fluctuations in the system's dynamics. Instead, we still operate in the original space and the empirical manifold introduces distortions in the estimated metric (see Fig. 1**F.i.**).

**Inference framework.** Our approach comprises three steps: **($\alpha$.)** Approximation of the geometric structure of the system's invariant density with metric learning, **($\beta$.)** estimation of the (latent) system state between consecutive observations guided by the invariant density (**path augmentation**), and **($\gamma$.)** data-driven estimation of the drift function (Fig. 1). We perform the two final steps in an iterative manner within an Expectation Maximisation (**E.M.**) framework (Dempster et al., 1977).

**($\alpha$.) Approximating the Riemannian geometry induced by the observations.** Although there are many methods for approximating Riemannian manifolds (Tenenbaum et al., 2000; Balasubramanian and Schwartz, 2002; Mead, 1992; Roweis and Saul, 2000), our objective is to obtain a representation that acts as a *local* constraint for subsequent state estimation between consecutive observations. We achieve this in two steps: **(i.)** We approximate in the ambient space $\mathcal{R}^d$ the metric $\mathfrak{h}$ induced by the observations (see Fig. 1**F.i.**). This identifies regions of the state space with high observation density (represented with small metric values). **(ii.)** We construct geodesics between consecutive observations on the empirical manifold $(\mathcal{M} \doteq \mathcal{R}^d, \mathfrak{h})$ (see Fig. 1**F.ii.**). The geodesics identify the most probable paths between consecutive observations, and each such path subsequently functions as a constraint during latent state estimation.

**(i.) Approximation of the invariant metric.** To approximate the (local) metric $\mathfrak{h}$ in a nonparametric form at locations $\mathbf{x}$ of the state space, we follow Arvanitidis et al. (2019), and consider the inverse of the weighted local diagonal covariance computed on the $K$ observations as

$$H_{dd}(\mathbf{x}) = \left( \sum_{k=1}^K w_k(\mathbf{x}) \left( \mathcal{O}_k^{(d)} - x^{(d)} \right)^2 + \epsilon \right)^{-1}, \tag{4}$$

with weights $w_k(\mathbf{x}) = \exp\left( -\frac{\|\mathcal{O}_k - \mathbf{x}\|_2^2}{2\sigma_{\mathcal{M}}^2} \right)$, and $A^{(d)}$ denoting the $d$-th dimensional component of the vector $\mathbf{A}$ for $\mathbf{A} \in \{\mathbf{x}, \mathcal{O}_k\}$. The parameter $\epsilon > 0$ is a small value ensuring non-zero diagonals of the weighted covariance matrix, while $\sigma_{\mathcal{M}}$ is a hyper-parameter characterising the curvature of the approximated manifold.

**(ii.)Constructing geodesics between consecutive observations.** To compute the geodesic curves connecting consecutive observations on the empirical manifold, we employ the approximated metric tensor $\mathbf{H}(\mathbf{x})$. We identify the geodesic curve $\gamma_{t'}^k$ between $\mathcal{O}_k$ and $\mathcal{O}_{k+1}$ as the curve with minimum energy that connects these two points, i.e., as the minimiser of the kinetic energy functional $\mathcal{E}(\gamma_{t'}^k) = \int_0^1 L_{\mathcal{M}}(\gamma_{t'}^k, \dot{\gamma}_{t'}^k)\, dt'$

$$\gamma_{t'}^{k*} = \underset{\substack{\gamma_{t'}^k, \\ \gamma_0^k = \mathcal{O}_k, \gamma_1^k = \mathcal{O}_{k+1}}}{\arg\min} \int_0^1 L_{\mathcal{M}}(\gamma_{t'}^k, \dot{\gamma}_{t'}^k) dt', \quad \text{with} \quad \int_0^1 L_{\mathcal{M}}(\gamma_{t'}^k, \dot{\gamma}_{t'}^k)\, dt' = \frac{1}{2} \int_0^1 \|\dot{\gamma}_{t'}^k\|_{\mathfrak{h}}^2, \tag{5}$$

where $L_{\mathcal{M}}(\gamma_{t'}^k, \dot{\gamma}_{t'}^k)$ is an appropriately constructed Lagrangian. The minimising curve of this functional is the same as the minimiser of the curve length functional $\ell(\gamma_{t'})$ (c.f. Eq. 33), i.e., the geodesic (Do Carmo and Flaherty Francis, 1992). This results in a system of second order differential equations (Eq. 36) (Arvanitidis et al., 2017; Do Carmo and Flaherty Francis, 1992) (Sec. A.3.2) with boundary conditions $\gamma_0^k = \mathcal{O}_k$ and $\gamma_1^k = \mathcal{O}_{k+1}$ that we solve with a probabilistic differential equation solver as in (Arvanitidis et al., 2019).

**($\beta$.) Latent state estimation: Geometry-guided augmentation.** To estimate the unobserved system state between consecutive observations $\mathcal{O}_k$ and $\mathcal{O}_{k+1}$, we perform variational inference (Beal, 2003)(see Sec. A.3). Given a prior diffusion process with drift $\hat{\mathbf{f}}(\cdot) : \mathcal{R}^d \to \mathcal{R}^d$ and diffusion $\sigma$,

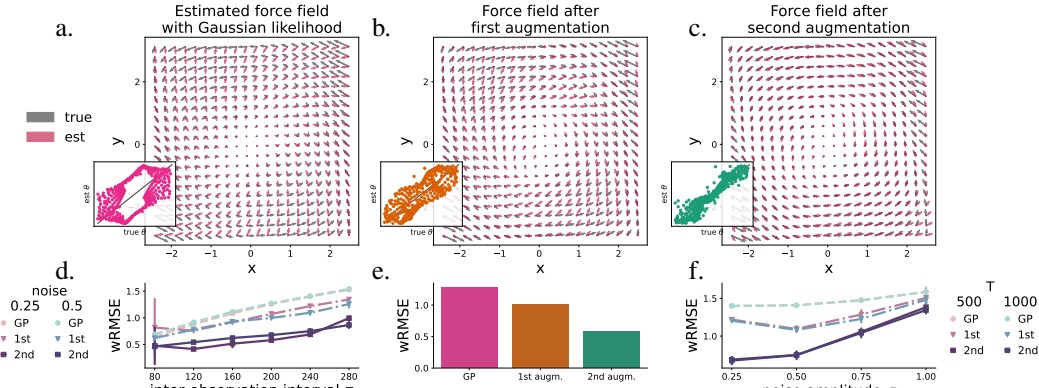

**Figure 2**

**Geometry-aware path augmentation improves drift inference after two iterations.** Estimated (*red*) vs. true (*grey*) force field with **a.)** Gaussian likelihood, **b.)** after one, and **c.)** after two augmentations. (**Insets**) True vs. estimated angles at grid points. **d.)** Weighted (by observation density) root mean square error (wRMSE) vs. inter-observation interval $\tau$ for different noise levels $\sigma = \{0.25, 0.5\}$ for drift estimated with a Gaussian likelihood (*gaus*-circles), after first augmentation (*1st*-triangles), and after second augmentation (*2nd*-squares) for $T = 500$ (time units). **e.)** wRMSE across iterations for the presented example. **f.)** wRMSE vs. noise amplitude $\sigma$ for different trajectory durations $T = \{500, 1000\}$ (time units) for inter-observation interval $\tau = 240$ ($dt$). Markers in **d.)** and **f.)** indicate augmentation steps. Error bars: one standard deviation over five independent runs.

we construct an **approximating process** conditioned **i.)** to pass through the observations, and **ii.)** to respect the local geometry of the invariant density as it is represented by the geodesics. The conditioned process is also a diffusion process with the same diffusion constant and an effective drift function $\mathbf{g}(\mathbf{x}, t)$ (Chetrite and Touchette, 2015; Majumdar and Orland, 2015). The path probability measure $Q_X(\mathbf{X}_{0:T})$ induced by the approximating process

$$Q_X(\mathbf{X}_{0:T}): \quad \mathrm{d}\mathbf{X}_t = \mathbf{g}\left(\mathbf{X}_t, t\right)\mathrm{d}t + \sigma\mathrm{d}\bar{\mathbf{W}}_t = \left(\hat{\mathbf{f}}(\mathbf{X}_t) + \mathbf{u}(\mathbf{X}_t, t)\right)\mathrm{d}t + \sigma\mathrm{d}\bar{\mathbf{W}}_t, \quad (6)$$

provides an approximation to the unobserved continuous system state. In Eq. 6 $\mathbf{u}(\cdot, \cdot) : \mathcal{R}^d \times \mathcal{R}^+ \to \mathcal{R}^d$ is a time-dependent control term that guides the approximating path distribution, through the observations, while staying in the vicinity of the corresponding geodesics between them.

More precisely, we obtain the controlled drift $\mathbf{g}\left(\mathbf{X}_t, t\right)$ from the solution of the variational problem of minimising the functional (see Sec. A.3.1)

$$\mathcal{F}[Q_X] = \mathcal{KL}\left(Q_X(\mathbf{X}_{0:T})||\mathrm{P}(\mathbf{X}_{0:T} \mid \hat{\mathbf{f}})\right) - \sum_{k=1}^{K}\left\langle \ln \mathrm{P}(\mathcal{O}_k \mid \mathbf{X}_{t_k})\right\rangle_Q + \left\langle \|\mathbf{\Gamma}_t - \mathbf{X}_{0:T}\|^2\right\rangle_Q$$

$$= \frac{1}{2}\int\limits_0^T \int \left[\|\mathbf{g}(\mathbf{x}, t) - \hat{\mathbf{f}}(\mathbf{x})\|_{\mathbf{D}}^2 + U_{\mathcal{O}}(\mathbf{x}, t) + \beta\, U_{\mathcal{G}}(\mathbf{x}, t)\right] q_t(\mathbf{x})\,\mathrm{d}\mathbf{x}\,\mathrm{d}t, \quad (7)$$

where $\mathbf{\Gamma}_t$ denotes the sequence of $K$ geodesics indexed by time $t$, $\mathbf{\Gamma}_t \dot{=} \{\gamma_{t'}^k\}_{t=(k-1)\tau + t'\tau}$, where $\gamma_{t'}^k$ is the geodesic connecting $\mathcal{O}_k$ and $\mathcal{O}_{k+1}$, and $t' \in [0, 1]$ denotes a rescaled time variable, and $\beta$ is a weighting term. In Eq. 7, the term $U_{\mathcal{O}}(\mathbf{x}, t) = -\sum_{t_k} \ln \mathrm{P}(\mathcal{O}_k \mid \mathbf{x})\, \delta(t - t_k)$ **forces the augmentation to pass through the observations at each bridge boundary**, while $U_{\mathcal{G}}(\mathbf{x}, t) \dot{=} \|\mathbf{\Gamma}_t - \mathbf{x}\|^2$ **guides the latent path towards the identified geodesics**.

This minimisation can be construed as a stochastic control problem (Opper, 2019) with the objective to identify a time-dependent drift adjustment $\mathbf{u}(\mathbf{x}, t) := \mathbf{g}(\mathbf{x}, t) - \hat{\mathbf{f}}(\mathbf{x})$ for the system with drift $\hat{\mathbf{f}}(\mathbf{x})$ so that the controlled dynamics fulfil the path constraints $U_{\mathcal{O}}(\mathbf{x}, t)$ and $U_{\mathcal{G}}(\mathbf{x}, t)$.

The optimal time-dependent control for the interval between $\mathcal{O}_k$ and $\mathcal{O}_{k+1}$ results from the solution of the backward equation (Kappen, 2005a; Maoutsa and Opper, 2022)

$$\frac{\partial \phi_t(\mathbf{x})}{\partial t} = -\mathcal{L}_{\hat{f}}^{\dagger} \phi_t(\mathbf{x}) + U_{\mathcal{G}}(\mathbf{x}, t)\phi_t(\mathbf{x}), \tag{8}$$

with terminal condition $\phi_{t_{k+1}}(\mathbf{x}) = \chi(\mathbf{x}) = \delta(\mathbf{x} - \mathcal{O}_{k+1})$ and with $\mathcal{L}_{\hat{f}}^{\dagger}$ denoting the adjoint Fokker-Planck operator for the process of Eq. 26. As shown in (Maoutsa and Opper, 2022) the optimal drift adjustment $\mathbf{u}(\mathbf{x}, t)$ can be expressed in terms of the difference of the logarithmic gradients of two probability flows

$$\mathbf{u}^*(\mathbf{x}, t) = D\Big(\nabla \ln q_{T-t}(\mathbf{x}) - \nabla \ln \rho_t(\mathbf{x})\Big), \tag{9}$$

where $\rho_t$ fulfils the forward (filtering) partial differential equation (PDE)

$$\frac{\partial \rho_t(\mathbf{x})}{\partial t} = \mathcal{L}_{\hat{f}}\rho_t(\mathbf{x}) - U(\mathbf{x}, t)\rho_t(\mathbf{x}), \tag{10}$$

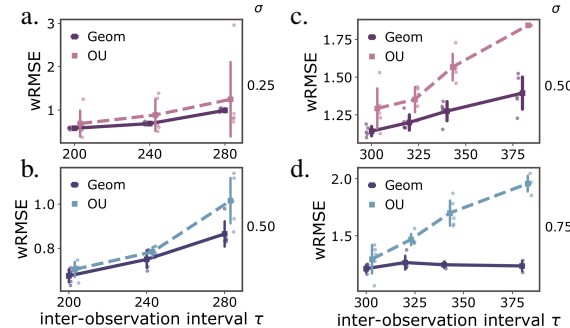

**Figure 3**

**Comparison of geometry-aware inference against inference with Ornstein-Uhlenbeck augmentation.** Weighted root mean square error (wRMSE) vs. different inter-observation intervals $\tau$ for different noise amplitudes for moderate inter-observation intervals with **a.)** $\sigma = 0.25$ and **b.)** $\sigma = 0.50$, and for large inter-observation intervals with **c.)** $\sigma = 0.50$ and **d.)** $\sigma = 0.75$, where only one observation per oscillation period is available. Error bars indicate one standard deviation over five independent runs.

while $q_t$ is the solution of a time-reversed PDE with initial condition $q_0(\mathbf{x}) \propto \rho_T(\mathbf{x})\chi(\mathbf{x})$

$$\frac{\partial q_t(\mathbf{x})}{\partial t} = -\nabla \cdot \left[\Big(\sigma^2 \nabla \ln \rho_{T-t}(\mathbf{x}) - \mathbf{f}(\mathbf{x}, T-t)\Big)q_t(\mathbf{x})\right] + \frac{\sigma^2}{2}\nabla^2 q_t(\mathbf{x}). \tag{11}$$

Thus, for each interval $[\mathcal{O}_k, \mathcal{O}_{k+1}]$ we identify the posterior path measure (minimiser of Eq. 37) by solving such a stochastic control problem for the time-varying control $\mathbf{u}(\mathbf{x}, t)$ of Eq. 9. This results in a set of $K - 1$ *independent* optimal control problems, that are solved in parallel for efficiency.

**($\gamma$.) Estimating the drift.** We approximate the drift function in a model independent framework by imposing a Gaussian process prior on the function values $\mathbf{f} \sim \mathrm{P}_o(\mathbf{f}) = \mathcal{GP}(\mathbf{m}^f, k^f)$, where $\mathbf{m}^f$ and $k^f$ denote the mean and covariance function of the Gaussian process. The optimal measure for the drift $Q_f$ is a Gaussian process given by (Batz et al., 2018)

$$Q_f \propto \mathrm{P}_o \exp\left(-\frac{1}{2}\int \|\mathbf{f}(\mathbf{x})\|_{\sigma^2}^2 A(\mathbf{x}) - 2\langle \mathbf{f}(\mathbf{x}), B(\mathbf{x})\rangle_{\sigma^2}\mathrm{d}\mathbf{x}\right), \tag{12}$$

with $A(\mathbf{x}) \doteq \int_0^T q_t(\mathbf{x})\mathrm{d}t$ and $B(\mathbf{x}) \doteq \int_0^T q_t(\mathbf{x})\mathbf{g}(\mathbf{x}, t)\mathrm{d}t$, where $q_t(\mathbf{x})$ denotes the marginal density of the constrained process' state obtained by the state estimation. The function $\mathbf{g}(\mathbf{x}, t)$ denotes the effective (time-dependent) drift of the constrained process (Eq. 6), resulting from the solution of the individual control problems accounting for the observations and the invariant geometry.

## 4 RESULTS

**Revealing stochastic dynamics in model systems.** To demonstrate the effectiveness of our approach, we inferred the stochastic dynamics of model systems, and compared the resulting estimates to those obtained from: **(i.)** Gaussian process regression without state estimation (**GP**), **(ii.)** path augmentation with Ornstein-Uhlenbeck dynamics (**OU**) (Batz et al., 2018), **(iii.)** sparse variational inference with state estimation (**SVISE**) (Course and Nair, 2023a), **(iv.)** basis function approximation of Kramers-Moyal coefficients, i.e. the drift function (**KM-basis**) (Nabeel et al., 2025), and **(v.)** latent SDE inference with amortized reparameterization with (**LatentSDE+GP-pre**) and without pre-training (**LatentSDE**) (Course and Nair, 2023b), **(vi.)** metric flow matching (**MFM**) (Kapusniak et al., 2024)(with RBF (Arvanitidis et al., 2021) and LAND metric (Arvanitidis et al., 2019) metric approximations), **(vii.)** generalized Schrödinger bridge matching (**GSBM**) (Liu et al., 2023),

| wRMSE ↓ | total duration T | Van der Pol | | | | | |
|---|---|---|---|---|---|---|---|
| | | $\tau = 80 \times \mathrm{d}t$ | $\tau = 120 \times \mathrm{d}t$ | $\tau = 160 \times \mathrm{d}t$ | $\tau = 200 \times \mathrm{d}t$ | $\tau = 240 \times \mathrm{d}t$ | $\tau = 280 \times \mathrm{d}t$ |
| **$\sigma = 0.25$** | | | | | | | |
| GP | 500 | $0.642 \pm 0.006$ | $0.879 \pm 0.005$ | $1.083 \pm 0.015$ | $1.258 \pm 0.011$ | $1.399 \pm 0.003$ | $1.528 \pm 0.0153$ |
| SVISE | 500 | $1.465 \pm 0.009$ | $0.857 \pm 0.021$ | $0.740 \pm 0.072$ | $0.592 \pm 0.026$ | $\mathbf{0.587 \pm 0.112}$ | $\mathbf{0.824 \pm 0.003}$ |
| KM-basis | 500 | $\mathbf{0.368 \pm 0.054}$ | $0.452 \pm 0.011$ | $0.671 \pm 0.023$ | $1.588 \pm 0.021$ | $1.751 \pm 0.008$ | $1.735 \pm 0.020$ |
| LatentSDE | 500 | $1.091 \pm 0.316$ | $1.091 \pm 0.039$ | $1.098 \pm 0.023$ | $1.089 \pm 0.036$ | $1.088 \pm 0.038$ | $1.091 \pm 0.039$ |
| LatentSDE+GP-pre | 500 | $1.095 \pm 0.038$ | $1.085 \pm 0.039$ | $1.101 \pm 0.034$ | $1.089 \pm 0.038$ | $1.106 \pm 0.045$ | $1.102 \pm 0.039$ |
| GSBM | 500 | $1.475$ | $1.479$ | - | | - | - |
| [SF]2M | 1500 | $1.741 \pm 0.304$ | $1.801 \pm 0.226$ | $1.745 \pm 0.322$ | $1.583 \pm 0.132$ | $1.816 \pm 0.228$ | $1.721 \pm 0.094$ |
| MFM$_{\mathrm{RBF}}$ | 1500 | $1.462 \pm 0.007$ | $1.469 \pm 0.005$ | $1.470 \pm 0.012$ | $1.469 \pm 0.008$ | $1.469 \pm 0.006$ | $1.466 \pm 0.008$ |
| MFM$_{\mathrm{LAND}}$ | 1500 | $1.463 \pm 0.007$ | $1.469 \pm 0.005$ | $1.469 \pm 0.012$ | $1.469 \pm 0.008$ | $1.469 \pm 0.006$ | $1.467 \pm 0.008$ |
| **Geometric$_{\mathrm{RBF}}$ (our)** | 500 | $0.419 \pm 0.052$ | $0.458 \pm 0.063$ | $0.493 \pm 0.031$ | $0.517 \pm 0.022$ | $0.657 \pm 0.040$ | $1.001 \pm 0.077$ |
| **Geometric (our)** | 500 | $0.474 \pm 0.034$ | $\mathbf{0.413 \pm 0.016}$ | $\mathbf{0.514 \pm 0.068}$ | $\mathbf{0.578 \pm 0.022}$ | $0.687 \pm 0.032$ | $0.993 \pm 0.037$ |
| **$\sigma = 0.50$** | | | | | | | |
| GP | 500 | $0.691 \pm 0.029$ | $0.916 \pm 0.014$ | $1.114 \pm 0.15$ | $1.272 \pm 0.030$ | $1.409 \pm 0.019$ | $1.542 \pm 0.044$ |
| SVISE | 500 | $1.235 \pm 0.083$ | $0.9935 \pm 0.015$ | $0.7505 \pm 0.052$ | $0.736 \pm 0.072$ | $1.3565 \pm 0.278$ | $1.425 \pm 0.086$ |
| KM-basis | 500 | $0.495 \pm 0.010$ | $0.727 \pm 0.008$ | $0.890 \pm 0.024$ | $1.683 \pm 0.020$ | $1.744 \pm 0.038$ | $1.732 \pm 0.065$ |
| LatentSDE | 500 | $1.158 \pm 0.036$ | $1.151 \pm 0.045$ | $1.160 \pm 0.032$ | $1.151 \pm 0.036$ | $1.146 \pm 0.033$ | $1.176 \pm 0.046$ |
| LatentSDE+GP-pre | 500 | $1.158 \pm 0.045$ | $1.159 \pm 0.034$ | $1.159 \pm 0.027$ | $1.151 \pm 0.034$ | $1.150 \pm 0.028$ | $1.191 \pm 0.052$ |
| GSBM | 500 | $4.129$ | $2.448$ | $2.448$ | $2.789$ | $11.416$ | $11.416$ |
| [SF]2M | 1500 | $1.869 \pm 0.482$ | $1.813 \pm 0.286$ | $1.484 \pm 0.096$ | $1.876 \pm 0.247$ | $1.753 \pm 0.158$ | $1.707 \pm 0.233$ |
| MFM$_{\mathrm{RBF}}$ | 1500 | $1.516 \pm 0.011$ | $1.525 \pm 0.006$ | $1.538 \pm 0.009$ | $1.537 \pm 0.017$ | $1.528 \pm 0.015$ | $1.544 \pm 0.019$ |
| MFM$_{\mathrm{LAND}}$ | 1500 | $1.517 \pm 0.011$ | $1.526 \pm 0.006$ | $1.536 \pm 0.009$ | $1.537 \pm 0.017$ | $1.528 \pm 0.015$ | $1.545 \pm 0.019$ |
| **Geometric$_{\mathrm{RBF}}$ (our)** | 500 | $0.653 \pm 0.014$ | $0.690 \pm 0.026$ | $0.694 \pm 0.026$ | $0.761 \pm 0.050$ | $0.798 \pm 0.047$ | $0.933 \pm 0.160$ |
| **Geometric (our)** | 500 | $\mathbf{0.462 \pm 0.019}$ | $\mathbf{0.541 \pm 0.023}$ | $\mathbf{0.621 \pm 0.012}$ | $\mathbf{0.675 \pm 0.030}$ | $\mathbf{0.750 \pm 0.038}$ | $\mathbf{0.865 \pm 0.057}$ |

**Table 1**

Performance comparison in terms of weighted root mean square error (wRMSE) of considered frameworks for different noise conditions $\sigma$ and inter-observation intervals $\tau$ for the Van der Pol system.

**(viii.)** simulation-free Schrödinger bridges via score and flow matching (**[SF]$^2$ M**) (Tong et al., 2023) (c.f. Sec. H.1). We tested our method on non-conservative systems inducing diverse types of invariant geometries: **(a.)** a Van der Pol system, **(b.)** an out-of-equilibrium process with harmonic trapping and circulation and a Gaussian repulsive obstacle in the centre introduced in Frishman and Ronceray (2020), **(c.)** a Hopf system, and **(d.)** a Selkov glycolysis model (Selkov, 1968) (see Sec. H). For most settings, the proposed framework outperformed existing methods, especially for large inter-observation intervals (Table 2 and 1).

We quantified the quality of the inference in terms of weighted root mean square error (**wRMSE**) between the estimated and ground truth drift functions evaluated on a $d-$dimensional grid spanning the state space volume of the observations. The weights for each grid point were obtained from a kernel density estimation of the observations. Thus misalignment of ground truth and estimated dynamics were penalised stronger for regions of the state space visited more frequently by the observed process.

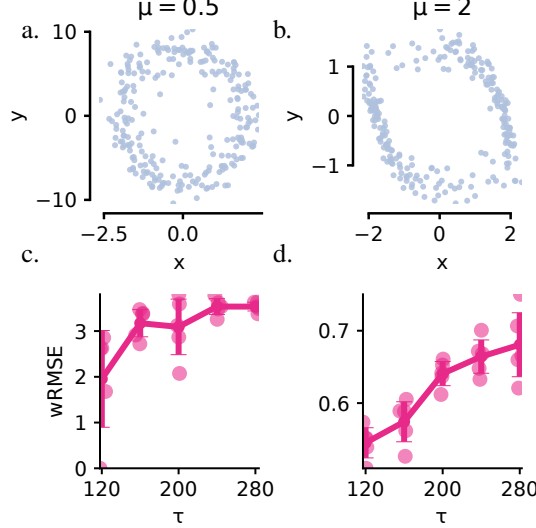

**Figure 4**

Geometry-aware inference provides accurate drift estimation for different empirical manifold geometries resulting from different parameter regimes of the Van der Pol system. **(a.-b.)** Empirical manifold for the Van der Pol system with different $\mu$ parameters. Notice the different scales on the axes. **(c.-d.)** Inference performance of the proposed framework against inter-observation interval $\tau$. Error bars indicate one standard deviation over five independent runs.

| wRMSE $\downarrow$ | Out of equilibrium system | | | Hopf | | | Selkov | |
|---|---|---|---|---|---|---|---|---|
| | $\tau = 150$ | $\tau = 200$ | $\tau = 250 \times dt$ | $\tau = 200$ | $\tau = 300$ | $\tau = 400 \times dt$ | $\tau = 100$ | $\tau = 200 \times dt$ |
| GP | $2.632 \pm 0.007$ | $3.387 \pm 0.012$ | $3.733 \pm 0.011$ | $0.781 \pm 0.006$ | $0.969 \pm 0.015$ | $1.069 \pm 0.006$ | $0.550 \pm 0.021$ | $0.682 \pm 0.040$ |
| SVISE | $35.204 \pm 39.888$ | $3.462 \pm 0.129$ | $7.540 \pm 7.602$ | $2.113 \pm 0.658$ | $4.960 \pm 2.687$ | $3.936 \pm 1.063$ | $5.793 \pm 0.028$ | $2.028 \pm 0.045$ |
| LatentSDE | $\mathbf{2.348} \pm 0.032$ | $\mathbf{2.340} \pm 0.047$ | $\mathbf{2.356} \pm 0.042$ | $1.168 \pm 0.052$ | $1.161 \pm 0.053$ | $1.173 \pm 0.046$ | $0.742 \pm 0.022$ | $0.747 \pm 0.021$ |
| **Geometric (ours)** | $2.762 \pm 0.132$ | $3.034 \pm 0.143$ | $2.693 \pm 0.992$ | $\mathbf{0.210} \pm 0.013$ | $\mathbf{0.237} \pm 0.010$ | $\mathbf{0.255} \pm 0.028$ | $\mathbf{0.414} \pm 0.245$ | $\mathbf{0.682} \pm 0.071$ |

**Table 2**

Performance comparison in terms of wRMSE for the considered frameworks for three different nonlinear dynamical systems and for increasing inter-observation interval $\tau$. Numbers indicate mean wRMSE and standard deviation of five independent runs for each setting.

For a system with a drift function following Van der Pol dynamics, we found that only after two E.M. iterations, the estimated force field (red arrows) is well aligned to the true force field that generated the observations (grey arrows) (Fig. 2a.). For comparison we demonstrate also the result of the estimation with Gaussian likelihood (GP), which results in a flow field orthogonal to the ground truth one.

We performed systematic estimations for this system under different noise conditions $\sigma$, observed at different inter-observation intervals $\tau$ for different lengths of trajectories $T$ (see Sec. H). For the examined noise amplitudes (Fig. 2 f.), the proposed path augmentation algorithm improves the naive estimation with Gaussian assumptions within two iterations (Fig. 2). For increasing noise the improvement contributed by our approach decreases (Fig. 2f.), as the invariant geometry is less well defined, but is still considerable.

**Impact of the geometry of empirical manifold.** We performed inference for different parameter values of the Van der Pol system ($\mu = 1$ (as above) and $\mu = 0.5$ and $\mu = 2$), that result in asymmetries of the invariant density (Fig. 4). We observed that the performance of all inference frameworks deteriorates for increasing asymmetry (larger dynamic range along one dimension), yet our method still delivered more accurate predictions compared to the other considered frameworks. Approximating the invariant geometry with a different metric learning method does not confer any considerable performance difference for our approach (c.f. Table 1 Geometric$_{RBF}$ where we employed the metric introduced in Arvanitidis et al. (2021) and further developed in Kapusniak et al. (2024), where a diagonal metric is approximated in terms positive linear combination of Gaussian RBFs centred at selected cluster centres.

**Impact of noise amplitude.** For systems with small dynamical noise (small $\sigma$), geodesics approximate the manifold structure better, however the path integral control is limited by the control costs proportional to inverse noise covariance. Our framework had comparable accuracy for all inter-observation lengths, but improvement was small for small lengths since in that setting the estimation with Gaussian likelihood already provides a good approximation of the ground truth drift.

We compared our method to the approach proposed in Batz et al. (2018). In this work, the authors perform augmentation with Ornstein-Uhlenbeck bridges, i.e. assuming linear underlying dynamics. We found that our approach delivered more accurate estimates for larger inter-observation intervals. For inter-observation intervals with only one observation per oscillation period (Fig. 3c.,d.), our approach delivered better results by considering additionally knowledge of the direction of movement in the state space (c.f. Sec. H). The variance of estimates of the proposed method was smaller compared to Batz et al. due to consistency imposed by conditioning on the invariant geometry of the system. Predictions improve with longer observation intervals $T$, and for decreasing noise amplitude $\sigma$. In both settings the invariant geometry is more well approximated by the empirical manifold.

State estimation with linear (Ornstein-Uhlenbeck) dynamics (Batz et al., 2018), is in general less capable of correctly estimating the latent system state and subsequently correctly approximating the unknown drift function especially as the length of the inter-observation interval $\tau$ increases.

**Effects of noise miss-estimation.** We further investigated the impact of noise misestimation on the accuracy of drift inference (S.I. Fig. 5). Our findings indicate that after two augmentations conditioned on the invariant geometry, small inaccuracies in the employed dynamical noise during the simulation of augmented paths have a negligible effect on the overall accuracy of the inferred drift. In particular, for small inter-observation intervals, the inference procedure remains highly

robust to misestimated noise amplitudes. As the inter-observation intervals increase, the effect of noise deviations on performance remains minimal, provided the noise used in the augmentation deviates by at most $\pm 0.1$ from the true noise amplitude. Thus, stochastic dynamics may still be identified even with inaccurate or misestimated diffusion constants.

Additional results are provided in the Supplement (see Sec. G).

## 5    DISCUSSION

Discovering unknown driving forces governing stochastic systems poses still a significant challenge, despite extensive existing research on that frontier. Our work demonstrates the benefits of integrating information from both the temporal and geometric structure of the observed data. Our findings showed a substantial improvement in estimating the underlying stochastic dynamics, especially in sparsely sampled, nonlinear systems driven by non-conservative forces.

We introduced **geometric inductive biases** into inference of stochastic systems by treating the deterministic flow field as a scaffold upon which system states fluctuate. We approximated this scaffold in terms of **distortions of a metric induced by the system's measurements**. This approach effectively approximates the low-dimensional invariant density (empirical manifold) without the need to project to a lower dimensional space, whose dimensionality would be hard to estimate due to the presence of fluctuations. The key insight is that **geodesics** computed on the empirical manifold with respect to the approximated metric constitute the **most probable path** of the unknown system between consecutive observations in the Onsager-Machlup sense. Using these **geodesics as control constraints**, we formulated a path-augmentation scheme that bridges sparse observations with trajectories consistent with both the temporal order and the geometry of the data.

Widely used inference methods, predominantly developed within the statistics community, often employ path (*data*) augmentation to approximate transition densities between successive observations. However, this approach suffers from several challenges: **1.)** First, the unobserved information between successive observations is an infinite-dimensional object, requiring the solution of a complex and computationally intensive problem (bridge sampling) (Gronau et al., 2017). We addressed this challenging problem using the computationally efficient framework developed in Maoutsa and Opper (2022). **2.)** Second, direct drift estimation from sparse observations results in estimated dynamics that significantly deviate from the ground truth. Thereby consecutive observations of the system have small probability under the law of the estimated SDE. This discrepancy, in turn, leads to several computational difficulties: **i)** Most bridge sampling schemes become too computationally demanding, or even fail, when attempting to generate transition densities between atypical states for the considered stochastic dynamics. For instance, the method of (Maoutsa and Opper, 2021) successfully generates transition densities between atypical states only for conservative systems through a reweighting with Brownian bridge dynamics. Alternatively, an exceedingly large number of samples would be required for accurate numerical approximation. **ii)** Second, iterative algorithms, such as Expectation Maximisation, which exhibit only *local* convergence (Romero et al., 2019), may converge to inaccurate solutions, when the initial estimation significantly deviates from the ground truth.

To overcome these limitations, we proposed incorporating the information ingrained in the local geometric structure of the observations into the state estimation (path augmentation). This approach is motivated by the observation that commonly employed path augmentation methods often yield transition densities that deviate substantially from the true underlying densities when observations are sparse (Fig. 1**E.**). This discrepancy arises from the fact that these approaches rely on trivial stochastic dynamics that fail to adequately capture the curvature of the ground truth transition densities when the observed system is nonlinear (Sec. G.3). Our numerical experiments demonstrate that, indeed, the proposed approach effectively recovers the underlying drift function for systems with steady-state probability currents (Ding et al., 2020).

**Relation to Schrödinger bridge sampling.**    The framework we employed for the augmentation relies on a deterministic particle formulation of the path integral control formalism (Kappen, 2005b). This framework can be connected to the dynamic Schrödinger bridge problem, if we consider transferring probability mass between two Dirac measures or very narrow Gaussians that sit on each observation, considering additionally a potential that constraints the intermittent dynamics similar

to the one considered in Neklyudov et al. (2023a). Thus, in principle, one can employ one of the recently developed alternative frameworks that solve the dynamic Schrödinger bridge problem for path augmentation The recent Bridge and Flow Matching frameworks (Lipman et al., 2022; Albergo et al., 2023; Shi et al., 2023; Liu et al., 2023) correspond to the control problem we formulate in the SI Eq. 32, without the control constraints. In contrast, the Generalised Schrödinger Bridge Matching (GSBM) framework proposed by Liu et al. (2023) uses a cost functional that is equivalent to the controlled cost we employ to construct our augmentations. In this setting, the penalty term corresponds to the geodesic proximity constraint used in our framework. The GSBM could, in principle, replace the particle-based framework we use. However, here, we employed a framework that relies on particle representations of the involved densities, which can be later easily employed to formulate the Monte Carlo approximations of the integrals involved in the Gaussian process inference for the drift (Eq. 42). Yet, the Gaussian variant of the GSBM framework that incorporates time-dependent penalty constraints (analogous to our geodesic constraints), might be an interesting avenue to explore for potential incorporation in our framework (Tong et al., 2023).

Similarly, for approximating the metric induced by the observations, we employed the framework of Arvanitidis et al. (2019), while we could have employed alternative metric learning approaches (Scarvelis and Solomon, 2022; Hauberg et al., 2012; Barua et al., 2025; Gruffaz and Sassen, 2025). However, the framework of Arvanitidis et al. (2019) perfectly fits the purposes of our work, because it employs a non-parametric (kernel) estimation for approximating the metric and computes the geodesics through GP regression. This allows to evaluate the geodesic equation at different increments, that is necessary for imposing the time dependent geodesic constraint. A similar metric approximation has been recently employed in Kapusniak et al. (2024) for metric flow matching, i.e., for augmentation that respects the geometry of the dataset. While our approach has a similar flavour to this work, our framework additionally requires the augmented data to be temporary ordered and to respect the stochastic flow of the estimated system. This results in learning a global drift that approximates the underlying stochastic dynamics, instead of learning a local drift that transports a snapshot of states from some initial to a final configuration.

**Limitations.** The proposed approach relies on the geometric characterisation of the invariant density of the system's dynamics. This requires sufficiently long observation windows to accurately characterise said density and correctly approximate the unobserved paths with geodesic curves. Thus, our approach is limited to systems where the invariant density can be approximated by a manifold where we can identify geodesics. An alternative method worth exploring would consider the learned invariant metric directly in the dynamics of the augmented process. Moreover, we have considered here inference of stochastic differential equations with known state independent diffusion. While this approach might seem limited, several processes with state dependent diffusion functions can be transformed into processes with state independent diffusions (Beskos et al., 2006a; Roberts and Stramer, 2001) through the Lamberti transform if they fulfil the appropriate conditions for the drift function.

## 6 REPRODUCIBILITY STATEMENT

We have taken several steps to ensure the reproducibility of our results. A detailed description of our methodology, including the inference framework and the geometry-aware path augmentation procedure, is provided in Section 2 of the main text and further elaborated in Appendix A. All theoretical aspects of our work, including the construction of the invariant metric, geodesics, and the stochastic control formulation, are presented in full in the supplementary material (Appendix A.3, A.3.2, and H). The implementation details of the Expectation–Maximisation scheme and Gaussian process inference are also included in the appendix. Our numerical experiments, benchmarks, and additional analyses (e.g., noise misestimation) are reported in the Supplement.

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

## SUPPLEMENTARY INFORMATION

## A   DRIFT INFERENCE FOR HIGH AND LOW FREQUENCY OBSERVATIONS

Effective dynamics of systems with many degrees of freedom or inherently stochastic are often described in terms of a stochastic differential equation (SDE)

$$\mathrm{d}\mathbf{X}_t = \mathbf{f}(\mathbf{X}_t)\mathrm{d}t + \boldsymbol{\eta}(t)\mathrm{d}t = \mathbf{f}(\mathbf{X}_t)\mathrm{d}t + \boldsymbol{\sigma}\mathrm{d}\mathbf{W}_t, \tag{13}$$

where the drift $\mathbf{f}(\cdot) : \mathcal{R}^d \to \mathcal{R}^d$ describes the deterministic forces acting on the system, while the delta-correlated Gaussian white noise term $\boldsymbol{\eta}(t)$, $\langle\boldsymbol{\eta}(t)\boldsymbol{\eta}(t')\rangle = \boldsymbol{\sigma}\delta(t - t')$ describes the effect of stochastic forces as a product of a diffusion matrix (or constant) $\boldsymbol{\sigma} : \mathcal{R}^{d\times d}$ that accounts for the magnitude of the stochastic forces acting on the system, and a $d$-dimensional Wiener process $\mathbf{W}_t$ that contributes random influences.

Often the detailed equation that governs the evolution of the state of the system is unknown. Therefore, understanding a system of interest often requires identification from time series observations of its state. In more practical terms, given some **prior probability** for the drift function, we want to compute the **posterior probability** $\mathrm{P}(\mathbf{f}|\{\mathcal{O}_k\}_{k=1}^K)$ that identifies the unknown drift function of Eq. 13 that most likely gave rise to the observations of the system state $\{\mathcal{O}_k\}_{k=1}^K$. The exact relationship between the observations and the system state will be defined more precisely in the following.

When a system is observed nearly continuously (inter-observation interval length $\tau$ much smaller than the characteristic time scale of the system $\tau \ll \tau_{\mathrm{char}}$), temporal methods regress the system state $\mathbf{X}_t$ against the state increments $\mathbf{Y}_t \doteq \frac{\mathbf{X}_{t+\tau} - \mathbf{X}_t}{\tau}$ to identify the drift function (Friedrich and Peinke, 1997; Ragwitz and Kantz, 2001). In a Bayesian framework, this corresponds to Gaussian process regression with a Gaussian likelihood (SI A.1). However, for large inter-observation intervals $\tau$, these methods fail (Batz et al., 2018), as the Gaussian likelihood assumption is invalid for general nonlinear systems with sparse observations (Fig.1**C.**). In such cases, the likelihood is a *path integral* over continuous trajectories of the unobserved process (SI A.2), making Gaussian-based estimates inaccurate (Fig. 1**C.**).

This underwhelming performance has motivated the development of methods that combine state estimation (or **path augmentation**) and dynamical inference. These methods reconstruct continuous paths to approximate transition densities between observations, enabling inference by estimating the system's state between observations. However, for large time intervals, transition densities are usually analytically intractable, except in a few trivial cases of scalar or linear processes. As a result, the prevailing strategy is to approximate transition densities by sampling marginal distributions of **diffusion bridges**, which are diffusion processes constrained by their initial and terminal states (Golightly and Wilkinson, 2008; Papaspiliopoulos et al., 2012; Sermaidis et al., 2013; Beskos et al., 2006b; Chib et al., 2006). Yet, existing methods employ path augmentation with simplified bridge dynamics (e.g., Brownian (Chib et al., 2006; Golightly and Wilkinson, 2008) or Ornstein-Uhlenbeck bridges (Batz et al., 2018)) that do not accurately reflect the underlying transition densities for nonlinear systems (Fig. 1**E.**).

An alternative path augmentation strategy would obtain a coarse drift estimate, typically achieved by assuming a Gaussian likelihood between observations (see SI Eq. 16), and would subsequently employ a stochastic bridge sampler (De Bortoli et al., 2021; Maoutsa and Opper, 2022; 2021) to construct stochastic bridges using the coarsely estimated nonlinear drift. However, for large inter-observation intervals, the coarsely estimated drift function often deviates significantly from the true function that generated the observations. Consequently, the observations frequently fall into low-

probability regions of the estimated diffusion dynamics (Fig. 1 **E.**), rendering the construction of diffusion bridges either too computationally demanding or impossible (Liu et al., 2020).

## A.1 HIGH FREQUENCY OBSERVATIONS

In an optimal but rather practically unrealistic scenario, we would observe the system (path) $\mathbf{X}_{0:T}$ in (nearly) continuous time, and thus we would try to identify the drift from $P(\mathbf{f}|X_{0:T})$. In such a case, the infinitesimal transition probabilities of the diffusion process between consecutive time-points are Gaussian, i.e.,

$$\mathrm{P}_f(\mathbf{X}_{0:T} \mid \mathbf{f}) \propto \exp\left(-\frac{1}{2\mathrm{d}t}\sum_t \|\mathbf{X}_{t+\mathrm{d}t} - \mathbf{X}_t - \mathbf{f}(\mathbf{X}_t)\mathrm{d}t\|_D^2\right). \tag{14}$$

Here we have introduced the weighted norm $\|\mathbf{u}\|_D \dot{=} \mathbf{u}^\top \cdot \mathbf{D}^{-1} \cdot \mathbf{u}$, with $\mathbf{D} \dot{=} \boldsymbol{\sigma}\boldsymbol{\sigma}^\top$ indicating the noise covariance.

In turn, the transition probabilities of a discretised drift-less process (a Wiener path) $P_{\mathcal{W}}(\mathbf{X}_{0:T})$ with same diffusion $\sigma$ is

$$\mathrm{P}_{\mathcal{W}}(\mathbf{X}_{0:T}) = \exp\left(-\frac{1}{2\mathrm{d}t}\sum_t \|\mathbf{X}_{t+\mathrm{d}t} - \mathbf{X}_t\|_D^2\right). \tag{15}$$

We can thus express the likelihood for the drift $f$ as the likelihood ratio between the transition probabilities of Eq. 14 and Eq. 15, which for diffusion processes is expressed by the Radon-Nykodym derivative between $\mathrm{P}_f(\mathbf{X}_{0:T}|f)$ and $\mathrm{P}_{\mathcal{W}}(\mathbf{X}_{0:T})$ for paths $\mathbf{X}_{0:T}$ within the time interval $[0, T]$ (Liptser and Shiryaev, 2013)

$$\mathcal{L}(\mathbf{X}_{0:T} \mid \mathbf{f}) = \exp\left(-\frac{1}{2}\sum_t \|\mathbf{f}(\mathbf{X}_t)\|_D^2 \mathrm{d}t + \sum_t \langle\mathbf{f}(\mathbf{X}_t), \mathbf{X}_{t+\mathrm{d}t} - \mathbf{X}_t\rangle_D\right), \tag{16}$$

where for brevity we have introduced the notation $\langle\mathbf{u}, \mathbf{v}\rangle_D \dot{=} \mathbf{u}^\top \cdot \mathbf{D}^{-1} \cdot \mathbf{v}$ for the weighted inner product with respect to the inverse noise covariance $\mathbf{D}^{-1}$. This expression results from applying the Girsanov theorem on the path measures induced by a process with drift $\mathbf{f}$ and a Wiener process, with same diffusion $\sigma$, and employing an Euler-Maruyama discretisation on the continuous path $\mathbf{X}_{0:T}$.

The likelihood of a continuously observed path of the SDE (Eq. 16) has a quadratic form in terms of the drift function. Therefore a Gaussian measure over function values (Gaussian process) is a natural conjugate prior for this likelihood. Thus, to identify the drift in a non-parametric form, we assume a Gaussian process prior for the function values $\mathbf{f} \sim \mathrm{P}_0(\mathbf{f}) = \mathcal{GP}(\mathbf{m}^f, k^f)$, where $\mathbf{m}^f$ and $k^f$ denote the mean and covariance function of the Gaussian process (Ruttor et al., 2013). The prior measure can be written as

$$\mathrm{P}_0(\mathbf{f}) = \exp\left(-\frac{1}{2}\int\int \mathbf{f}(\mathbf{x})\left(k^f(\mathbf{X}, \mathbf{X}')\right)^{-1}\mathbf{f}(\mathbf{X}')\mathrm{d}\mathbf{X}\mathrm{d}\mathbf{X}'\right), \tag{17}$$

if we consider a zero mean Gaussian process $\mathbf{m}^f = \mathbf{0}$.

Bayesian inference for the drift function $\mathbf{f}$ requires the computation of a probability distribution in the function space, the posterior probability distribution $\mathrm{P}_f(\mathbf{f} \mid \mathbf{X}_{0:T})$. From the Bayes' rule the posterior can be written as

$$\mathrm{P}_f(\mathbf{f} \mid \mathbf{X}_{0:T}) = \frac{\mathrm{P}_0(\mathbf{f})\mathcal{L}(\mathbf{X}_{0:T} \mid \mathbf{f})}{Z} \propto \mathrm{P}_0(\mathbf{f})\mathcal{L}(\mathbf{X}_{0:T} \mid \mathbf{f}), \tag{18}$$

where $Z$ denotes a normalising factor defined as

$$Z = \int \mathrm{P}_0(\mathbf{f})\mathcal{L}(\mathbf{X}_{0:T} \mid \mathbf{f})\mathcal{D}\mathbf{f}, \tag{19}$$

where $\mathcal{D}\mathbf{f}$ denotes integration over the Hilbert space $\mathbf{f} : H_0[\mathbf{f}] < \infty$. Here we have expressed the prior probability over functions as $\mathrm{P}_0(\mathbf{f}) = e^{-H_0[\mathbf{f}]}$. In Ruttor et al. (2013) the authors show that in this continuous-time setting, nonparametric estimation of the drift can be attained through a Gaussian process regression (Rasmussen, 2003) with the objective to identify the mapping from

the system state $\mathbf{X}_t$ to state increments $\mathrm{d}\mathbf{X}_t$. More precisely, we consider as the regressor the $N$ observations of the system state $\mathbf{X}_t$ and as the associated response variables the state increments

$$\mathbf{Y}_t = \frac{\mathbf{X}_{t+\mathrm{d}t} - \mathbf{X}_t}{\mathrm{d}t}, \tag{20}$$

and select the kernel function of the Gaussian process as $k^f(\mathbf{X}, \mathbf{X}')$.

If we denote with $\mathcal{X} = \{\mathbf{X}_t\}_{t=0}^{T-\mathrm{d}t}$ and $\mathcal{Y} = \{\mathbf{Y}_t\}_{t=0}^{T-\mathrm{d}t}$ the set of state observations and observation increments, the mean of the posterior process over drift functions $\mathbf{f}$ can be expressed as

$$\bar{\mathbf{f}}(\mathbf{x}) = k^f(\mathbf{x}, \mathcal{X})^\top \left( \mathcal{K} + \frac{\mathbf{D}}{\mathrm{d}t} I_N \right)^{-1} \mathcal{Y}, \tag{21}$$

where we abused the notation and denoted with $k^f(\mathbf{x}, \mathcal{X})$ the vector resulting from evaluating the kernel $k^f$ at points $\mathbf{x}$ and $\{\mathcal{O}_k\}_{k=1}^{K-1}$. Similarly $\mathcal{K} = k^f(\mathcal{X}, \mathcal{X})$ stands for the $(K-1) \times (K-1)$ matrix resulting from evaluation of the kernel on all observation pairs. In a similar vein, the posterior variance can be written as

$$\Sigma^2(\mathbf{x}) = k^f(\mathbf{x}, \mathbf{x}) - k^f(\mathbf{x}, \mathcal{X})^\top \left( \mathcal{K} + \frac{\mathbf{D}}{\mathrm{d}t} \right)^{-1} k^f(\mathbf{x}, \mathcal{X}), \tag{22}$$

where the term $\mathbf{D}/\mathrm{d}t$ plays the role of observation noise.

## A.2 LOW FREQUENCY OBSERVATIONS

As the inter-observation interval increases (*low frequency observations*), the validity of the Gaussian likelihood used in drift estimation diminishes as the transition density is no longer Gaussian. Consequently, methods for drift estimation with Gaussian assumptions (Friedrich and Peinke, 1997; Ruttor et al., 2013) become increasingly inaccurate. To discount the effects of low frequency sampling, Lade (Lade, 2009) proposed a method to compute finite-time corrections for drift estimates, which has been mainly applied to one-dimensional problems (Honisch and Friedrich, 2011). In parallel, the statistics community has proposed path augmentation techniques that involve sampling with a simplified system's dynamics between time-consecutive observations to augment the observed trajectory to a nearly continuous-time path (Golightly and Wilkinson, 2008; Papaspiliopoulos et al., 2012; Sermaidis et al., 2013; Beskos et al., 2006b; Chib et al., 2006). However, for large inter-observation intervals and nonlinear systems, the augmented trajectories match poorly the underlying path statistics and these methods often exhibit poor convergence rates or fail to identify the correct dynamics (Figure 1 c. and d.). We note that path augmentation using Ornstein-Uhlenbeck bridges and local linearisation of the **ground truth** dynamics provides a reasonable approximation of the underlying transition density up to a certain inter-observation interval. Nevertheless, during inference, the ground truth dynamics is unknown, and the proposed local linearisations based on inaccurate drift estimates (Batz et al., 2018) perform poorly in this sparsely sampled regime.

As the inter-observation interval $\tau$ increases, if the system is nonlinear, the likelihood assumed between two consecutive observations is no longer Gaussian, but is rather expressed as a *path integral*

$$P(\mathcal{O}_{1:K} \mid \mathbf{f}) = \int P(\mathcal{O}_{1:K} \mid \mathbf{X}_{0:T}) P(\mathbf{X}_{0:T} \mid \mathbf{f}) \mathcal{D}(\mathbf{X}_{0:T}), \tag{23}$$

where $\mathcal{O}_{1:K} \doteq \{\mathcal{O}_k\}_{k=1}^{K}$ identifies the set of $K$ observations collected within the interval $[0, T]$, $P(\mathbf{X}_{0:T} \mid \mathbf{f})$ the prior path probability resulting from a diffusion process with drift $\mathbf{f}(\mathbf{x})$, $\mathcal{D}(\mathbf{X}_{0:T})$ identifies the formal volume element on the path space, and $P(\mathcal{O}_{1:K} \mid \mathbf{X}_{0:T})$ stands for the likelihood of observations given the latent path $\mathbf{X}_{0:T}$.

However, the path integral of Eq. 23 is in general intractable for nonlinear systems. thus we need to simultaneously estimate the drift and latent state of the diffusion process, i.e., to approximate the joint posterior measure of latent paths and drift functions $P(\mathbf{X}_{0:T}, \mathbf{f} \mid \mathcal{O}_{1:K})$. Therefore we consider the unobserved continuous path $\mathbf{X}_{0:T}$ as latent random variables and employ an Expectation Maximisation (EM) algorithm to identify

a maximum a posteriori estimate for the drift function. More precisely, we follow an iterative algorithm, where at each iteration $n$ we alternate between the two following steps: An **Expectation** step, where given a drift estimate $\hat{\mathbf{f}}^n(\mathbf{x})$ we construct an approximate posterior over the latent variables $Q(\mathbf{X}_{0:T}) \approx \mathrm{P}(\mathbf{X}_{0:T} \mid \boldsymbol{\mathcal{O}}_{1:K}, \hat{\mathbf{f}}^n(\mathbf{x}))$, and compute the expected log-likelihood of the augmented path

$$\mathfrak{L}\big(\hat{\mathbf{f}}^n(\mathbf{x}), Q\big) = \mathbb{E}_Q\Big[\ln \mathscr{L}\big(\mathbf{X}_{0:T}, \boldsymbol{\mathcal{O}}_{1:K} \mid \hat{\mathbf{f}}^n(\mathbf{x})\big)\Big]. \tag{24}$$

A **Maximisation** step, where we update the drift estimation by maximising the expected log likelihood

$$\mathbf{f}^{n+1}(\mathbf{x}) = \arg\max_f \Big[\mathfrak{L}\big(\mathbf{f}^n(\mathbf{x}), Q\big) - \ln \mathrm{P}_0\big(\mathbf{f}^n(\mathbf{x})\big)\Big]. \tag{25}$$

In Eq. 25, $\mathrm{P}_0$ denotes the Gaussian process prior over function values.

### A.3 Approximate posterior over paths.

To obtain an approximate posterior over the latent paths we perform **variational inference** (Beal, 2003). In this section, we first formulate the approximate posterior over paths (conditional distribution for the path given the observations) by considering only individual observations as constraints (Section A.3.1). However, this approach results computationally taxing calculations during path augmentation, since the observations are atypical states of the initially estimated drift. To overcome this issue, we subsequently extend the formalism (Section A.3.2) to incorporate constraints that consider also the local geometry of the observations.

### A.3.1 Approximate posterior over paths without geometric constraints

Given a drift function (or a drift estimate) $\hat{\mathbf{f}}(\mathbf{x})$ we can apply variational techniques to approximate the posterior measure over the latent path conditioned on the observations $\boldsymbol{\mathcal{O}}_{1:K}$. We consider that the **prior process** (the process without considering the observations $\boldsymbol{\mathcal{O}}_{1:K}$) is described by the equation

$$\mathrm{P}(\mathbf{X}_{0:T} \mid \hat{\mathbf{f}}) : \qquad \mathrm{d}\mathbf{X}_t = \hat{\mathbf{f}}(\mathbf{X}_t)\mathrm{d}t + \sigma \mathrm{d}\mathbf{W}_t. \tag{26}$$

We will define an approximating (posterior) process that is conditioned on the observations. The conditioned process is also a diffusion process with the same diffusion as Eq. 26 but with a modified, time-dependent drift $g(x, t)$ that accounts for the observations (Chetrite and Touchette, 2015; Majumdar and Orland, 2015). We identify the approximate posterior measure $Q$ with the posterior measure induced by an approximating process that is conditioned by the observations $\boldsymbol{\mathcal{O}}_{1:K}$ (Opper, 2019), with governing equation

$$Q(\mathbf{X}_{0:T}) : \qquad \mathrm{d}\mathbf{X}_t = \mathbf{g}(\mathbf{X}_t, t)\mathrm{d}t + \sigma \, \mathrm{d}W_t = \Big(\hat{\mathbf{f}}(\mathbf{X}_t) + \mathbf{u}(\mathbf{X}_t, t)\Big)\mathrm{d}t + \sigma \, \mathrm{d}\mathbf{W}_t. \tag{27}$$

The effective drift $\mathbf{g}(\mathbf{X}_t, t)$ of Eq. 27 may be obtained from the solution of the variational problem of minimising the free energy

$$\mathcal{F}[Q] = \mathcal{KL}\Big(Q(\mathbf{X}_{0:T})\|\mathrm{P}(\mathbf{X}_{0:T} \mid \hat{\mathbf{f}})\Big) - \sum_{k=1}^{K} \Big\langle \ln \mathrm{P}(\boldsymbol{\mathcal{O}}_k \mid \mathbf{X}_{t_k}) \Big\rangle_Q. \tag{28}$$

By applying the Cameron-Girsanov-Martin theorem we can express the Kullback-Leibler divergence between the two path measures induced by the diffusions with drift $\hat{\mathbf{f}}(\mathbf{x})$ and $\mathbf{g}(\mathbf{x}, t)$ as

$$\mathcal{KL}\Big(Q(\mathbf{X}_{0:T})||\mathrm{P}(\mathbf{X}_{0:T}|\hat{\mathbf{f}})\Big) = \left\langle \ln\left(\frac{\mathrm{d}Q(\mathbf{X}_{0:T})}{\mathrm{d}P\left(\mathbf{X}_{0:T}|\hat{\mathbf{f}}\right)}\right)\right\rangle_Q \tag{29}$$

$$= \left\langle \left(-\frac{1}{2}\int_0^T \|\hat{\mathbf{f}}(\mathbf{X}_t) - \mathbf{g}(\mathbf{X}_t, t)\|_{\mathbf{D}}^2 \mathrm{d}t + \int_0^T \frac{\hat{\mathbf{f}}(\mathbf{X}_t) - \mathbf{g}(\mathbf{X}_t, t)}{\mathbf{D}}\mathrm{d}\mathbf{W}_t\right)\right\rangle_Q$$

$$= \left\langle \left(-\frac{1}{2}\int_0^T \|\hat{\mathbf{f}}(\mathbf{X}_t) - \mathbf{g}(\mathbf{X}_t, t)\|_{\mathbf{D}}^2 \mathrm{d}t + V_T\right)\right\rangle_Q \tag{30}$$

$$= \frac{1}{2}\int_0^T \int \|\mathbf{g}(\mathbf{x}, t) - \hat{\mathbf{f}}(\mathbf{x})\|_{\mathbf{D}}^2\, q_t(\mathbf{x})\, \mathrm{d}\mathbf{x}\, \mathrm{d}t + \mathfrak{C}, \tag{31}$$

where $q_t(\mathbf{x})$ stands for the marginal density for $\mathbf{X}_t$ of the approximate process. In the third line we have introduced the random variable $V_T = \int_0^T \frac{\hat{\mathbf{f}}(\mathbf{X}_t) - \mathbf{g}(\mathbf{X}_t, t)}{\mathbf{D}}\mathrm{d}\mathbf{W}_t$. Under the assumption that the function $\ell(\mathbf{X}_t) = \hat{\mathbf{f}}(\mathbf{X}_t) - \mathbf{g}(\mathbf{X}_t, t)$ is bounded, piece-wise continuous, and in $L^2[0, \infty)$, $V_T$ follows the distribution $\mathcal{N}\left(V_T \mid 0, \int_0^T \ell^2(s)\mathrm{d}s\right)$, which for a given $T$ will result into a constant $\mathfrak{C}$. Thus the second term in Eq. 31 is not relevant for the minimisation of the free energy and will be omitted.

We can thus express the free energy of Eq. 28 as (Opper, 2019)

$$\mathcal{F}[Q] = \frac{1}{2}\int_0^T \int \left[\|\mathbf{g}(\mathbf{x}, t) - \hat{\mathbf{f}}(\mathbf{x})\|_{\mathbf{D}}^2 + U(\mathbf{x}, t)\right] q_t(\mathbf{x})\, \mathrm{d}\mathbf{x}\, \mathrm{d}t, \tag{32}$$

where the term $U(\mathbf{x}, t)$ accounts for the observations $U(\mathbf{x}, t) = -\sum_{t_k} \ln \mathrm{P}(\mathcal{O}_k \mid \mathbf{x})\, \delta(t - t_k)$.

The minimisation of the functional of the free energy can be construed as a stochastic control problem (Opper, 2019) with the objective to identify a time-dependent drift adjustment $\mathbf{u}(\mathbf{x}, t) := \mathbf{g}(\mathbf{x}, t) - \hat{\mathbf{f}}(\mathbf{x})$ for the system with drift $\hat{\mathbf{f}}(\mathbf{x})$ so that the controlled dynamics fulfil the constraints imposed by the observations.

### A.3.2 APPROXIMATE POSTERIOR OVER PATHS WITH GEOMETRIC CONSTRAINTS

The previously described construction of the approximate measure in terms of stochastic bridges is relevant when the observations have non vanishing probability under the law of the prior diffusion process of Eq. 26. However, when the prior process (with the estimated drift $\hat{f}$) differs considerably from the process that generated the observations, such a construction might either provide a bad approximation of the underlying path measure, or show slow numerical convergence in the construction of the diffusion bridges. To overcome this issue, we consider here additional constraints for the posterior process that force the paths of the posterior measure to respect the local geometry of the observations. In the following we provide a brief introduction on the basics of Riemannian geometry and consequently continue with the geometric considerations of the proposed method.

**Riemannian geometry.** A $d$-dimensional **Riemannian manifold** (Do Carmo and Flaherty Francis, 1992; Lee, 2018) $(\mathcal{M}, \mathfrak{h})$ embedded in a $d$-dimensional ambient space $\mathcal{X} = \mathcal{R}^d$ is a smooth curved $d$-dimensional surface endowed with a smoothly varying inner product (Riemannian) **metric** $\mathfrak{h} : \mathbf{x} \to \langle\cdot|\cdot\rangle_\mathbf{x}$ on $\mathcal{T}_\mathbf{x}\mathcal{M}$. A tangent space $\mathcal{T}_\mathbf{x}\mathcal{M}$ is defined at each point $\mathbf{x} \in \mathcal{M}$. The Riemannian metric $\mathfrak{h}$ defines a canonical volume measure on the manifold $\mathcal{M}$. Intuitively this characterises how to compute inner products locally between points on the tangent space of the manifold $\mathcal{M}$, and therefore determines also how to compute norms and thus distances between points on $\mathcal{M}$.

A **coordinate chart** $(G, \phi)$ provides the mapping from an open set $G$ on $\mathcal{M}$ to an open set $V$ in the Euclidean space. The dimensionality of the manifold is $d$ if for each point $\mathbf{x} \in \mathcal{M}$ there exists

a local neighborhood $G \subset \mathcal{R}^d$. We can represent the metric $\mathfrak{h}$ on the local chart $(G, \phi)$ by the positive definite matrix (**metric tensor**) $H(\mathbf{x}) = (\mathfrak{h}_{i,j})_{\mathbf{x}, 0 \le i,j, \le d} = \left( \langle \frac{\partial}{\partial x_i} | \frac{\partial}{\partial x_j} \rangle_{\mathbf{x}} \right)_{0 \le i,j, \le d}$ at each point $\mathbf{x} \in G$.

For $\mathbf{v}, \mathbf{w} \in \mathcal{T}_{\mathbf{x}} \mathcal{M}$ and $\mathbf{x} \in G$, their inner product can be expressed in terms of the matrix representation of the metric $\mathfrak{h}$ on the tangent space $\mathcal{T}_{\mathbf{x}} \mathcal{M}$ as $\langle \mathbf{v} | \mathbf{w} \rangle_{\mathbf{x}} = \mathbf{v}^\top H(\mathbf{x}) \mathbf{w}$, where $H(\mathbf{x}) \in \mathcal{R}^{d \times d}$.

The **length of a curve** $\gamma : [0,1] \to \mathcal{M}$ on the manifold is defined as the integral of the norm of the tangent vector

$$\ell(\gamma_{t'}) = \int_0^1 \|\dot{\gamma}_{t'}\|_{\mathfrak{g}} \mathrm{d}t' = \int_0^1 \sqrt{\dot{\gamma}_{t'}^\top H(\gamma_{t'}) \dot{\gamma}_{t'}} \mathrm{d}t', \tag{33}$$

where the dotted letter indicates the velocity of the curve $\dot{\gamma}_{t'} = \partial_{t'} \gamma_{t'}$. A **geodesic curve** is a locally length minimising smooth curve that connects two given points on the manifold.

**Riemannian geometry of observations.** For approximating the posterior over paths we take into account the geometry of the invariant density as it is represented by the observations. To that end, we consider systems whose dynamics induce invariant (inertial) manifolds that contain the global attractor of the system and on which system trajectories concentrate (Wiggins, 1994; Mohammed and Scheutzow, 1999; Girya and Chueshov, 1995; Fenichel and Moser, 1971; Arnold, 1990; Carverhill, 1985). We assume thus that the continuous-time trajectories $\mathbf{X}_{0:T} \in \mathcal{R}^d$ of the underlying system concentrates on an invariant manifold $\mathcal{M} \in \mathcal{R}^{m \le d}$ of dimensionality $m$ (possibly) smaller than $d$. The discrete-time observations $\mathcal{O}_k$ are thus samples of the manifold $\mathcal{M}$. The central premise of our approach is that **unobserved paths between successive observations will be lying either *on* or *in the vicinity* of the manifold** $\mathcal{M}$. In particular, we postulate that unobserved paths should lie **in the vicinity of geodesics that connect consecutive observations** on $\mathcal{M}$. To that end we propose a path augmentation framework that constraints the augmented paths to lie in the vicinity of identified geodesics between consecutive observations.

However, while this view of a lower dimensional manifold embedded in a higher dimensional ambient space helps to build our intuition for the proposed method, for computational purposes we adopt a complementary view inspired by the discussion in (Fröhlich et al., 2021). According to this view, we consider the entire observation space $\mathcal{R}^d$ as a smooth Riemannian manifold, $\mathcal{M} \dot{=} \mathcal{R}^d$, characterised by a Riemannian metric $\mathfrak{h}$. The effect of the nonlinear geometry of the observations is then captured by the metric $\mathfrak{h}$. Thus to approximate the geometric structure of the system's invariant density, we learn the Riemannian metric tensor $H : \mathcal{R}^d \to \mathcal{R}^{d \times d}$ and compute the geodesics between consecutive observations according to the learned metric. Intuitively according to this view the observations $\{\mathcal{O}_k\}_{k=1}^K$ introduce distortions in the way we compute distances on the state space.

In effect this approach does not reduce the dimensionality of the space we operate, but changes the way we compute inner products and thus distances, lengths, and geodesic curves on $\mathcal{M}$. The alternative perspective of working on a lower dimensional manifold would strongly depend on the correct assessment of the dimensionality of said manifold. For example, one could use a Variational Autoencoder to approximate the observation manifold and subsequently obtain the Riemannian metric from the embedding of the manifold mediated by the decoder. However, our preliminary results of such an approach revealed that such a method requires considerable fine tuning to adapt to the characteristics of each dynamical system and is sensitive to the estimation of the dimensionality of the approximated manifold.

To learn the Riemannian metric and compute the geodesics we follow the framework proposed by Arvanitidis et al. in (Arvanitidis et al., 2019). In particular, we approximate the local metric induced by the observations at location $\mathbf{x}$ of the state space, in a non-parametric form by the inverse of the weighted local diagonal covariance computed on the observations as (Arvanitidis et al., 2019)

$$H_{dd}(\mathbf{x}) = \left( \sum_{i=1}^K w_i(\mathbf{x}) \left( x_i^{(d)} - x^{(d)} \right)^2 + \epsilon \right)^{-1}, \tag{34}$$

with weights $w_i(\mathbf{x}) = \exp\left( -\frac{\|\mathbf{x}_i - \mathbf{x}\|_2^2}{2\sigma_{\mathcal{M}}^2} \right)$, and $x^{(d)}$ denoting the $d$-th dimensional component of the vector $\mathbf{x}$. The parameter $\epsilon > 0$ ensures non-zero diagonals of the weighted covariance matrix, while $\sigma_{\mathcal{M}}$ characterises the curvature of the manifold.

Between consecutive observations for each interval $[\mathcal{O}_k, \mathcal{O}_{k+1}]$, we identify the geodesic $\gamma_{t'}^k$ as the energy minimising curve, i.e., as the minimiser of the kinetic energy functional $\mathcal{E}(\gamma_{t'}^k) = \int_0^1 L_\mathcal{M}(\gamma_{t'}^k, \dot{\gamma}_{t'}^k) \, dt'$

$$\gamma_{t'}^{k*} = \underset{\gamma_{t'}^k, \gamma_0^k = \mathcal{O}_k, \gamma_1^k = \mathcal{O}_{k+1}}{\arg\min} \int_0^1 L_\mathcal{M}(\gamma_{t'}^k, \dot{\gamma}_{t'}^k) \, dt',$$

$$\text{with} \quad \int_0^1 L_\mathcal{M}(\gamma_{t'}^k, \dot{\gamma}_{t'}^k) dt' = \frac{1}{2} \int_0^1 \|\dot{\gamma}_{t'}^k\|_\mathfrak{h}^2, \tag{35}$$

where $L_\mathcal{M}(\gamma_{t'}^k, \dot{\gamma}_{t'}^k)$ denotes the Lagrangian. The minimising curve of this functional is the same as the minimiser of the curve length functional $\ell(\gamma_{t'})$ (Eq. 33), i.e., the geodesic (Do Carmo and Flaherty Francis, 1992).

By applying calculus of variations, the minimising curve of the functional $\mathcal{E}(\gamma_{t'}^k)$ can be obtained from the Euler-Lagrange equations, resulting in the following system of second order differential equations (Arvanitidis et al., 2017; Do Carmo and Flaherty Francis, 1992)

$$\ddot{\gamma}_t^k = -\frac{1}{2} H(\gamma_t^k)^{-1} \left( 2 \left( I \otimes (\dot{\gamma}_t^k)^\top \right) \frac{\partial \text{vec}[H(\gamma_t^k)]}{\partial \gamma_t^k} \dot{\gamma}_t^k - \frac{\partial \text{vec}[H(\gamma_t^k)]^\top}{\partial \gamma_t^k} \left( \dot{\gamma}_t^k \otimes \dot{\gamma}_t^k \right) \right), \tag{36}$$

with boundary conditions $\gamma_0^k = \mathcal{O}_k$ and $\gamma_1^k = \mathcal{O}_{k+1}$, where $\otimes$ stands for the Kroenecker product, and $\text{vec}[A]$ denotes the vectorisation operation of matrix $A$ through stacking the columns of $A$ into a vector. We follow Arvanitidis et al. (2019) and obtain the geodesics by approximating the solution of the boundary value problem of Eq. 36 with a probabilistic differential equation solver.

**Extended free energy functional.** We denote the collection of individual geodesics by $\Gamma_t \doteq \{\gamma_{t'}^k\}_{t=(k-1)\tau+t'\tau}$, where $\gamma_{t'}^k$ is the geodesic connecting $\mathcal{O}_k$ and $\mathcal{O}_{k+1}$, and $t' \in [0,1]$ denotes a rescaled time variable. Additional to the constraints imposed in the previously explained setting (Sec A.3.1), here we add an extra term in the free energy $U_\mathcal{G}(\mathbf{x}, t) \doteq \|\Gamma_t - \mathbf{x}\|^2$ that accounts for the local geometry of the invariant density, and guides the latent path towards the geodesic curves $\gamma_{t'}^k$ that connect consecutive observations

$$\mathcal{F}[Q] = \frac{1}{2} \int\limits_0^T \int \left[ \|g(\mathbf{x}, t) - \hat{f}(\mathbf{x})\|_D + U_\mathcal{O}(\mathbf{x}, t) + \beta U_\mathcal{G}(\mathbf{x}, t) \right] q_t(\mathbf{x}) \, d\mathbf{x} \, dt. \tag{37}$$

Here we denote the observation term by $U_\mathcal{O}(\mathbf{x}, t) \doteq -\sum_{t_k} \ln \mathrm{P}(\mathcal{O}_k | \mathbf{x}) \delta(t - t_k)$, while $\beta$ stands for a weighting constant that determines the relative weight of the geometric term in the control objective.

Following (Opper, 2019), for each inter-observation interval $[\mathcal{O}_k, \mathcal{O}_{k+1}]$ we identify the posterior path measure (minimiser of Eq. 37) by the solution of a stochastic optimal control problem (Maoutsa and Opper, 2022) with the objective to obtain a time-dependent drift adjustment $\mathbf{u}(\mathbf{x}, t) := \mathbf{g}(\mathbf{x}, t) - \hat{\mathbf{f}}(\mathbf{x})$ for the system with drift $\hat{\mathbf{f}}(\mathbf{x})$ with initial and terminal constraints defined by $U_\mathcal{O}(\mathbf{x}, t)$, and additional path constraints $U_\mathcal{G}(\mathbf{x}, t)$.

For the case of exact observations, i.e., for an observation process $\psi(\mathbf{x}) = \mathbf{x}$, we can compute the drift adjustment for each of the $K - 1$ inter-observation intervals independently. Thus for each interval between consecutive observations, we identify the optimal control $\mathbf{u}(\mathbf{x}, t)$ required to construct a stochastic bridge following the dynamics of Eq. 26 with initial and terminal states the respective observations $\mathcal{O}_k$ and $\mathcal{O}_{k+1}$.

The optimal drift adjustment for such a stochastic control problem for the inter-observation interval between $\mathcal{O}_k$ and $\mathcal{O}_{k+1}$ can be obtained from the solution of the backward equation (see (Maoutsa and Opper, 2022))

$$\frac{\partial \phi_t(\mathbf{x})}{\partial t} = -\mathcal{L}_{\hat{f}}^\dagger \phi_t(\mathbf{x}) + U_\mathcal{G}(\mathbf{x}, t) \phi_t(\mathbf{x}), \tag{38}$$

with terminal condition $\phi_T(\mathbf{x}) = \chi(\mathbf{x}) = \delta(\mathbf{x} - \mathcal{O}_{k+1})$ and with $\mathcal{L}_{\hat{f}}^\dagger$ denoting the adjoint Fokker-Planck operator for the process of Eq. 26. As shown in (Maoutsa and Opper, 2022) the optimal drift

adjustment $\mathbf{u}(\mathbf{x}, t)$ can be expressed in terms of the difference of the logarithmic gradients of two probability flows

$$\mathbf{u}^*(\mathbf{x}, t) = D\Big(\nabla \ln q_{T-t}(\mathbf{x}) - \nabla \ln \rho_t(\mathbf{x})\Big), \tag{39}$$

where $\rho_t$ fulfils the forward (filtering) partial differential equation (PDE)

$$\frac{\partial \rho_t(\mathbf{x})}{\partial t} = \mathcal{L}_{\hat{f}} \rho_t(\mathbf{x}) - U_{\mathcal{G}}(\mathbf{x}, t) \rho_t(\mathbf{x}), \tag{40}$$

while $q_t$ is the solution of a time-reversed PDE that depends on the logarithmic gradient of $\rho_t(\mathbf{x})$

$$\frac{\partial q_t(\mathbf{x})}{\partial t} = -\nabla \cdot \left[ \Big(\sigma^2 \nabla \ln \rho_{T-t}(\mathbf{x}) - \mathbf{f}(\mathbf{x}, T-t)\Big) q_t(\mathbf{x}) \right] + \frac{\sigma^2}{2} \nabla^2 q_t(\mathbf{x}), \tag{41}$$

with initial condition $q_0(\mathbf{x}) \propto \rho_T(\mathbf{x}) \chi(\mathbf{x})$ .

For the numerical solution of the control problem we use the numerical framework accompanying Maoutsa and Opper (2022), where the path constraints associated with the geodesic curves are imposed through the two staged process for particle propagation described in the paper for path constraints, with the particle reweighting being performed through optimal transport implemented using the PyEMD python toolbox (Pele and Werman, 2009).

More precisely, according to this framework we propagate a particle representation of the probability density $\rho_t(\mathbf{x})$ according to the filtering equation of Eq. 40. This follows the dynamics of the uncontrolled process with drift $\hat{\mathbf{f}}$ and particle reweighting at each time step as determined by the path constrained (potential) $U_{\mathcal{G}}(\mathbf{x}, t)$, that quantifies the proximity to the geodesic at each time point. In the particle representation we apply this reweighting in the form of a deterministic optimal transportation of the particles

(Reich, 2013)

### A.4 Approximate posterior over drift functions.

For a fixed path measure $Q$, the optimal measure for the drift $Q_f$ is a Gaussian process given by

$$Q_f \propto \mathrm{P}_f \exp\left(-\frac{1}{2} \int \|\mathbf{f}(\mathbf{x})\|_D^2 A(\mathbf{x}) - 2\langle \mathbf{f}(\mathbf{x}), B(\mathbf{x})\rangle_D \, \mathrm{d}\mathbf{x}\right), \tag{42}$$

with

$$A(\mathbf{x}) \doteq \int_0^T q_t(\mathbf{x}) \mathrm{d}t,$$

and

$$B(\mathbf{x}) \doteq \int_0^T q_t(\mathbf{x}) g(\mathbf{x}, t) \mathrm{d}t,$$

where $q_t(\mathbf{x})$ denotes the marginal constrained density of the state $\mathbf{X}_t$. The function $\mathbf{g}(\mathbf{x}, t)$ denotes the effective drift.

We assume a Gaussian process prior for the unknown function $\mathbf{f}$, i.e., $\mathbf{f} \sim \mathrm{P}_0(\mathbf{f}) = \mathcal{GP}(\boldsymbol{m}^f, k^f)$ where $m^f$ and $k^f$ denote the mean and covariance function of the Gaussian process. Following Ruttor et al. (Ruttor et al., 2013), we employ a sparse kernel approximation for the drift $f$ by optimising the function values over a sparse set of $S$ inducing points $\{Z_i\}_{i=1}^S$. We obtain the resulting drift from

$$\hat{\mathbf{f}}_S(\mathbf{x}) = k^{\mathbf{f}}(\mathbf{x}, \mathcal{Z})\left(I + \Lambda \mathcal{K}_S\right)^{-1} \mathbf{d}, \tag{43}$$

where we have defined introduced the notation $\mathcal{K}_S \doteq k^f(\mathcal{Z}, \mathcal{Z})$

$$\Lambda = \frac{1}{\sigma^2} \mathcal{K}_S^{-1} \left(\int k^f(\mathcal{Z}, \mathbf{x}) A(\mathbf{x}) k^f(\mathbf{x}, \mathcal{Z}) \mathrm{d}\mathbf{x}\right) \mathcal{K}_S^{-1}. \tag{44}$$

$$\mathbf{d} = \frac{1}{\sigma^2} \mathcal{K}_S^{-1} \left(\int k^f(\mathcal{Z}, \mathbf{x}) B(\mathbf{x}) \mathrm{d}\mathbf{x}\right) \mathcal{K}_S^{-1}, \tag{45}$$

The associated variance results similarly from the equation

$$\Sigma_S^2(\mathbf{x}) = k^f(\mathbf{x}, \mathbf{x}) - k^f(\mathbf{x}, \mathcal{Z})\, (I + \Lambda\, \mathcal{K}_S)^{-1}\, \Lambda\, k^f(\mathcal{Z}, \mathbf{x}). \tag{46}$$

We employ a sample based approximation of the densities in Eq. 42 resulting from the particle sampling of the path measure $Q$ resulting from the geometric augmentation, i.e. the integrals over $\int q_t(\mathbf{x})$ are over the samples of the augmented paths. Thus by representing the densities by samples, we can rewrite the density $p_t(x)$ in terms of a sum of Dirac delta functions centered around the particles positions

$$p_t(\mathbf{x}) \approx \frac{1}{N} \sum_{j=1}^{N} \delta(\mathbf{x} - \mathbf{X}_j(t)),$$

and replace the Riemannian integrals with summation over particles, i.e. perform a Monte Carlo integration. Here $\mathbf{X}_j(t)$ represents the position of the $j$-th particle at time point $t$.

# B    SPARSE GAUSSIAN PROCESS ESTIMATION

Since the amount of required observations for accurate drift estimation is generally large for systems with nonlinear dynamics, regular Gaussian process regression becomes computationally intensive. Its computational complexity scales as $\mathcal{O}(N^3)$ with the number of observations $N$ due to the $N \times N$ kernel matrix inversions required for inference (c.f. Eq.22 and Rasmussen (2003)). Therefore, Ruttor et al. (2013) employ the sparse (low dimensional approximation) counterpart of Gaussian process regression (Titsias, 2009; Csató and Opper, 2002) that reduces significantly the computation time by reducing the computational complexity to $\mathcal{O}(NM^2)$, where $M \ll N$ denotes the number of selected sparse (inducing) points. Here we present briefly the derivation.

For sparse Gaussian process drift inference, we augment the distributions with $M$ inducing points $\mathbf{z} = [z_1, \ldots, z_M]$ with inducing values $\mathbf{u} = [\mathbf{f}(z_m)]_{m=1}^{M}$ that are jointly Gaussian distributed with the latent function values $\{\mathbf{f}(\mathbf{X}_t)\}_{t=0}^{T}$.

As demonstrated previously the true posterior for function values $\mathbf{f}$ is expressed as a product

$$P_f(\mathbf{f}) = \frac{1}{Z} P_o(\mathbf{f}) e^{-\mathcal{A}(\mathbf{f})}, \tag{47}$$

where $Z$ a normalisation constant, $\mathcal{A}(\mathbf{f}) = \frac{1}{2}\mathbf{f}^T \Lambda \mathbf{f} - \mathbf{a}^T \mathbf{f}$ a quadratic form of $\mathbf{f}$ (see Eq. 16), while $P_o(\mathbf{f})$ denotes a prior Gaussian measure. Thus the posterior $P_f(\mathbf{f})$ is also Gaussian. In Eq. 47 $\Lambda \doteq \text{diag}[\Delta t\, D^{-1}, \ldots, \Delta t\, D^{-1}]$, and $\mathbf{a}^T = [\mathbf{D}^{-1}\Delta X_0, \ldots, \mathbf{D}^{-1}\Delta X_{T-1}]$.

To employ sparse Gaussian process inference, we approximate $P_f$ with $Q_f = \mathcal{GP}\left(m^q(\cdot), k^q(\cdot, \cdot)\right)$, with mean and variance functions to be calculated, depending only on the *smaller* subset ($M \ll N$) of inducing function values $\mathbf{u}$,

$$Q_f(\mathbf{f}) \propto R(\mathbf{u}) P_o(\mathbf{f}). \tag{48}$$

The effective likelihood $R(\mathbf{u})$ is chosen as the minimiser of the Kullback-Leibler divergence $\mathcal{KL}\left(Q_f \| P_f\right)$.

We may now express the prior $P_o(\mathbf{f})$ and the approximate marginal $Q_f(\mathbf{f})$ in terms of the inducing points

$$P_o(\mathbf{f}) = P_o(\mathbf{f}|\mathbf{u}) P_o(\mathbf{u}), \tag{49}$$

and

$$Q_f(\mathbf{f}) = Q_f(\mathbf{f}|\mathbf{u}) Q_f(\mathbf{u}) = P_o(\mathbf{f}|\mathbf{u}) Q_f(\mathbf{u}), \tag{50}$$

under the assumption that the posterior conditional $Q_f(\mathbf{f}|\mathbf{u})$ matches the prior conditional $P_o(\mathbf{f}|\mathbf{u})$.

We select the effective likelihood $R(u)$ as the minimiser of the relative entropy between $Q_f$ and $P_f$

$$
\begin{aligned}
\mathcal{KL}\left(Q_f || P_f\right) &= \int Q_f(\mathbf{f}) \ln \frac{Q_f(\mathbf{f})}{P_f(\mathbf{f})} \mathrm{d}\mathbf{f} \\
&= \int P_o(\mathbf{f}|\mathbf{u}) Q_f(\mathbf{u}) \ln \frac{P_o(\mathbf{f}) R(\mathbf{u})}{\frac{1}{Z} P_o(\mathbf{f}) e^{-\mathcal{A}(\mathbf{f})}} \mathrm{d}\mathbf{f}\mathrm{d}\mathbf{u} \\
&= \int P_o(\mathbf{f}|\mathbf{u}) Q_f(\mathbf{u}) \ln \frac{P_o(\mathbf{f}) R(\mathbf{u})}{\frac{1}{Z} P_o(\mathbf{f}|\mathbf{u}) e^{-\mathcal{A}(\mathbf{f}|\mathbf{u})} P_o(\mathbf{u})} \mathrm{d}\mathbf{f}\mathrm{d}\mathbf{u} \\
&= \int P_o(\mathbf{f}|\mathbf{u}) Q_f(\mathbf{u}) \ln \frac{P_o(\mathbf{u}) R(\mathbf{u})}{\frac{1}{Z} e^{-\mathcal{A}(\mathbf{f}|\mathbf{u})} P_o(\mathbf{u})} \mathrm{d}\mathbf{f}\mathrm{d}\mathbf{u} \\
&= \int P_o(\mathbf{f}|\mathbf{u}) Q_f(\mathbf{u}) \ln \frac{R(\mathbf{u})}{\frac{1}{Z} e^{-\mathcal{A}(\mathbf{f}|\mathbf{u})}} \mathrm{d}\mathbf{f}\mathrm{d}\mathbf{u} \\
&= \ln Z + \int Q_f(\mathbf{u}) \ln \left( \frac{e^{\ln R(\mathbf{u})}}{e^{-\mathbb{E}_o[\mathcal{A}(\mathbf{f}|\mathbf{u})]}} \right) \mathrm{d}\mathbf{u}.
\end{aligned}
\tag{51}
$$

In Eq. 51 in the second line, we have introduced Eq. 47-Eq. 50. In the third line we have introduced $\frac{P_0(\mathbf{f})}{P_0(\mathbf{f}|\mathbf{u})} = P_0(\mathbf{u})$ from Eq. 49. In the final line we rearranged the terms that do not depend on $\mathbf{f}$ outside of the integral over $\mathbf{f}$, moved the $\ln Z$ term out of the integration over $\mathbf{u}$, and denoted $\mathbb{E}_0[\cdot] = \int P_0(\mathbf{f}|\mathbf{u}) \, \mathrm{d}\mathbf{f}$.

To minimise the relative entropy $\mathcal{KL}\left[Q_f || P_f\right]$ we conclude that the optimal choice for the effective likelihood $R(\mathbf{u})$ is

$$
R(\mathbf{u}) \propto e^{-\mathbb{E}_o[\mathcal{A}(\mathbf{f}|\mathbf{u})]}.
\tag{52}
$$

Given the quadratic form of $A(\mathbf{f})$ we may write the conditional expectation in Eq. 52 as a quadratic form too

$$
\begin{aligned}
\mathbb{E}_o\left[\mathcal{A}(\mathbf{f}|\mathbf{u})\right] &= \frac{1}{2} \mathbb{E}_o\left[\mathbf{f}|\mathbf{u}\right]^T \Lambda \mathbb{E}_o\left[\mathbf{f}|\mathbf{u}\right] + \frac{1}{2} \mathrm{Tr}\left(\mathrm{Cov}_o[\mathbf{f}|\mathbf{u}]\Lambda\right) - a^T \mathbb{E}_o\left[\mathbf{f}|\mathbf{u}\right] \\
&= \frac{1}{2} \mathbb{E}_o\left[\mathbf{f}|\mathbf{u}\right]^T \Lambda \mathbb{E}_o\left[\mathbf{f}|\mathbf{u}\right] - a^T \mathbb{E}_o\left[\mathbf{f}|\mathbf{u}\right] + \mathrm{const.},
\end{aligned}
\tag{53}
$$

where in the last line we take into account that the term $\mathrm{Tr}\left(\mathrm{Cov}_o[\mathbf{f}|\mathbf{u}]\Lambda\right)$ is independent of the sparse function values $\mathbf{u}$ (c.f. Ruttor et al. (2013)).

In particular, the conditional expectation of function values $f$ conditioned on the inducing point function values $\mathbf{u} \equiv \mathcal{U}$ at inducing point locations $\mathbf{z} \equiv \mathcal{Z}$ equals

$$
\bar{f}^s(\mathbf{x}) = \mathbb{E}_o\left[f|\mathbf{u}\right] = k(\mathbf{x}, \mathcal{Z}) k(\mathcal{Z}, \mathcal{Z})^{-1} \mathcal{U},
\tag{54}
$$

while the covariance equals

$$
(\Sigma^s)^2(\mathbf{x}) = k(\mathbf{x}, \mathbf{x}) - k(\mathbf{x}, \mathcal{Z}) k(\mathcal{Z}, \mathcal{Z})^{-1} k(\mathcal{Z}, \mathbf{x}),
\tag{55}
$$

where we have employed similar notation for the kernel functions as in Eqs. 21-22.

## C   THEORETICAL EVIDENCE THAT MAY SUPPORT THE USE OF GEODESICS AS GEOMETRIC CONSTRAINTS

The Onsager-Machlup functional for diffusion processes has been known in theoretical physics as a characteriser of the most probable path (MPP) between two pre-defined states of the process. In (Onsager and Machlup, 1953), Onsager and Machlup used the thermal fluctuations of a diffusion process to show that the probability density of a path $\boldsymbol{\gamma} \in C^1\left([0, T], \mathcal{R}^d\right)$ in $\mathcal{R}^d$ over finite interval can be expressed as a Boltzmann factor

$$
\mathrm{P}(\boldsymbol{\gamma}) \sim \exp\left[-\int_0^T L(\boldsymbol{\gamma}(t), \dot{\boldsymbol{\gamma}}(t)) dt\right],
\tag{56}
$$

where

$$L(\boldsymbol{\gamma}(t), \dot{\boldsymbol{\gamma}}(t)) = \frac{1}{2}\|\frac{\dot{\boldsymbol{\gamma}}(t) - \mathbf{f}(\boldsymbol{\gamma}(t))}{\mathbf{D}}\|^2 + \frac{1}{2}\nabla \cdot \mathbf{f}(\boldsymbol{\gamma}(t)).[1] \tag{57}$$

The function $L(\boldsymbol{\gamma}(t), \dot{\boldsymbol{\gamma}}(t))$ is known as the **Onsager-Machlup** function (action), while its integral over time is known as Onsager-Machlup action functional. It has been used as Lagrangian in Euler-Lagrange minimisation schemes to identify the **most probable path (MPP)** of a diffusion process between two given points in the state space (Graham, 1977; Stratonovich, 1971).

Stratonovich (Stratonovich, 1971) considered the probability that a sample of a multidimensional diffusion process will lie in the vicinity of (within a tube of infinitesimal thickness around) an idealised smooth path in the state space. To compute this probability he constructed a probability functional which is identical to the Onsager-Machlup functional considered as Lagrangian for the diffusion process. Duerr et al. (Dürr and Bach, 1978) considered scalar diffusion processes and constructed the Onsager-Machlup function from the asymptotic limit of the transition probability between the starting and end state of the path using a Girsanov transformation.

Considering Brownian motions defined on a Riemannian manifold $(\mathcal{M}, \mathbf{g})$ with associated Riemannian metric $\mathbf{g}$, the Onsager-Machlup functional can be expressed as the integral over the Lagrangian (Takahashi and Watanabe, 1981; Graham, 1980; Grong and Sommer, 2022)

$$L(\boldsymbol{\gamma}, \dot{\boldsymbol{\gamma}}) = \frac{1}{2}\|\dot{\boldsymbol{\gamma}}(t)\|_{\mathbf{g}}^2 - \frac{1}{12}S(\boldsymbol{\gamma}(t)), \tag{58}$$

where $\| \cdot \|_{\mathbf{g}}$ denotes the Riemannian norm on the tangent space $\mathcal{T}_X\mathcal{M}$ of the manifold with respect to the metric $\mathbf{g}$, and $S(\cdot)$ stands for the scalar curvature of the manifold at each point. The first term is the Lagrangian used to identify geodesic curves on manifolds (c.f. A.3.2)

In our proposed formalism, for computational purposes we have assumed the entire $\mathcal{R}^d$ as smooth manifold. We can identify the first term of Eq. 58 with the Lagrangian we optimised for computing the geodesics on the manifold $(\mathcal{R}^d, \mathbf{g})$, where $\mathbf{g}$ is the metric learned from the observations.

However the system we observed was a diffusion process defined in $\mathcal{R}^d$ with an Euclidean metric. Constructing a path augmentation scheme that guides the augmented paths towards the geodesics of a diffusion defined with respect to a different metric raises questions about the validity of our approach. Here we should note that diffusions with a general state dependent diffusion coefficient $\boldsymbol{\sigma} \in \mathcal{R}^{d \times m}$, and $m$-dimensional Brownian motion, can be considered as evolving on the manifold $(\mathcal{R}^d, \mathbf{g})$, with the associated metric $\mathbf{g} = (\boldsymbol{\sigma}\boldsymbol{\sigma}^\top)^{-1}$ (Capitaine, 2000). Thus it may be possible to associate the metric learned from the data with the metric arising from a state dependent diffusion by applying a transformation akin to an inverse Lamperti transform (Øksendal, 2003) to transform our learned SDE to one that would have induced the learned metric due to the state dependent diffusion. The existence of such a transformation would justify the proposed method. Our empirical results demonstrate that such a transformation may be possible.

## D DOES THE PROPOSED APPROACH INVALIDATE THE MARKOVIAN PROPERTY OF THE DIFFUSION PROCESS?

The proposed path augmentation seemingly invalidates the Markovian property of the diffusion process. According to the Markov property of the diffusion of Eq. 1, the system state $\mathbf{X}_{k\tau+\delta t}$ should depend only the state $\mathbf{X}_{k\tau}$, i.e., the observation $\mathcal{O}_k$. The proposed augmentation makes the state $\mathbf{X}_{k\tau+\delta t}$ depending not only on the next observation $\mathcal{O}_{k+1} = \mathbf{X}_{(k+1)\tau}$, but also on past and future states that lie in the vicinity of these observations.

We effectively construct the augmented paths to compute the likelihood of a drift estimate. To compute this likelihood we require to evaluate the transition probabilities between consecutive observations. Since for general nonlinear systems the transition probabilities are in general intractable, we have to resort to numerical approximations. Ideally we would approximate the transition density

---

[1]Onsager and Machlup's initial work concentrated around linear processes and therefore the functional initially introduced by the did not include the second term with the divergence of $\mathbf{f}$ as this is a constant for linear $\mathbf{f}$. It was later added to the OM function to account for trajectory entropy corrections (Taniguchi and Cohen, 2007; Adib, 2008)

with a bridge sampler that would consider the nonlinear estimated SDE conditioned to pass though consecutive observations. However for coarse drift estimates, the observations have zero probability under the law of the estimated SDE, and construction of those bridges would result either in very taxing computations or would fail altogether. Instead, here, we compute the likelihood of a "corrected" estimate (the correction resulting from the invariant density) under which the observations have non-zero probability, and subsequently re-estimate the drift on the augmented path with this "corrected" estimate. By taking into account the local geometry of the observations, we provide systematic corrections for the misestimated drift function to generate the augmented paths. This effectively nudges the augmentation process towards the second observation of each inter-observation interval through the path constraint that forces the augmented paths towards the geodesics.

## E   RELATED WORK AND POSITIONING OF THE PRESENT WORK

Here, we briefly review further related work on inference or modelling of SDEs and position our work further with respect to the existing literature.

▷ **Modelling general SDEs from state observations.**   As already mentioned in the Introduction and in Sec. A existing inference methods for SDEs can be broadly clustered in temporal and geometric methods, where the former accounts for the temporal order of the observations, while the latter approximate the invariant system density and discard any time information.

**Temporal methods** rely on the Euler-Maruyama discretisation of the SDE paths approximating conditional expectations of state increments (i.e. the Krammers Moyal coefficients). They model the drift either in terms of Gaussian processes (Ruttor et al., 2013; Batz et al., 2018; Hostettler et al., 2018; Zhao et al., 2020; Yildiz et al., 2018), basis functions (Nabeel et al., 2025; Ragwitz and Kantz, 2001; Friedrich and Peinke, 1997; Peinke et al., 1997; Friedrich et al., 2000; Ferretti et al., 2020) or libraries of functions (Boninsegna et al., 2018; Huang et al., 2022), kernel regression (Lamouroux and Lehnertz, 2009; Jiang and Knight, 1997), dynamic mode decomposition to learn the eigenfunctions of the Koopman operator (Klus et al., 2020), by approximating the central moments of the transition densities (Stanton, 1997), or by applying generalised methods of moments (Hansen and Scheinkman, 1993).

As explicitly detailed in Sec. A, most temporal methods do not provide accurate drift estimates when the interval between observations is large. The two prevailing approaches to mitigate this finite-sampling rate effects is to either account for the systematic bias introduced by the finite sampling rate by estimating an explicit correction term for the inferred drift (Ragwitz and Kantz, 2001; 2002; Kleinhans et al., 2005; Kleinhans and Friedrich, 2007), or by performing state estimation for the unobserved paths (also known as path or data augmentation) and then estimating the drift from the continuous paths.

The former approach works only for scalar systems, while the latter employs simplified bridge dynamics (e.g., Brownian (Chib et al., 2006; Eraker, 2001; Sermaidis et al., 2013) or Ornstein Uhlenbeck (Batz et al., 2018; Billio et al., 1998) bridges) that are analytically tractable or computationally non-demanding. However, for large $\tau$ and for nonlinear systems, these simplified bridge dynamics match poorly the underlying path statistics. (Fig. 1 **D.**). It is important to mention here, that path augmentation with Ornstein Uhlenbeck bridges similar to Batz et al. (2018) provides a good approximation of the underlying transition density, when the underlying linear process employed for each bridge has a drift that comes from the local linearisation of the **ground truth** drift function. However, during inference the true dynamics are unknown and the local linearisations on inaccurate drift estimates employed in Batz et al. (2018) provide imprecise approximations for large $\tau$.

Alternative methods, employ variational inference (Batz et al., 2016; Opper, 2019; Duncker et al., 2019; Verma et al., 2024) and approximate the posterior path measure with a tractable Gaussian process induced by a time–varying linear SDE. This results in ODEs for the posterior mean and covariance matrix and an ELBO that is optimized directly (Archambeau et al., 2007; Duncker et al., 2019).

Building on the building on a rich line of work on neural ODEs, neural SDEs (Li et al., 2020) employ gradient-based stochastic variational inference and the stochastic adjoint sensitivity method to compute gradients of solutions of stochastic equations with respect to their parameters. Building on

these methods, Course and Nair (2023b) remove the need for adjoint-based gradient computations by combining amortized inference with a reparametrization of the ELBO by assuming a latent linear process that generates the latent path.

**Geometric approaches** on the other hand, discard the temporal structure of the observations, and treat them as samples of the invariant density. Thereby these methods either employ density estimation to identify the drift as the gradient of a potential Kutoyants and Kutojanc (2004), or resort to spectral approximations of the generator of the diffusion process through manifold learning.

Manifold learning methods employ often the *diffusion maps* algorithm, introduced by Coiffman and colleagues Singer and Coifman (2008), to learn the dominant part of the spectrum of the transfer operator of the observed diffusion process Coifman et al. (2005); Nadler et al. (2006); Giannakis (2019); Ferguson et al. (2011); Talmon and Coifman (2015). In essence, these methods, learn from the data the few leading eigenfunctions of the Laplace–Beltrami operator that captures the Riemannian geometry of the observations, and consider them as a parametrisation of the manifold representing the invariant density.

▷ **Modelling SDEs from population level snapshots/boundary conditions.** With the recent computational advances in solving optimal transport problems, a large volume of recent works focuses on implementations of solutions of the Schroedinger bridge sampling problem with potential additional path constraints. These mostly generative methods focus on transporting the data distribution from some initial boundary condition to some terminal one, and most of these learn an stochastic equation to perform this transport through Schroedinger bridge sampling. Generalized Schroedinger Bridge Matching (GSBM) follows an alternating scheme that learns both drift and marginals. Given the boundary conditions, the framework minimises a kinetic term, that is then used in a stochastic optimal control problem conditioned on the initial and terminal conditions and a path cost that accounts for additional constraints. Action matching Neklyudov et al. (2023b) introduces a simulation-free variational objective that identifies a time-dependent scalar potential (entropic action) $s_t$ whose gradient $\nabla s_t$ transports the densities via the continuity from the initial to the boundary condition through the continuity equation. In its entropic formulation the $\nabla s_t$ can be considered as drift of the underlying SDE whose marginal match the boundary conditions. However, by construction, the framework can oly recover gradient drifts and are therefore not suitable for identifying general stochastic systems with rotational dynamics.

**Geometry aware generative methods** MFM generalizes CFM by learning interpolants that account for the geometry of the data. However MFM does not assume a stochastic underlying process, as our framework does, only a deterministic interpolation (transport) that respects the data manifold. However, by assuming a specific noise amplitude for the underlying SDE, one can consider the flow field as generated by the effective drift of a probability flow ODE associated with the considered SDE and make inferences about the underlying drift function.

**Approximating observation geometry in the ambient space.** In our work, we approximate the geometry induced by the observations by endowing the ambient space $\mathcal{R}^d$ with an observation-dependent Riemannian metric $H(x)$ (Eq. 4) that encodes the local anisotropy of the data distribution. In our framework this metric acts as a constraint for data-augmentation and as a geometric inductive bias for drift function inference: augmented paths are encouraged to remain in regions where the metric $H(x)$ induces smaller distances, i.e. in the vicinity of geodesics computed with respect to this metric, thereby aligning the augmented paths with the empirical observation geometry.

This perspective connects to a growing body of work that **approximates Riemannian metrics directly in the ambient space** as a proxy for the unknown curved low-dimensional data manifold, instead of first estimating its intrinsic dimensionality and then constructing explicit low-dimensional embeddings.

In parallel, an increasing body of literature focuses on endowing generative models with geometric constraints or inductive biases. While most methods function in autoencoder-like setting, by learning an embedding function for projecting to a lower dimensional space that respects prescribed or learned geometric constraints Duque et al. (2022); Kalatzis et al. (2020); Arvanitidis et al. (2017) geometry, "Riemannian", similar to our proposed method, operate in the ambient space by directly a Riemannian geometry embedded there and define normalizing flows or other generative processes

directly on the manifold of interest. Mathieu and Nickel (2020) introduce a framework for continuous normalizing flows defined in the ambient space, respecting a prescribed Riemannian geometry. Similarly, De Bortoli et al. (2022) proposed a score-based generative model that models target densities with support on prescribed Riemannian manifolds in terms of a time-reversal of Langevin dynamics.

Metric flow matching (Kapusniak et al., 2024) interpolates data distributions that respect the geodesic interpolants computed according to the metric induced by the observations. They employ data-dependent metrics in the ambient high dimensional space to design interpolants with low kinetic energy under the learned geometry and to constrain generative paths to the data manifold. Our construction is conceptually similar with these approaches in that we also avoid explicit low-dimensional embeddings and instead approximate the observation manifold through a Riemannian metric living in the ambient space. However, in contrast to methods focused on deterministic transport or simulation-free matching, we use the learned metric to regularize continuous-time diffusion bridges and drift inference, so that the recovered stochastic dynamics are geometrically consistent with the observation-induced invariant measure.

**Positioning of the present work.** Our approach combines the nonparametric flexibility of Gaussian-process–based drift inference from time-series data with recent geometric ideas for population-level SDE modeling. Similar to Metric Flow Matching (Kapusniak et al., 2024), we posit that augmented trajectories should remain on the manifold induced by the observations: both frameworks estimate a data-adapted Riemannian metric and construct interpolants (geodesics and bridges) that respect this geometry. The GSBM framework (Liu et al., 2023) employs a stochastic control objective that is similar to the objective we consider for constructing the augmented paths. However, unlike our framework, GSBM does not introduce geometric constraints for the augmented paths. However, the path constraint they consider can be formulated with geometric considerations as we did in our comparisons here. Finally, whereas these methods typically learn a drift that transports a single source distribution to a single terminal snapshot, yielding thus a locally valid dynamics, our method, akin to multi-marginal bridge sampling (Shen et al., 2024), fits a sequence of bridges across multiple time points to recover a **single global drift** consistent with the underlying drift dynamics.

## F  GEOMETRIC CONSTRAINTS ON INFERENCE.

Our method bridges the gap between approaches that rely only on the temporal structure of observations and those that approximate the invariant density, while ignoring temporal order. Motivated by advances in geometric statistics (Miolane et al., 2020; Sommer, 2020), and the growing interest on the concept of manifold hypothesis (Fefferman et al., 2016; Shnitzer et al., 2020), i.e., the consideration that the state of multi-dimensional dynamical systems often resides in low-dimensional regions of the state space, several recent methods integrate geometric and temporal constraints in stochastic system identification. In *Langevin regression* framework (Callaham et al., 2021), the Kramers-Moyal (KM) coefficients are estimated and low sampling effects are accounted for by solving an adjoint Fokker-Planck equation, with regularisation via moment matching (Lade, 2009). Tong et al. (2020) consider the manifold of the observations for inference of cellular dynamics. Their method employs dynamic optimal transport to interpolate between measured distributions constrained to lie in the vicinity of the observations. While sharing similar intuitions with our method, Tong et al. do not employ SDE modelling for inherently stochastic cellular dynamics and do not consider the underlying geometry of the observations, relying solely on constraints penalizing pairwise distances between them. Shnitzer et al. (Shnitzer et al., 2020; 2016) employ diffusion maps to approximate the eigenfunctions of the backward Kolmogorov operator (the generator of the stochastic Koopman operator (Giannakis, 2019; Črnjarić-Žic et al., 2020)). By evolving the dominant operator eigenspectrum with a Kalman filter, they account for the temporal order of observations. However, their approach is limited to conservative systems and requires the presence of a spectral gap in the approximated operator's spectrum.

## G ADDITIONAL RESULTS

### G.1 INFERENCE WITH NOISE MISS-ESTIMATION

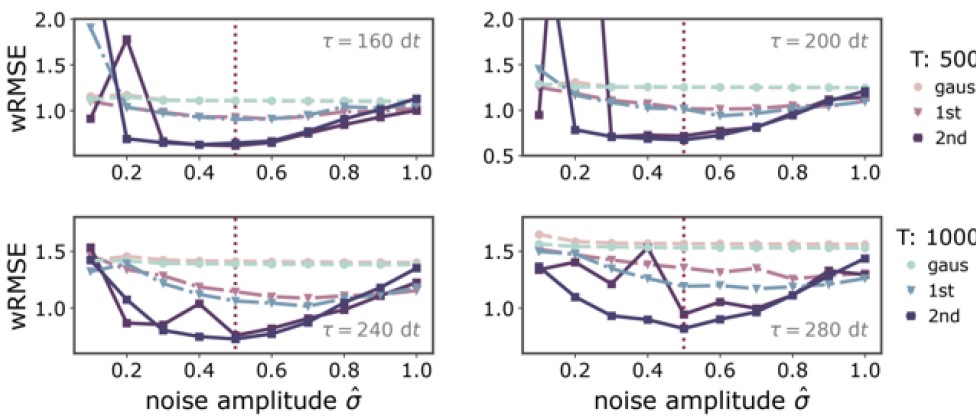

**Figure 5**

**Small noise misestimation has small impact on estimation accuracy.** Weighted root mean square error (wRMSE) vs. noise amplitude $\sigma$ employed in the augmentation for different inter-observation intervals with **a.)** $\tau = 160\,dt$ **b.)** $\tau = 200\,dt$, **c.)** $\tau = 240\,dt$ **d.)** $\tau = 280\,dt$. Pink-purple lines correspond to estimation with total simulation length $T = 500$ time units, and blue markers correspond to total simulation length of $T = 1000$ time units. Red dotted line identifies the noise amplitude employed in the simulation of the observations.

### G.2 INFERENCE PERFORMANCE DETERIORATES WITH INCREASING INTER-OBSERVATION INTERVAL FOR EXISTING FRAMEWORKS

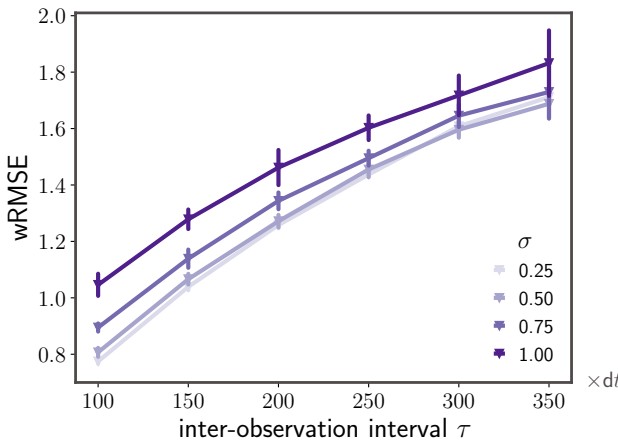

**Figure 6**

**Increasing observation interval between successive observations $\tau$ deteriorates performance quantified by increasing weighted root mean squared error (wRMSE) for Gaussian process-based inference.** Weighted root mean square error between estimated and ground truth drift vector fields for increasing observation interval $\tau$ between subsequent observations for different noise conditions (indicated by different hues). Observations were collected from a Van der Pol oscillator system simulated with $dt = 0.01$ for $T = 500$ time units. Error bars indicate one standard deviation over ten independent realizations.

We computed the weighted root mean square error (wRMSE) between ground truth flow fields and estimated ones for several commonly applied inference frameworks. We observed that the performance of all of them deteriorates once the inter-observation interval becomes large.

We started with the method that motivated our research, approximating drift functions through Gaussian processes, the method outlined in Ruttor et al. (2013). The method approximates the drift

functions with Gaussian process regression, using the system state $\mathbf{X}_t$ as the regressor and state increments as the response variable $\mathbf{Y}_t \doteq \frac{\mathbf{X}_{t+\tau} - \mathbf{X}_t}{\tau}$. This is the Bayesian counterpart of earlier methods encountered in physics literature (Friedrich and Peinke, 1997; Ragwitz and Kantz, 2001), providing additionally uncertainty estimation through the Gaussian process approximation.

As is evident from Figure 6 the discrepancy between ground truth and estimated vector fields increases for increasing temporal distance between successive observations. This should be understood, under the consideration that inference of the drift based on regression on state increments results from an approximation relying on a truncated Ito-Taylor expansion. This is also the starting point of the Euler Maruyama discretisation. As the time interval between successive steps of this approximation increases, the truncated approximation does not longer hold, and higher order terms should be considered.

### G.3 Inference based on Euler-Maruyama discretisation does not account for the curvature of the trajectories in the state space

To be more precise, a general SDE of the form

$$d\mathbf{X}_t = \mathbf{f}(\mathbf{X}_t, t)dt + \boldsymbol{\sigma}(\mathbf{X}_t, t)d\mathbf{W}_t. \tag{59}$$

is a shorthand for the integral equation

$$\mathbf{X}_t = \mathbf{X}_{t_0} + \int_{t_0}^{t} \mathbf{f}(\mathbf{X}_s, s) \, ds + \int_{t_0}^{t} \boldsymbol{\sigma}(\mathbf{X}_s, s) \, d\mathbf{W}_s, \tag{60}$$

where as previously in this manuscript, we consider the stochastic integrals in the **Itô sense**. (Here we start from a more general formulation of the stochastic equation with both diffusion and drift terms being state- and time-dependent to highlight that also for more general SDEs our geometric argument is valid.)

Applying the Itô formula on each integrand, and integrating from $t_0$ to $t$, we obtain the Itô-Taylor expansion of Eq. 59

$$\mathbf{f}(\mathbf{X}_t, t) = \mathbf{f}(\mathbf{X}_{t_0}, t_0) + \int_{t_0}^{t} \frac{\partial \mathbf{f}(\mathbf{X}_s, s)}{\partial s} \, ds + \int_{t_0}^{t} \sum_u \frac{\partial \mathbf{f}(\mathbf{X}_s, s)}{\partial X^{(u)}} f_u(\mathbf{X}_s, s) \, ds$$

$$+ \int_{t_0}^{t} \sum_u \frac{\partial \mathbf{f}(\mathbf{X}_s, s)}{\partial X^{(u)}} \left[ \boldsymbol{\sigma}(\mathbf{X}_s, s) \, d\mathbf{W}_s \right]_u + \int_{t_0}^{t} \frac{1}{2} \sum_{u,v} \frac{\partial^2 \mathbf{f}(\mathbf{X}_s, s)}{\partial X^{(u)} \partial X^{(v)}} \left[ \boldsymbol{\sigma}(\mathbf{X}_s, s) \, \boldsymbol{\sigma}^\top(\mathbf{X}_s, s) \right]_{uv} ds$$

$$= \mathbf{f}(\mathbf{X}_{t_0}, t_0) + \int_{t_0}^{t} \mathcal{L}_s^\dagger \mathbf{f}(\mathbf{X}_s, s) \, ds + \sum_\nu \int_{t_0}^{t} \mathcal{L}_{W,\nu} \mathbf{f}(\mathbf{X}_s, s) \, dW_s^{(\nu)}, \tag{61}$$

and

$$\boldsymbol{\sigma}(\mathbf{X}_t, t) = \boldsymbol{\sigma}(\mathbf{X}_{t_0}, t_0) + \int_{t_0}^{t} \frac{\partial \boldsymbol{\sigma}(\mathbf{X}_s, s)}{\partial s} \, ds + \int_{t_0}^{t} \sum_u \frac{\partial \boldsymbol{\sigma}(\mathbf{X}_s, s)}{\partial X^{(u)}} f_u(\mathbf{X}_s, s) \, ds$$

$$+ \int_{t_0}^{t} \sum_u \frac{\partial \boldsymbol{\sigma}(\mathbf{X}_s, s)}{\partial X^{(u)}} \left[ \boldsymbol{\sigma}(\mathbf{X}_s, s) \, d\mathbf{W}_s \right]_u + \int_{t_0}^{t} \frac{1}{2} \sum_{u,v} \frac{\partial^2 \boldsymbol{\sigma}(\mathbf{X}_s, s)}{\partial X^{(u)} \partial X^{(v)}} \left[ \boldsymbol{\sigma}(\mathbf{X}_s, s) \, \boldsymbol{\sigma}^\top(\mathbf{X}_s, s) \right]_{uv} ds$$

$$= \boldsymbol{\sigma}(\mathbf{X}_{t_0}, t_0) + \int_{t_0}^{t} \mathcal{L}_s^\dagger \boldsymbol{\sigma}(\mathbf{X}_s, s) \, ds + \sum_\nu \int_{t_0}^{t} \mathcal{L}_{W,\nu} \boldsymbol{\sigma}(\mathbf{X}_s, s) \, dW_s^{(\nu)}, \tag{62}$$

where we have used the fact that the product of stochastic differentials due to the Ito isometry and multiplication rules equals the noise covariance times the time step

$$dX_t^{(u)} dX_t^{(v)} = \left[ \boldsymbol{\sigma} \boldsymbol{\sigma}^\top \right]_{uv} dt,$$

where

$$dX_s^{(u)} = f_u \, ds + \sum_{j=1}^{m} \sigma_{uj} \, dW_s^{(j)},$$

while the superscripts/subscripts $u, v$ indicate dimensional components.

In the above equations, we have introduced the operators acting on test-functions $\mathbf{h} : \mathcal{R}^D \to \mathcal{R}^D$

$$\mathcal{L}_t^\dagger \mathbf{h} = \frac{\partial \mathbf{h}}{\partial t} + \sum_u \frac{\partial \mathbf{h}}{\partial X^{(u)}} f_u + \frac{1}{2} \sum_{u,v} \frac{\partial^2 \mathbf{h}}{\partial X^{(u)} \partial X^{(v)}} \left[ \boldsymbol{\sigma}(\mathbf{X}_s, s) \, \boldsymbol{\sigma}^\top(\mathbf{X}_s, s) \right]_{uv} \tag{63}$$

and

$$\mathcal{L}_{W,v} \mathbf{h} = \sum_u \frac{\partial \mathbf{h}}{\partial X^{(u)}} \, \boldsymbol{\sigma}_{uv}, \qquad \text{for } v = 1, \dots, n. \tag{64}$$

With these expressions, the original integral equation for $\mathbf{X}_t$ can be written as

$$\mathbf{X}_t = \mathbf{X}_{t_0} + \mathbf{f}(\mathbf{X}_{t_0}, t_0)(t - t_0) + \boldsymbol{\sigma}(\mathbf{X}_{t_0}, t_0)(\mathbf{W}_t - \mathbf{W}_{t_0}) + \tag{65}$$

$$R_1 = \begin{cases} + \int_{t_0}^t \int_{t_0}^s \mathcal{L}_u^\dagger \mathbf{f}(\mathbf{X}_u, u) \, \mathrm{d}u \, \mathrm{d}s + \sum_\nu \int_{t_0}^t \int_{t_0}^s \mathcal{L}_{W,\nu} \mathbf{f}(\mathbf{X}_u, u) \, \mathrm{d}W_u^{(\nu)} \, \mathrm{d}s \\ + \int_{t_0}^t \int_{t_0}^s \mathcal{L}_u^\dagger \boldsymbol{\sigma}(\mathbf{X}_u, u) \, \mathrm{d}u \, \mathrm{d}\mathbf{W}_s + \sum_\nu \int_{t_0}^t \int_{t_0}^s \mathcal{L}_{W,\nu} \boldsymbol{\sigma}(\mathbf{X}_u, u) \, \mathrm{d}W_u^{(\nu)} \, \mathrm{d}\mathbf{W}_s. \end{cases}$$

In the last equation, dropping the terms in the remainder $R_1$ results in the Euler–Maruyama integration scheme (Jentzen and Kloeden, 2011). Introducing the discrete time and noise increments

$$\Delta t_n = t_{n+1} - t_n = \int_{t_n}^{t_{n+1}} \mathrm{d}s, \quad \Delta \mathbf{W}_n = \mathbf{W}_{t_{n+1}} - \mathbf{W}_{t_n} = \int_{t_n}^{t_{n+1}} \mathrm{d}\mathbf{W}_s, \tag{66}$$

we result in the discrete time equation commonly used for numerical integration of SDEs

$$\mathbf{X}_{n+1} = \mathbf{X}_n + \mathbf{f}(\mathbf{X}_n, t_n) \, \Delta t_n + \boldsymbol{\sigma} \, \Delta \mathbf{W}_n. \tag{67}$$

This is also the starting point of most inference methods that employ the regression scheme mentioned above by approximating the drift as

$$\hat{\mathbf{f}}(\mathbf{X}_n, t_n) \approx \frac{\mathbf{X}_{n+1} - \mathbf{X}_n}{\Delta t} \sim \mathcal{N}\left( \mathbf{0}, \frac{\boldsymbol{\sigma} \, \boldsymbol{\sigma}^\top}{\Delta t} \right). \tag{68}$$

This discretisation is a zero-order approximation of the true dynamics, and assumes that $\mathbf{f}(\cdot)$ remains constant throughout the interval $\Delta t$, i.e. throughout the inter-observation interval $\tau$ in the inference setting. However as $\tau$ increases, higher-order terms in the remainder $R_1$ of the Itô-Taylor expansion become significant, since the assumption that the drift is approximately constant over $\tau$ does not hold.

We can glean onto the terms that become important once the inter-observation interval becomes large, by applying the Itô formula onto each one of the integrands in $R_1$ separately **for the specific setting we consider in this manuscript**, i.e. that of time-independent drift function $\mathbf{f}(\mathbf{x})$ and constant diffusion matrix $\boldsymbol{\sigma}$. In the following, we demonstrate that the leading-order error of this approximation is governed by the intrinsic geometry of the drift vector field. This provides further insight and a geometric explanation for the deterioration of inference methods for increasing inter-observation interval $\tau$.

In short we show that, inference methods based on the Euler-Maruyama discretisation-based inference effectively assume that the vector field between consecutive observations $\mathbf{X}_n$ and $\mathbf{X}_{n+1}$ does not change. Our analysis shows this is equivalent to assuming trajectories are straight lines ($\mathbf{J}_f \mathbf{f} \parallel \mathbf{f}$) and the Itô correction is constant. In reality, trajectories curve ($\mathbf{J}_f \mathbf{f}$ has also a perpendicular component), and this curvature itself changes along the vector field. The Euler-Maruyama discretisation-based inference scheme systematically misses these higher-order geometric features, leading to biased drift estimates.

### G.3.1 First remainder term $R_{1,a}$

We denote the first term of the reminder by $R_{1,a}$

$$R_{1,a} = \int_{t_0}^t \int_{t_0}^s \mathcal{L}_u^\dagger \mathbf{f}(\mathbf{X}_u) \, \mathrm{d}u \, \mathrm{d}s. \tag{69}$$

Applying Itô's formula to the integrand $\mathcal{L}_t^\dagger \mathbf{f}(\mathbf{X}_u, u)$, we get

$$d\mathcal{L}_u^\dagger \mathbf{f}(\mathbf{X}_u) = \frac{\partial}{\partial u}\mathcal{L}_u^\dagger \mathbf{f}(\mathbf{X}_u)\, du + \sum_{j=1}^d \frac{\partial \mathcal{L}_u^\dagger \mathbf{f}}{\partial X^{(j)}}(\mathbf{X}_u)\, dX_u^{(j)} + \frac{1}{2}\sum_{j,k=1}^d \frac{\partial^2 \mathcal{L}_u^\dagger \mathbf{f}}{\partial X^{(j)}\partial X^{(k)}}(\mathbf{X}_u)\,\left[\boldsymbol{\sigma}\boldsymbol{\sigma}^\top\right]_{jk}\, du. \tag{70}$$

Plugging in the original SDE $dX_u^{(j)} = f_j\, du + \sum_{\nu=1}^m \sigma_{j\nu}\, dW_u^{(\nu)}$, and integrating over the time from $t_0$ to $u$

$$\mathcal{L}_u^\dagger \mathbf{f}(\mathbf{X}_u) = \mathcal{L}_{t_0}^\dagger \mathbf{f}(\mathbf{X}_{t_0}) + \int_{t_0}^u \left( \frac{\partial}{\partial w}(\mathcal{L}_w^\dagger \mathbf{f}(\mathbf{X}_w)) + \sum_j \frac{\partial(\mathcal{L}_w^\dagger \mathbf{f})}{\partial X^{(j)}}f_j + \frac{1}{2}\sum_{j,k}\frac{\partial^2(\mathcal{L}_w^\dagger \mathbf{f})}{\partial X^{(j)}\partial X^{(k)}}[\boldsymbol{\sigma}\boldsymbol{\sigma}^\top]_{jk} \right) dw$$

$$+ \int_{t_0}^u \sum_j \frac{\partial(\mathcal{L}_w^\dagger \mathbf{f})}{\partial X^{(j)}}[\boldsymbol{\sigma} d\mathbf{W}_w]_j\, dw. \tag{71}$$

Applying Fubini's theorem in the original double integral, we change the order of integration

$$\int_{t_0}^t \int_{t_0}^s \phi(u)\, du\, ds = \int_{t_0}^t (t-u)\,\phi(u)\, du, \tag{72}$$

and we obtain

$$R_{1,a} = \int_{t_0}^t \int_{t_0}^s \mathcal{L}_u^\dagger \mathbf{f}(\mathbf{X}_u)\, du\, ds = \int_{t_0}^t (t-u)\left[ \underbrace{\sum_j \frac{\partial \mathcal{L}_u^\dagger \mathbf{f}}{\partial X^{(j)}}f_j}_{R_{1,a}^1} + \underbrace{\frac{1}{2}\sum_{j,k}\frac{\partial^2 \mathcal{L}_u^\dagger \mathbf{f}}{\partial X^{(j)}\partial X^{(k)}}[\boldsymbol{\sigma}\boldsymbol{\sigma}^\top]_{jk}}_{R_{1,a}^2} \right] du$$

$$+ \int_{t_0}^t (t-u)\underbrace{\sum_j \frac{\partial \mathcal{L}_u^\dagger \mathbf{f}}{\partial X^{(j)}}\,[\boldsymbol{\sigma}\, d\mathbf{W}_u]_j\, du}_{R_{1,a}^3} + \frac{\tau^2}{2}\mathcal{L}_t^\dagger \mathbf{f}(\mathbf{X}_{t_0}). \tag{73}$$

In the previous equation we have dropped the term $\frac{\partial}{\partial w}\left(\mathcal{L}_w^\dagger \mathbf{f}(\mathbf{X}_w)\right)$ that is equal to zero and that would require the drift $\mathbf{f}$ to be time-dependent to be non-negligible.

**First component $R_{1,a}^1$ of remainder term $R_{1,a}$: Flow curvature term.** The Itô/Backward Kolmogorov generator applied to a vector field $\mathbf{f}$ can be written as

$$\mathcal{L}^\dagger \mathbf{f} = \mathbf{J}_f \mathbf{f} + \frac{1}{2}\Delta_D \mathbf{f}. \tag{74}$$

In Eq. 74, $\mathbf{J}_f \doteq \nabla \mathbf{f}$ denotes the Jacobian of $\mathbf{f}$, $\mathbf{D} \doteq \boldsymbol{\sigma}\boldsymbol{\sigma}^\top$ the noise covariance, and $\Delta_\mathbf{D} \doteq \sum_{j,k}\mathbf{D}_{jk}\partial^2_{X^{(j)}X^{(k)}}$ is the noise-covariance weighted Laplacian operator. Thus each component of $\mathcal{L}^\dagger \mathbf{f}$ comprises the directional derivative of the drift $\mathbf{J}_f \mathbf{f}$ plus an anisotropic/noise-covariance weighted Laplacian of $\mathbf{f}$, which in component-wise form is expressed as

$$\left[\mathcal{L}^\dagger \mathbf{f}\right]_i = \sum_k \frac{\partial f_i}{\partial X^{(k)}}\, f_k + \frac{1}{2}\sum_{k,\ell}\mathbf{D}_{k\ell}\frac{\partial^2 f_i}{\partial X^{(k)}\partial X^{(\ell)}}. \tag{75}$$

Differentiating wrt to $X^{(j)}$ yields

$$\frac{\partial}{\partial X^{(j)}}\left[\mathcal{L}^\dagger \mathbf{f}\right]_i = \sum_k \frac{\partial^2 f_i}{\partial X^{(j)}\partial X^{(k)}}\, f_k + \sum_k \frac{\partial f_i}{\partial X^{(k)}}\frac{\partial f_k}{\partial X^{(j)}} + \frac{1}{2}\sum_{k,\ell}\mathbf{D}_{k\ell}\frac{\partial^3 f_i}{\partial X^{(j)}\partial X^{(k)}\partial X^{(\ell)}}, \tag{76}$$

and thus we rewrite the $i$-th component of the term $R_{1,a}^1$ as

$$\left[R_{1,a}^1\right]_i = \int_{t_0}^t (t-u)\left[\sum_{j,k}\frac{\partial^2 f_i}{\partial X^{(j)}\partial X^{(k)}}\,f_k\,f_j + \sum_{j,k}\frac{\partial f_i}{\partial X^{(k)}}\,\frac{\partial f_k}{\partial X^{(j)}}\,f_j + \frac{1}{2}\sum_{j,k,\ell}\mathbf{D}_{k\ell}\frac{\partial^3 f_i}{\partial X^{(j)}\partial X^{(k)}\partial X^{(\ell)}}\,f_j\right]_i \mathrm{d}u.$$

$$(77)$$

The third-order state-derivative in the last summand of Eq. 77, indicates that this term is inactive for linear or quadratic drift functions $\mathbf{f}$.

We re-write again this part of the remainder in a more compact vector notation in terms of the directional derivative of $(\mathbf{J}_f\mathbf{f})$ and $\frac{1}{2}\Delta_D\mathbf{f}$ along the vector field as

$$R_{1,a}^1 = \int_{t_0}^t (t-u)\Big[\underbrace{\nabla(\mathbf{J}_f\mathbf{f})\cdot\mathbf{f}}_{\text{flow curvature}} + \underbrace{\nabla(\tfrac{1}{2}\Delta_D\mathbf{f})\cdot\mathbf{f}}_{\text{diffusive term along the flow}}\Big]\,\mathrm{d}u. \qquad (78)$$

This part of the remainder captures two geometric effects that standard inference methods neglect: the **intrinsic curvature of deterministic flow trajectories in state space**, and the **systematic bias introduced by the spatial variation of both drift and diffusion** along these trajectories, when both drift and diffusion are assumed as constant between inter-observation intervals.

- To understand the **first term**, $\nabla(\mathbf{J}_f\mathbf{f})\cdot\mathbf{f}$, from a geometric perspective, let us consider a deterministic dynamical system with dynamics $\dot{\mathbf{x}}_t = \mathbf{f}(\mathbf{x}_t)$. A trajectory initiated from an initial condition $\mathbf{x}_0$ traces a streamline in the state space $\mathcal{R}^d$. We express the acceleration of this trajectory in terms of the directional derivative

$$\ddot{\mathbf{x}}_t = \frac{\mathrm{d}}{\mathrm{d}t}\mathbf{f}(\mathbf{x}_t) = \mathbf{J}_f(\mathbf{x}_t)\cdot\mathbf{f}(\mathbf{x}_t) = \mathbf{J}_f\cdot\mathbf{f}. \qquad (79)$$

  The acceleration vector admits a natural orthogonal decomposition comprising a component parallel to the vector field $\mathbf{f}$ and an orthogonal component to $\mathbf{f}$

$$\mathbf{J}_f\cdot\mathbf{f} = P_\parallel(\mathbf{f})\,\mathbf{J}_f\cdot\mathbf{f} + P_\perp(\mathbf{f})\,\mathbf{J}_f\cdot\mathbf{f}. \qquad (80)$$

  Here $P_\parallel(\mathbf{f}(\mathbf{x})) = \frac{\mathbf{f}(\mathbf{x})\mathbf{f}^\top(\mathbf{x})}{\|\mathbf{f}(\mathbf{x})\|^2}$ and $P_\perp(\mathbf{f}(\mathbf{x})) = \mathbb{I} - P_\parallel(\mathbf{f}(\mathbf{x}))$ stand for the parallel and orthogonal projectors. The parallel component quantifies the rate of change of speed along the trajectory, whilst the perpendicular component defines the **curvature vector** $\kappa_{\text{flow}}(x)$ (Kühnel, 2002), which quantifies the bending of the trajectories

$$\boldsymbol{\kappa}_{\text{flow}}(\mathbf{x}) = \frac{P_\perp(\mathbf{f}(\mathbf{x}))\mathbf{J}_f(\mathbf{x})\mathbf{f}(\mathbf{x})}{\|\mathbf{f}(\mathbf{x})\|^2}. \qquad (81)$$

  When $\boldsymbol{\kappa}_{\text{flow}} = 0$, the trajectories are straight lines in the state space, while when $\|\boldsymbol{\kappa}_{\text{flow}}\| > 0$ they are curved.

  The term $\nabla(\mathbf{J}_f\mathbf{f})\cdot\mathbf{f}$ quantifies the **evolution of the trajectory curvature** [2] as the system moves along the flow field. From Eq. 77 we have for each dimensional component $i$ of this term

$$[\nabla(\mathbf{J}_f\mathbf{f})\cdot\mathbf{f}]_i = \sum_{j,k}\frac{\partial^2 f_i}{\partial X^{(j)}\partial X^{(k)}}f_k f_j + \sum_{j,k}\frac{\partial f_i}{\partial X^{(k)}}\frac{\partial f_k}{\partial X^{(j)}}f_j$$
$$= [\mathbf{f}^\top(\nabla^2 f_i)\mathbf{f}] + [\mathbf{J}_f^2\mathbf{f}]_i. \qquad (82)$$

  We observe that this term captures the effects of how both second-order spatial variation of the flow field (the Hessian $\nabla^2 f_i$) and the Jacobian of the acceleration ($\mathbf{J}_f^2\mathbf{f}$) influence the evolution of trajectories.

---

[2]More precisely the directional derivative of the acceleration, $\mathbf{J}_f(\mathbf{x})\cdot\mathbf{f}$ along the flow direction, or the **rate at which the acceleration changes along the flow, or a measure of how the local curvature of $\mathbf{f}$ as a vector field influences trajectory evolution**.

- In Eq. 82, the **first sub-term**, $\mathbf{f}^\top(\nabla^2 f_i)\mathbf{f}$, represents the **second directional derivative** of $f_i$ along the flow direction. In regions where the Hessian $\nabla^2 \mathbf{f}$ is large (as is for the case of a highly nonlinear drift), this term becomes significant, and it vanishes for linear or constant drift $\mathbf{f}$.
- The **second sub-term**, $\mathbf{J}_f^2 \mathbf{f} = \mathbf{J}_f(\mathbf{J}_f \mathbf{f})$, of Eq. 82 represents the action of the Jacobian operator on the acceleration vector. Geometrically, it describes how the local linearised field acts on the acceleration as we move an infinitesimal step along the flow field.

By temporal integration we have

$$R_{1,a}^1 = \int_{t_0}^{t} (t-u)\nabla(\mathbf{J}_f \mathbf{f}) \cdot \mathbf{f} \, \mathrm{d}u \sim \frac{\tau^2}{2}\nabla(\mathbf{J}_f \mathbf{f}) \cdot \mathbf{f}, \tag{83}$$

indicating that the evolution of trajectory curvature introduces an $O(\tau^2)$ correction to the transition density.

Drift inference based on Euler–Maruyama–type discretisation ignores between others the term $R_{1,a}^1$ introducing thereby a mean bias at each point $\mathbf{x}$ in the state space,

$$\beta_{1,a}^1(\mathbf{x}) = \frac{1}{\tau} R_{1,a}^1 \approx \frac{\tau}{2}\left[\nabla(\mathbf{J}_f \mathbf{f}) \cdot \mathbf{f}\right](\mathbf{x}). \tag{84}$$

- The **second term** in Eq.78, $\nabla(\frac{1}{2}\Delta_D \mathbf{f}) \cdot \mathbf{f}$, accounts for the diffusion part of the backward generator acting on the vector field $\mathbf{f}$. The anisotropic Laplacian $\Delta_D \mathbf{f}$ quantifies the **diffusion–weighted second-order spatial variation of the vector field**

$$[\Delta_D \mathbf{f}]_i = \sum_{j,k} D_{jk}\frac{\partial^2 f_i}{\partial X^{(j)}\partial X^{(k)}} = \nabla \cdot (\mathbf{D}\,\nabla f_i). \tag{85}$$

The directional derivative quantifies how this term evolves along the flow field

$$\left[\nabla\left(\tfrac{1}{2}\Delta_D \mathbf{f}\right) \cdot \mathbf{f}\right]_i = \frac{1}{2}\sum_{j,k,\ell} D_{k\ell}\frac{\partial^3 f_i}{\partial X^{(j)}\partial X^{(k)}\partial X^{(\ell)}} f_j. \tag{86}$$

This term captures how the diffusion-weighted spatial variation of the flow field varies across the state space. As trajectories traverse regions of varying drift curvature, the effective Itô correction itself changes, introducing systematic bias in inference methods that assume that drift is piece-wise constant in-between observations.

**Second component $R_{1,a}^2$ of remainder term $R_{1,a}$.**

$$R_{1,a}^2 = \int_{t_0}^{t}(t-u)\frac{1}{2}\sum_{j,k}\frac{\partial^2\left(\mathcal{L}_u^\dagger \mathbf{f}\right)}{\partial X^{(j)}\partial X^{(k)}}\left[\boldsymbol{\sigma}\boldsymbol{\sigma}^\top\right]_{jk}\mathrm{d}u. \tag{87}$$

For the $i$-th dimensional component we have

$$\begin{aligned}
\frac{\partial^2}{\partial X^{(h)}\partial X^{(j)}}\left[\mathcal{L}_u^\dagger f\right]_i &= \sum_k \frac{\partial^3 f_i}{\partial X^{(h)}\partial X^{(j)}\partial X^{(k)}} f_k + \sum_k \frac{\partial^2 f_i}{\partial X^{(j)}\partial X^{(k)}}\frac{\partial f_k}{\partial X^{(h)}} \\
&\quad + \sum_k \frac{\partial^2 f_i}{\partial X^{(h)}\partial X^{(k)}}\frac{\partial f_k}{\partial X^{(j)}} + \sum_k \frac{\partial f_i}{\partial X^{(k)}}\frac{\partial^2 f_k}{\partial X^{(h)}\partial X^{(j)}} \\
&\quad + \frac{1}{2}\sum_{k,\ell}\mathbf{D}_{k\ell}\frac{\partial^4 f_i}{\partial X^{(h)}\partial X^{(j)}\partial X^{(k)}\partial X^{(\ell)}}.
\end{aligned} \tag{88}$$

For this remainder term, we have for each dimensional component $i$

$$\left[R_{1,a}^2\right]_i = \int_{t_0}^{t}(t-u)\frac{1}{2}\sum_{j,k}\mathbf{D}_{jk}\left[\frac{\partial^2}{\partial X^{(k)}\partial X^{(j)}}\left[\mathcal{L}_u^\dagger \mathbf{f}\right]_i\right]\mathrm{d}u \,. \tag{89}$$

Geometrically, $R_{1,a}^2$ captures the **diffusion-weighted second-order spatial variation** of the generator $\mathcal{L}_u^\dagger \mathbf{f}$ across the $\sqrt{\tau}$-sized ellipsoid set by $\mathbf{D}$, i.e. the anisotropic Laplacian $\Delta_D(\mathcal{L}_u^\dagger \mathbf{f})$, the diffusion-weighted second spatial variation of the drift along the flow. Dropping this term in inference amounts to assuming $\mathcal{L}_u^\dagger \mathbf{f}$ is locally flat and results in an $O(\tau)$ drift bias of size $\beta_{1,a}^2 \approx (\tau/4)\,\Delta_D(\mathcal{L}_u^\dagger \mathbf{f})$, underestimating anisotropy and the evolution of curvature of the flow field, so inferred flowlines appear too straight.

**Third component $R_{1,a}^3$ of remainder term $R_{1,a}$.**

$$R_{1,a}^3 = \int_{t_0}^t (t - u) \sum_j \frac{\partial \mathcal{L}_u^\dagger \mathbf{f}}{\partial X^{(j)}} \, [\sigma \, \mathrm{d}\mathbf{W}_u]_j \, \mathrm{d}u, \tag{90}$$

$$\left[R_{1,a}^3\right]_i = \int_{t_0}^t (t - u) \sum_{j,m} \frac{\partial}{\partial X^{(j)}} \left[\mathcal{L}_u^\dagger \mathbf{f}\right]_i \sigma_{jm} \, \mathrm{d}\mathbf{W}_u^{(m)} \mathrm{d}u, \tag{91}$$

This is a martingale term capturing the stochastic coupling between diffusion and the spatial inhomogeneity of the generator. In inference, this term doesn't introduce bias, since $\langle R_{1,a}^3 \rangle = 0$. However, neglecting this term, ignores a second–order variance contribution with $\mathrm{Var}(R_{1,a}^3/\tau) = O(\tau)$.

To understand better when these remainder terms contribute significantly to the discretisation, we study here closer how each term contributes to the remainder term $R_1, a$ for **(i)** a linear system under different parameter regimes that influence the curvature of the flow field, **(ii)** a nonlinear system.

G.4 ABLATIONS WITH RESPECT TO METRIC LEARNING ALGORITHM

To probe the robustness of our framework, when we employ a different approach to estimate the metric, following (Kapusniak et al., 2024) we tested our method when we employ an radial based function approximation to estimate the diagonal metric, similar to Arvanitidis et al. (2017). In the following table we report the performance of our method when we employ the locally adaptive normal distribution framework (LAND) for approximation the metric (as in the main text) Arvanitidis et al. (2019) and when we employ the radial basis function variant of the metric approximation (RBF) for the Van der Pol system for different inter-observation intervals and noise conditions.

## H   DETAILS ON NUMERICAL EXPERIMENTS

We simulated a two dimensional Van der Pol oscillator with drift function

$$f_1(x, y) = \mu(x - \frac{1}{3}x^3 - y) \tag{92}$$

$$f_2(x, y) = \frac{1}{\mu}x, \tag{93}$$

starting from initial condition $x0 = [1.81, -1.41]$ and under noise amplitudes $\sigma = \{0.25, 0.50, 0.75, 1.00\}$ for total duration of $T = \{500, 1000\}$ time units. The employed inter-observation intervals $\tau = \{80, 120, 160, 200, 240, 280, 320\} \times dt$. The last inter-observation interval exceeds the half period of the oscillator and thus samples only a single state per period. This resulted in erroneous estimates. In this setting this indicates the upper limit of $\tau$ for which we can provide estimates. However for any inference method, if the observation process samples only one observation per period, identifying the underlying force field without additional assumptions is not possible with temporal methods. The discretisation time-step used for simulation of the ground truth dynamics, and path augmentation $\delta t = 0.01$. For sampling the controlled bridges we employed $N = 100$ particles evolving the associated ordinary differential equation as described in (Maoutsa and Opper, 2022). The logarithmic gradient estimator used $M = 40$ inducing points. The sparse Gaussian process for estimating the drift was based on a sparse kernel approximation of $S = 300$ points. In the presented simulation we have employed a weighting parameter $\beta = 0.5$ (Eq. 37). This provides a moderate pull towards the invariant density. The example in Figure 2 was constructed with $\beta = 1$ and provides a better approximation of the transition density, than $\beta = 0.5$.

For the **out-of-equilibrium process** with harmonic trapping and circulation and a Gaussian repulsive obstacle in the centre we followed the description presented in Frishman and Ronceray (2020) following the drift

$$f_\mu(\mathbf{x}) = -\Omega_{\mu\nu}x_\nu + \alpha e^{-x^2/2\sigma^2}x_\mu \quad \text{with} \quad \Omega = \begin{pmatrix} 2 & 2 \\ -2 & 2 \end{pmatrix}, \tag{94}$$

for $\alpha = 10$ and simulated the stochastic system with noise amplitude $\sigma = 0.5$ on a time grid of $dt = 0.01$ steps, observed at inter-observation intervals $\tau = \{150, 200, 250\} \times dt$ and for total duration $T = 1000$ time units.

For the **Hopf system** we used the drift

$$f_1(x_1, x_2) = z_2, \tag{95}$$
$$f_2(x_1, x_2) = -z_1 + (\mu - z_1^2) z_2, \tag{96}$$

with $\mu = 0.35$ and integrated the system with noise amplitude $\sigma = 0.15$ on a timegrid with $dt = 0.01$ resolution, observed at $\tau = \{200, 300, 400\} \times dt$ time intervals. This is the normal form of the Hopf bifurcation.

For the **Selkov glycolysis model** (Selkov, 1968) we employed the drift

$$f_1(x_1, x_2) = -x_1 + \alpha x_2 + x_1^2 x_2, \tag{97}$$
$$f_2(x_1, x_2) = 0.6 - \alpha x_2 - x_1^2 x_2, \tag{98}$$

with $a = 0.06$ and noise amplitude $\sigma = 0.05$ for inter-observation intervals $\tau = \{100, 200\} \times dt$ and simulation time grid of $dt = 0.01$ spacing and for total duration $T = 1000$ time units.

This model is a minimal two-variable model of glycolytic oscillations, first introduced in (Selkov, 1968). It describes the autocatalytic feedback processes in the glycolysis pathway, focusing on how simple nonlinear interactions can give rise to oscillatory dynamics in concentrations of intermediates. The first state variable $x_1$ represents the concentration of adenosine diphosphate, while $x_2$ represents the concentration of a glycolytic intermediate.

### H.0.1   ON COMPUTATION OF GEODESIC CURVES

For the computation of geodesic curves we followed the framework introduced in (Arvanitidis et al., 2019). The geodesic equation relies on a non-parametric estimation of the Riemannian metric, which

is constructed using kernel-weighted local diagonal covariances, and has computational complexity $\mathcal{O}(ND)$, where $D$ is the dimensionality of the problem and $N$ denotes the number of samples. The computational cost of solving the geodesic equation scales sublinearly with increasing dimensionality.

## H.1 DETAILS ON BASELINE METHODS

We compared the performance of our method against a series of competing methods for inference of stochastic dynamics. In particular, we compared our method against methods specifically designed for inference of stochastic systems from single trajectories, and against systems that infer population dynamics.

We employed the following methods that assume single trajectories for drift inference:

1. Gaussian process regression without state estimation (**GP**)

2. path augmentation with Ornstein-Uhlenbeck dynamics with Gaussian process inference (**OU**) (Batz et al., 2018)

3. sparse variational inference with state estimation (**SVISE**) (Course and Nair, 2023a)

4. basis function approximation of Kramers-Moyal coefficients, i.e. the drift function (**KM-basis**) (Nabeel et al., 2025)

5. latent SDE inference with amortized reparameterization with (**LatentSDE+GP-pre**) and without pre-training (**LatentSDE**) (Course and Nair, 2023b).

We further compared our method with recent Schroedinger bridge generating frameworks that primary aim to infer population dynamics from snapshot data. In particular we considered the following frameworks:

I. Metric Flow Matching (**MFM**) (Kapusniak et al., 2024)

II. Generalized Schrödinger Bridge Matching (**GSBM**) (Liu et al., 2023)

III. Wasserstein Lagrangian Flows-Action Matching (**WLF-AM**) (Neklyudov et al., 2023b)

IV. Simulation-free Schrödinger bridges via score and flow matching (**[SF]$^2$ M**) (Tong et al., 2023)

For these methods, we clustered the observations of each system into *disjoint* subsets of adjacent points. We employed the k-Nearest neighbours algorithm (Fix, 1985; Cover and Hart, 1967) to construct the clusters as local neighbourhoods on the state space, comprising each at most 64 and minimum 20 observations. We paired each cluster $\mathcal{J}_b$ with the set $\mathcal{J}_b^+ \doteq \{\, \mathcal{O}_{k+1} : \mathcal{O}_k \in \mathcal{J}_b \,\}$ of the next observation of each cluster member. We then considered each cluster pair $(\mathcal{J}_b, \mathcal{J}_b^+)$ as the initial and terminal condition for a Schrödinger bridge problem, i.e.

$$\pi_0^b \doteq \{\, \mathcal{O}_k : \mathcal{O}_k \in \mathcal{J}_b \,\} \tag{99}$$

$$\pi_1^b \doteq \{\, \mathcal{O}_\ell : \mathcal{O}_\ell \in \mathcal{J}_b^+ \,\}. \tag{100}$$

These serve as samples of the densities required as boundaries conditions for the Schrödinger bridges.

For the multi-marginal setting, starting from the cluster that contained the observation $\mathcal{O}_1$ and subsequently created a sequence of cluster following the time ordering of the observations, i.e.

$$\pi_i^0 = \{\, \mathcal{O}_{k+i} : k \in \mathcal{J}_0 \,\}. \tag{101}$$

We then employed a sequence of 50 marginal densities $\{\pi_i^0\}_{i=0}^{49}$ as snapshot observations required by the framework.

**Metric Flow Matching.** For the Metric Flow Matching framework, we trained on observations resulting from total simulation length $T_{\text{MFM}} = 3\,T = 1500$ (time units) to ensure sufficient data for each bridge. For each constructed bridge indexed by $b$, the flow network trained with the flow matching objective represents the velocity of the samples $\mathbf{u}_b(\mathbf{x}, t)$ transferred within the normalised

time $t \in [0, 1]$ from the initial boundary condition to the terminal one. We approximate a time-independent local drift $\hat{\mathbf{f}}_b(\mathbf{x})$ by rescaling the velocity field $\mathbf{u}_b(\mathbf{x}, t)$ with the inter-observation interval $\tau$, i.e.,

$$\hat{\mathbf{f}}_b(\mathbf{x}) \;=\; \frac{1}{\tau}\,\mathbf{u}_b(\mathbf{x}, t). \tag{102}$$

To obtain a global drift estimate from the individual local estimates, we compute "responsibilities" or weights of each individual drift for each point $\mathbf{x}_m$ of a pre-defined two-dimensional evaluation grid that covers the state space region occupied by the observations. These weights indicate how relevant each bridge $b$ was for estimating the drift at each grid point $\mathbf{x}_m$. For each bridge, we compute support weights $\omega_b(\mathbf{x})$ on the grid employing kernel density estimation (KDE) over the bridge boundary condition samples. Then, for each grid point $\mathbf{x}_m$, we compute bridge responsibilities as

$$\rho_b(\mathbf{x}_m) \;=\; \frac{\omega_b(\mathbf{x}_m)}{\sum_{j=1}^{B} \omega_j(\mathbf{x}_m)}, \qquad \sum_{b=1}^{B} \rho_i(\mathbf{x}_m) = 1. \tag{103}$$

We estimate the global drift at each grid point by weighting the local estimated drifts with the corresponding bridge responsibility, i.e.,

$$\hat{\mathbf{f}}(\mathbf{x}_m) \;=\; \sum_{b=1}^{B} \rho_b(\mathbf{x}_m)\,\hat{\mathbf{f}}_b(\mathbf{x}_m). \tag{104}$$

## I ALGORITHMIC DETAILS

Here we provide the outline algorithm for each constituent component of our work. Algorithm A1 provides the main skeleton of the framework. For the geometric approximation and the construction of the geodesics we defer the readers to Arvanitidis et al. (2019). Algorithm A2 outlines the solution of the control problem that implements the path augmentation. This part is an adapted version of the main algorithm proposed by Maoutsa and Opper (2021). Finally, Algorithm A3 describes the solution of the Gaussian process inference given the path augmentations (bridges) created for each augmentation pair.

---

**Algorithm A1:** Skeleton of the proposed framework.

**Input:** $\mathcal{O} = \{(\mathbf{x}_k, t_k)\}_{k=1}^{K}$: observed states at timepoint $t_k$
**Output:** $\hat{\mathbf{f}}$: posterior estimate of the drift function
  $\boldsymbol{B}^{(j)}$: augmented paths of latent states (optional)

  `// initialise f̂ with a coarse drift estimate`
1 Initialise drift estimate $\hat{\mathbf{f}}^{(0)}$ according to Eq. 20
  `// Approximate Riemannian metric from observations (Eq. 34)`
2 $H_{dd} = \text{ApproximateMetric}\big(\{\mathcal{O}_{\boldsymbol{k}}\}_{k=1}^{K}\big)$
  `//`
  `// Construct geodesics between 𝒪ₖ and 𝒪ₖ₊₁ under the estimated metric`
  `as shortest paths`
3 $\boldsymbol{\Gamma}^{(\ell)} = \text{ConstructGeodesics}(\mathcal{O}_k, \mathcal{O}_{k+1}, H_{dd})$
  `// Γᵏ = {γₜ'ᵏ}ₖ₌₁ᴷ geodesic curves between selected observation pairs`
4 **for** *each iteration $j$* **do**
     `// augment paths along geodesics using particle flow`
5    $\boldsymbol{B}^{(j)} = \text{AugmentPaths}(\{\mathcal{O}\}_{k=1}^{K}, \boldsymbol{\Gamma}^{(j)}, \hat{\mathbf{f}}^{(j-1)})$
     `// uses the deterministic particle flow / bridge construction`
     `(Alg. A2) to sample augmented trajectories with f̂⁽ʲ⁻¹⁾`
     `// Gaussian process inference of the drift function`
6    $\hat{\mathbf{f}}^{(\ell)} = \text{GPDriftInference}(\{\mathcal{O}\}_{k=1}^{K}, \boldsymbol{B}^{(j)})$
     `// update GP posterior over f using original and augmented data`
7 **end**

---

---

**Algorithm A2:** Path augmentation algorithm employing Deterministic Particle Flow control

---

**Input:** $N, M$: scalars, number of particles and number of inducing points

$t_k, t_{k+1}, dt$: scalars, initial and final timepoints, and discretisation step

$\mathcal{O}_k, \mathcal{O}_{k+1}$: $1 \times d$, $1 \times d$ initial and target state

$\hat{\mathbf{f}}$: current drift estimate

$\sigma$: noise amplitude

$\boldsymbol{\gamma}_t$: geodesic curve in functional representation

**Output:** $F$: $d \times N \times (t_{k+1} - t_k)/dt$, sample representation of $q_t(x)$

---

1   $\ell = \frac{(t_{k+1}-t_k)}{dt}$          // number of timesteps

            // Forward filtering $\rho_t(x)$ (Eq. 40)

2   $\epsilon = 10^{-3}$

3   $\mathbf{Z}_{ti=0} = \mathcal{N}(\mathcal{O}_k, \epsilon \mathbf{I}_d)$          // initialise particles' positions

4   $\mathbf{Z}_{ti=1} = \mathbf{Z}_0 + dt \left( \hat{\mathbf{f}}(\mathbf{Z}_0, t_0) - \frac{1}{2}\sigma^2 \frac{\mathbf{Z}_0 - \mathcal{O}_k}{\epsilon} \right)$     // 1st step is with analytic score

5   For $ti = 2 : \ell$           // deterministic propagation

6        $\mathbf{Z}_{ti+1} = \mathbf{Z}_{ti} + dt \left( \hat{\mathbf{f}}(\mathbf{Z}_{ti}, t) - \frac{1}{2}\sigma^2 \nabla \log \rho(\mathbf{Z}_{ti}; \mathbf{Z}_{ti}) \right)$

7        $W = \exp\left( -U(\mathbf{Z}_{ti+1}, t)\, dt \right)$

8        $T^* = \text{EnsembleTransformParticleFilter}(\mathbf{Z}_{ti+1}, W)$

9        $\mathbf{Z}_{ti+1} = \mathbf{Z}_{ti+1} \cdot T^*$

          // Time-reversed propagation of flow $q_t(\mathbf{x})$

10   $\mathbf{B}_{ti=\ell} = \mathcal{N}(\mathcal{O}_{k+1}, \epsilon \mathbf{I}_d)$        // initialise particles' positions

          // 1st step is stochastic

11   $\mathbf{B}_{ti=\ell-1} = \mathbf{B}_\ell - dt \left( \hat{\mathbf{f}}(\mathbf{B}_\ell, t_1) + \frac{1}{2}\sigma^2 \nabla \log \rho(\mathbf{B}_\ell; \mathbf{Z}_\ell) - \frac{1}{2}\sigma^2 \frac{\mathbf{B}_\ell - \mathcal{O}_{k+1}}{\epsilon} \right)$

12   For $ti = \ell - 2 : 0$          // deterministic propagation

13        $\mathbf{B}_{ti-1} = \mathbf{B}_{ti} - dt \left( \hat{\mathbf{f}}(\mathbf{B}_{ti}, t) - \frac{1}{2}\sigma^2 \nabla \log \rho(\mathbf{B}_{ti}, \mathbf{Z}_{ti}) + \frac{1}{2}\sigma^2 \nabla \log q(\mathbf{B}_{ti}, \mathbf{B}_{ti}) \right)$

          // Compute control $u(x\,t)$ and controlled paths $\mathbf{F}_{0:T}$

14   For $ti = 1 : \ell$

15        $\mathbf{u}(\mathbf{x}, ti) = \sigma^2 \nabla \log q(\mathbf{x}; \mathbf{B}_{ti}) - \sigma^2 \nabla \log \rho(\mathbf{x}; \mathbf{Z}_{ti})$

16        $\mathbf{F}_{ti+1} = \mathbf{F}_{ti} + dt \left( \hat{\mathbf{f}}(\mathbf{F}_{ti}, t) + \mathbf{u}(\mathbf{F}_{ti}, ti) - \frac{1}{2}\sigma^2 \frac{\mathbf{F}_0 - \mathcal{O}_k}{\epsilon} \right)$

---

With the notation $\nabla \log q(\mathbf{x}; \mathbf{B}_{ti})$ we indicate the score function estimation in a functional form ($\mathbf{x}$) based on the density represented by the particles $\mathbf{B}_{ti}$, while $\nabla \log q(\mathbf{F}_{ti}; \mathbf{B}_{ti})$ indicates the same score function evaluated at locations $\mathbf{F}_{ti}$.

---

**Algorithm A3:** Gaussian process drift inference from an augmented path measure (part I)

---

**Input:** $\mathcal{Z} = \{\mathbf{z}_i\}_{i=1}^{S}$: inducing points for the sparse GP (Sp)

$\{\mathbf{X}_j(t_\ell)\}_{j=1,\ldots,N}^{\ell=1,\ldots,T'}$: particle positions from the path measure $Q$ (BALL2)

$\{\mathbf{g}(\mathbf{X}_j(t_\ell), t_\ell)\}$: effective drift evaluated along particles (gbALL2)

$k^{\mathbf{f}}$: kernel with lengthscales $\ell_1, \ell_2, \ell_3$ (shared across dimensions)

$g$: diffusion amplitude, $\sigma^2 = g^2$

$\Delta t$: time step of the particle simulation

$d$: state dimension, $N$: number of particles, $T'$: number of time steps

**Output:** Approximations $I_1^{(i)}, I_2^{(i)}$ of the integrals over $A(\mathbf{x})$ and $B(\mathbf{x})$

                               `// 0. shorthand and initialisation`

1   Set $S \leftarrow |\mathcal{Z}|$ (number of inducing points)

2   Initialise $I_1 \in \mathcal{R}^{S \times S \times d}$ and $I_2 \in \mathcal{R}^{S \times d}$ to zero

3   Initialise $\Lambda \in \mathcal{R}^{S \times S \times d}$ and $\mathbf{d} \in \mathcal{R}^{S \times d}$ to zero

                 `// 1. compute kernel matrices on the inducing points`

4   Construct the inducing–inducing kernel matrix

$$\mathcal{K}_S = k^{\mathbf{f}}(\mathcal{Z}, \mathcal{Z}) \in \mathcal{R}^{S \times S}$$

and compute a regularised inverse

$$\mathcal{K}_S^{-1} = \left(\mathcal{K}_S + \varepsilon I\right)^{-1}, \quad \varepsilon \approx 10^{-3}.$$

5   Define the kernel map to inducing points

$$k^{\mathbf{f}}(\mathcal{Z}, \mathbf{x}) = \left(k^{\mathbf{f}}(\mathbf{z}_i, \mathbf{x})\right)_{i=1}^{S} \in \mathcal{R}^{S}.$$

                 `// 2. sample-based approximation of $A(\mathbf{x})$ and $B(\mathbf{x})$`

6   **for** $i = 1, \ldots, d$ **do**

                                 `// loop over state dimensions`

7      **for** $\ell = 1, \ldots, T'$ **do**

                                        `// loop over time`

8          Let $\mathbf{X}(t_\ell) \in \mathcal{R}^{d \times N}$ be the particle positions at time $t_\ell$

9          For each particle position $\mathbf{X}_j(t_\ell)$, compute

$$\mathbf{k}_j = k^{\mathbf{f}}(\mathcal{Z}, \mathbf{X}_j(t_\ell)) \in \mathcal{R}^{S}.$$

         Stack them column-wise to obtain

$$K_\ell = \left[\mathbf{k}_1, \ldots, \mathbf{k}_N\right] \in \mathcal{R}^{S \times N}.$$

10         Let $g_i(\mathbf{X}_j(t_\ell), t_\ell)$ denote the $i$-th component of the effective drift at particle $j$ and time $t_\ell$

                   `// accumulate Monte Carlo estimates of the integrals`

11         Update

$$I_1^{(i)} \leftarrow I_1^{(i)} + K_\ell K_\ell^{\top}, \qquad I_2^{(i)} \leftarrow I_2^{(i)} + K_\ell \mathbf{g}_i(t_\ell),$$

         where $\mathbf{g}_i(t_\ell) = \left(g_i(\mathbf{X}_1(t_\ell), t_\ell), \ldots, g_i(\mathbf{X}_N(t_\ell), t_\ell)\right)^{\top}$.

12      **end**

                    `// normalise by time and number of particles`

13     

$$I_1^{(i)} \leftarrow \frac{\Delta t}{N} I_1^{(i)}, \qquad I_2^{(i)} \leftarrow \frac{\Delta t}{N} I_2^{(i)}.$$

14   **end**

---

In this algorithm Here $I_1^{(i)}$ approximates $\int k^{\mathbf{f}}(\mathcal{Z}, \mathbf{x}) A(\mathbf{x}) k^{\mathbf{f}}(\mathbf{x}, \mathcal{Z}) \, \mathrm{d}\mathbf{x}$, and $I_2^{(i)}$ approximates $\int k^{\mathbf{f}}(\mathcal{Z}, \mathbf{x}) B_i(\mathbf{x}) \, \mathrm{d}\mathbf{x}$.

---

**Algorithm A4:** Gaussian process drift inference from an augmented path measure (part II)

---

**Input:** Same inputs as Alg. 3
$I_1^{(i)}, I_2^{(i)}$: Monte Carlo approximations from Alg. 3
$\mathcal{K}_S, \mathcal{K}_S^{-1}$: inducing–inducing kernel matrix and its regularised inverse

**Output:** Component-wise drift estimators $\hat{f}_i(\mathbf{x})$, $i = 1, \ldots, d$
Expected negative log data likelihood $\mathcal{L}_{\text{path}}$ under $Q_f$

// 3. compute $\Lambda$ and $\mathbf{d}$ for each component

1 **for** $i = 1, \ldots, d$ **do**

// match Eq. equation 42 with sparse GP parametrisation

2
$$\Lambda^{(i)} \leftarrow \frac{1}{\sigma^2} \mathcal{K}_S^{-1} I_1^{(i)} \mathcal{K}_S^{-1}, \qquad \mathbf{d}^{(i)} \leftarrow \frac{1}{\sigma^2} \mathcal{K}_S^{-1} I_2^{(i)}.$$

3 **end**

4 This matches the definitions

$$\Lambda = \frac{1}{\sigma^2} \mathcal{K}_S^{-1} \left( \int k^{\mathbf{f}}(\mathcal{Z}, \mathbf{x}) A(\mathbf{x}) k^{\mathbf{f}}(\mathbf{x}, \mathcal{Z}) \mathrm{d}\mathbf{x} \right) \mathcal{K}_S^{-1}, \quad \mathbf{d} = \frac{1}{\sigma^2} \mathcal{K}_S^{-1} \left( \int k^{\mathbf{f}}(\mathcal{Z}, \mathbf{x}) B(\mathbf{x}) \mathrm{d}\mathbf{x} \right).$$

// 4. define the component-wise drift estimators

5 For each component $i = 1, \ldots, d$, define

$$\hat{f}_i(\mathbf{x}) = k^{\mathbf{f}}(\mathbf{x}, \mathcal{Z}) \left( I + \Lambda^{(i)} \mathcal{K}_S \right)^{-1} \mathbf{d}^{(i)},$$

so that the full drift estimate is

$$\hat{\mathbf{f}}_S(\mathbf{x}) = \left( \hat{f}_1(\mathbf{x}), \ldots, \hat{f}_d(\mathbf{x}) \right)^{\top}.$$

// 5. compute expected negative log data likelihood under $Q_f$

6 Initialise accumulators $S_{\|f\|} \leftarrow 0$, $S_{\nabla \cdot f} \leftarrow 0$, $S_{f \cdot g} \leftarrow 0$

7 **for** $\ell = 1, \ldots, T'$ **do**

8  For all particle positions $\mathbf{X}_j(t_\ell)$, evaluate $\hat{\mathbf{f}}_S(\mathbf{X}_j(t_\ell))$

9  Accumulate

$$S_{\|f\|} \leftarrow S_{\|f\|} + \sum_{j=1}^{N} \|\hat{\mathbf{f}}_S(\mathbf{X}_j(t_\ell))\|^2,$$

$$S_{f \cdot g} \leftarrow S_{f \cdot g} + \sum_{j=1}^{N} \hat{\mathbf{f}}_S(\mathbf{X}_j(t_\ell))^{\top} \mathbf{g}(\mathbf{X}_j(t_\ell), t_\ell),$$

and compute the trace of the Jacobian $\nabla \cdot \hat{\mathbf{f}}_S(\mathbf{X}_j(t_\ell))$ via automatic differentiation, accumulating it into $S_{\nabla \cdot f}$

10 **end**

11 Approximate the expected negative log data likelihood (up to constants) as

$$\mathcal{L}_{\text{path}} = \frac{\Delta t}{N} \left( \tfrac{1}{2} S_{\|f\|} + S_{\nabla \cdot f} + S_{f \cdot g} \right),$$

which corresponds to evaluating the quadratic form in Eq. equation 42 under the approximate posterior $Q_f$.

---

## J  IMPACT STATEMENT

The aim of this work is to advance the field of dynamical inference for stochastic systems. While we do not foresee any direct societal consequences directly impinging from our work, we recognize that stochastic systems could be applied in military contexts, financial engineering, or more recently in machine learning for data (such as image, audio, video) generation. Still, the proposed method does not propose interventions that might lead to unfavourable societal outcomes. Overdamped Langevin systems are widespread in areas such as physics, biology, neuroscience, and ecology. We anticipate that our contributions will thus help these disciplines by offering a tool to identify and further study relevant systems.

Our contribution emphasises the importance of incorporating concepts from the evolving field of geometric statistics into system identification methods for stochastic systems. Although geometric and topological properties of invariant densities have been extensively studied in the context of deterministic systems, comparable attention is lacking for their stochastic counterparts. Our work further highlights that in settings where the amount of augmented data exceeds the number of observations, data augmentation frameworks can enhance inference accuracy by incorporating domain knowledge or other relevant information, such as the geometry of the system's invariant density we consider here. Many algorithms used for data augmentation, including the expectation maximisation algorithm employed in our work (Romero et al., 2019), show only **local convergence**. As a result, when the initial estimate deviates significantly from the true value, naive data augmentation methods may converge to suboptimal solutions that fail to accurately identify the underlying system.

## K  LLMS USAGE STATEMENT

During the preparation of this manuscript, we used general-purpose large language models (e.g., the GPT family) for grammar and writing polishing, minor rephrasing and condensing parts of the text, for limited code assistance (such as handling error messages and for parallelising and speeding up parts of the code), and for getting feedback on the finished draft. We did not rely on LLMs to generate research ideas, methods, experimental designs, analyses, or conclusions. All technical content, experiments, and claims were designed, implemented, and verified by the authors, who take full responsibility for the paper. Moreover, we did not embed any executable instructions, hidden prompts, or other mechanisms intended to influence the peer-review process in the manuscript or its supplementary materials.

