# OpenReview forum: "From geometry to dynamics: Learning overdamped Langevin dynamics from sparse observations with geometric constraints"
_ICLR.cc/2026/Conference — Submitted to ICLR 2026_

### Official Review · Reviewer_F9vm · 2025-10-25

**Soundness:** 3
**Presentation:** 1
**Contribution:** 2
**Rating:** 4
**Confidence:** 3

**Summary:**

This paper proposes a geometry-aware EM framework for learning overdamped Langevin dynamics from sparse temporal observations. The authors propose a novel, geometry-aware inference framework that formulates the problem as one of stochastic control.

The method combines:

1. Approximating the Riemannian metric from observations and constructing geodesics.
2. Using these geodesics as soft constraints to guide a path augmentation (diffusion bridge sampling) step, which reconstructs a likely continuous-time trajectory.
3. This augmented path is then used within an Expectation-Maximisation framework to infer the system's drift function via Gaussian Process (GP) regression.

The approach is evaluated on four 2D synthetic dynamical systems.

**Strengths:**

1. **Incorporating geometry** - Using geometry to regularise temporal inference is a relevant idea, and the paper effectively leverages it for sparse SDE identification
2. **Soundness of formulation** - The overall stochastic control + GP inference framework is coherent and conceptually bridges geometric learning, Schrödinger bridges, and system identification
3. **Relevance** - Sparse-sampling identification of physical systems is a real-world problem and the proposed setup is well-motivated within that context

**Weaknesses:**

1. **Synthetic, low-dim scope** - All evaluations are 2D toy examples; no real-world or higher-dimensional experiments are presented, raising questions about practical significance
2. **Positioning** - The proposed geodesic-based augmentation step appears very close in spirit to Generalized Schrödinger Bridge Matching (GSBM) and especially Metric Flow Matching (MFM) [1], which already learn and use data-dependent metrics and geodesic interpolants (Eqs. 2–3 in the paper being reviewed). The core geometric contribution of this paper thus appears to overlapwith MFM yet this is not acknowledged.
3. **Lack of ablations** - No sensitivity analysis with respect to the learned metric quality. No comparison or ablation replacing the first step (geometry + bridging) with GSBM or MFM. This would isolate the true contribution of the proposed path augmentation scheme versus these other geodesic-based interpolators.
4. **Scalability and Significance** -  The method's scalability is a significant concern. The use of Gaussian Processes for drift inference, even in its sparse form (as mentioned in the Appendix), limits the practical application to very low-dimensional systems and thus potential impact.
5. **Presentation** - The paper contains several editing issues: ,issing implementation details on the metric, typo in Fig. 1D (“obsevrations”), text occluded around line 244, and inconsistent or missing bolding of best results in Table 1. They reduce readability and professionalism.

 [1] Kapusniak, Kacper et al. Metric Flow Matching for Smooth Interpolations on the Data Manifold, 2024

**Questions:**

1. **Positioning** - Could the authors explicitly position their method against GSBM and MFM, and possibly include ablations or replacements using these existing geometric bridge formulations for the first step?
2. **Real-world relevance** - What are the real systems that method could be applied for? Following in that case, could the authors provide empirical evidence?
3. **Scalability** - Given the complexity of the GP, do the authors foresee this being a limiting factor for real world applications? What would be the alternative here?

---

> ### Author Response · Authors · 2025-12-03
>
> We thank the reviewer for their insightful comments.
>
>
> ## Detailed responses
> In the following we address their suggestions in a point-by-point manner.
>
>
>
> ### **1. Relation to Generalisaed Schroedinger bridge matching and Metric Flow Matching and additional ablations of our work with different metric estimation method (RBF in table 1)**
> > Positioning - The proposed geodesic-based augmentation step appears very close in spirit to Generalized Schrödinger Bridge Matching (GSBM) and
> especially Metric Flow Matching (MFM) [1], which already learn and use data-dependent metrics and geodesic interpolants (Eqs. 2–3 in the paper being
> reviewed). The core geometric contribution of this paper thus appears to overlapwith MFM yet this is not acknowledged.
>
> > Positioning - Could the authors explicitly position their method against GSBM and MFM?
>
> We thank the reviewer for bringing up these important related works. Indeed we became aware of the Metric flow matching work after the submission of the manuscript. While our work has been developed earlier than this framework (or in parallel) the MFM method shares the same geometric considertions with our approach, i.e. that the interpolation between the initial and terminal conditions should respect the geometry of the data manifold. While this framework does not consider stochastic dynamics (infers a deterministic drift) and considers observation son the population level, we can compare our method to this work by considering the probability flow ODE drift in the comparison and clustering nearby observations in the state space to create population level observations.
> On the other hand GSBM considers a similar stochastic control formulation as we do, only with boundary conditions formulated in terms of distributions. This has implications on the computational demands on the solutions that requires several iterations of the forward and backward flow, while our framework requires for each bridge one forward and one backward sampling. Moreover, this work does propose any explicit geometric considerations or connects the bridges or the geometry/manifold of the data, however one can easily formulate the path constraint of the control cost to respect geometric constraints, as we did in our comparisons, where we set the path cost of this framework to be the identified geodesic from our framework. Yet, on our hand this framework was too computationally heavy, and thus in our tables we report only mean from fewer than 5 instances for each setting due to time constraints of the rebuttal period.
> We have added a related work  Section E in the Supplement where we discuss the connection of our method to MFM and GSBM.
>
> Regarding the ablations, we have repeated the inference with our method by employing the radial basis function framework from Arvanitidis et a. 2021 for metric estimation similar to the MFM framework, and as detailed in Table one the performance is comparable but slightly worse to the initial results we presented.
>
>
>
> ----
>
> ### **2. Additional baseline methods**
>  > Lack of ablations - No sensitivity analysis with respect to the learned metric quality. No comparison or ablation replacing the first step (geometry +
> bridging) with GSBM or MFM. This would isolate the true contribution of the proposed path augmentation scheme versus these other geodesic-based interpolators.
>
> We thank the reviewer for proposing these ablation experiments.
> In the revised version of our manuscript we have now included two series of experiments in response to the reviewers comments.
>  - quantification of the robustness of the metric estimation method against the hyper-parameter selection for the metric estimation
>  - the proposed ablation where the augmentation is performed via:
>     - **Metric flow matching** (Kapuśniak, NeurIPS 2024),
>     - **Generalized Schrodinger Bridge Matching** (Liu et al, ICLR 2024), and
>     - **Simulation-free Schroedinger bridges via score and flow matching** ([SF]^2M) (Tong et al., AISTATS 2024).

---

> ### Author Response · Authors · 2025-12-03
>
> ### 3
> > Scalability - Given the complexity of the GP, do the authors foresee this being a limiting factor for real world applications? What would be the alternative
> here.
>
>
>
> >  Real-world relevance - What are the real systems that method could be applied for? Following in that case, could the authors provide empirical evidence?
>
>
>
> We thank the reviewer for these comments.  We consider the potential impact of our paper two-fold. First is to propose an approach that reconciles the two prevailing approaches to solve the system identification problem for stochastic dynamical systems, and the second is to propose geometric considerations for inference of stochastic dynamical systems, when observed from single state trajectories.The systems we employed to demonstrate our approach are models representing real dynamical systems, thus we do not agree with the concern about significance.
>
> It is true that Gaussian processes are best fitted for low dimensional systems, but the inference part of our framework can be easily replaced by a neural network and thus scale to higher dimensions. Since the rebuttal time was not sufficient to demonstrate this in the revisions, e will demonstrate such an example upon acceptance of the paper.
>
>
> ----
>
> ### **4. Fixed editing issues.**
> > Presentation - The paper contains several editing issues: missing implementation details on the metric, typo in Fig. 1D (“obsevrations”), text occluded
> around line 244, and inconsistent or missing bolding of best results in Table 1. They reduce readability and professionalism.
>
> We thank very much the reviewer for these suggestions to improve the presentation of our paper. We have now:
> - updated Figure 1
> - fixed the obscured word in line 243
> - bolded the best results in Table 1.
>
> [1] Kapusniak, Kacper et al. Metric Flow Matching for Smooth Interpolations on the Data Manifold, 2024

---

### Official Review · Reviewer_817P · 2025-10-26

**Soundness:** 3
**Presentation:** 3
**Contribution:** 3
**Rating:** 6
**Confidence:** 3

**Summary:**

The authors proposed a new method that leverage both information from path invariant metric to better infer drift of stochastic differential equations (SDEs) with low sampling rate. The method resembles an EM algorithm that in step 1 it infers the possible path between observations and in step 2 the algorithm estimates drift.

**Strengths:**

The method leveraged multiple source of information in a way to some extend convincing.
The geometric constrains potentially provides a way to share information among paths if multiple are observed, as oppose to e.g., latent SDE that not quite share information among multiple paths.
The test results appear good in some cases.

**Weaknesses:**

- Since the algorithm is trying to use invariant metric as a source of information, which needs to be estimated from the samples on one path --- this requires the pooled sample to faithfully reproduce invariant metric. As the authors' results suggested in the out of equilibrium system the method can fail.
- The mathematical justification is a bit weak, but heuristics like this work has value.

**Questions:**

- Can the author clarify a bit more how they method compare to latent SDE with some geometric informed prior process? Or an alternative fitting procedure with 1) do latent SDE to fill in observations with an initial prior drift 2) use the filled path to fit drift and use that as the prior drift in latent SDE again?
- Can the author clarify where they think the gain is coming from? Is it from geometric constrain or fitting the invariant metric. E.g., suppose I am willing to assume drift is a gradient field then fitting the invariant metric would give us a lot of information about the drift itself but geometric constrain might not be?

---

> ### Author Response · Authors · 2025-11-29
>
> We thank the reviewer for the positive assessment of our work and for their very insightful feedback.
>
>
> ## Summary of revisions
> **Overall the main updates included in the updated version of the manuscript in response to their suggestions are:**
>  - **Additional theoretical results that justify the geometric approximation.**
>  - **Additional comparisons with LatentSDE framework.**
>
>
> ---
> ## Detailed responses
> In the following we address their comments and suggestions in a point-by-point manner.
>
>
>
>
>
> ### **1. Additional mathematical justification**
> > The mathematical justification is a bit weak, but heuristics like this work has value.
>
> We appreciate the reviewer’s comment. In the updated version of the manuscript we have extended the appendix section [*Inference based on Euler-Maruyama discretisation does not account for the curvature of the trajectories in the state space*], where we analyse the remainder term that becomes non-negligible as the inter-observation interval $\tau$ increases. Moreover, in response to reviewer **uqCb**’s suggestion, we added a supplementary section that motivates our Riemannian manifold assumption (Sec. ).
> We would welcome any specific pointers on further justifications the reviewer would find useful.
>
> ---
>
> ### **2. Clarification of contribution**
> > Can the authors clarify where they think the gain is coming from? Is it from geometric constrain or fitting the invariant metric. E.g., suppose I am willing to
> assume drift is a gradient field then fitting the invariant metric would give us a lot of information about the drift itself but geometric constrain might not?
>
> We consider the gain to be coming from the geometric constraint. For systems where the drift can be written as the gradient of a potential, density estimation ("fitting the invariant metric") suffices to approximate drift, independent of sampling rate, if the invariant density is sufficiently sampled. In the case of a fixed point system, consecutive observations will be located in nearby regions in the state space, as the state of the system will fluctuate around the deterministic fixed point value. Thus, if one would perform path augmentation, they would observe that the second observation of each inter-observation interval falls in a high probability region of the transition density, thus no guidance from the geometric constraint is required. In contrast, for the systems considered here, path augmentation with coarsely estimated drift (from regressing the conditional state increments) results in bridges where the second boundary point of the bridge/the end observation of the observation pair, falls in very low probability regions (see Fig. 1 D upper sub-plot). Thus the nudging towards the geodesic interpolant mediated/provided by the geometric characterisation nudges the augmentation towards the correct direction, and

---

> > ### Author Response · Authors · 2025-11-29
> >
> > ### **3. Clarification that extracting a geometric prior drift from the data to use in latent SDE is not straightforward and additional comparisons with introducing priors in Latent SDE**
> > > Can the authors clarify a bit more how the method compares to latent SDE with some geometric informed prior process? Or an alternative fitting
> > procedure with 1) do latent SDE to fill in observations with an initial prior drift 2) use the filled path to fit drift and use that as the prior drift in latent SDE again?
> >
> > We thank the reviewer for this question and interesting suggestions. \
> > Regarding the first part of the question about training the latent SDE method with some geometric informed prior, we find that an interesting idea, but we wouldn't know how to define a geometric informed prior drift that matches the observations. **This is precisely the reason why we developed our approach from a control perspective, where we guide the augmentation towards the invariant geometry. Otherwise, if we would be able to define a geometry-informed prior we would have employed said prior drift to perform the path augmentation without requiring the approximation of the metric and the computation of geodesics to guide the augmentation.**
> >
> > However, if the reviewer has a concrete idea on how to define a geometric informed prior drift, we are very willing to implement their proposal during the discussion process.
> >
> > Instead of a geometric prior, we repeated the estimation with latent SDE by initialising the drift network with the drift estimated with simple Gaussian process regression (as we do for the first step of our method). We performed the initilisation through supervised pre-training on samples from the Gaussian process on the observations. We report the results in Table 2, but briefly we comment here that we observed negligible differences compared to the random initlisation of the drift network.

---

### Official Review · Reviewer_fNt7 · 2025-11-03

**Soundness:** 2
**Presentation:** 1
**Contribution:** 3
**Rating:** 4
**Confidence:** 2

**Summary:**

The paper proposes methods to solve for the underlying time-homogenous drift $f(X)$ of a system driven by Langevin dynamics with diffusion $\sigma$ from observed samples.  The authors propose to first estimate a Riemannian metric from the observed samples and construct geodesics between observed time points.   These geodesic paths are used to augment a stochastic optimal control problem between adjacent timepoints with a state-cost on generated trajectories which penalizes distance from calculated geodesics.   Gaussian process inference is used to update estimates of the drift across iterations.

**Strengths:**

The paper frames the problem well, gives extensive references to temporal and geometric methods, and provides results across a range of settings.    Thus, the proposed geometric constraints are well motivated (although the exposition in App C regarding Onsager-Machlup is not referenced anywhere in the main text, cf. L417).   The paper uses simulation with a learned drift & geometric constraints to improve upon Ornstein-Uhlenbeck bridges in previous work Batz et. al 2018.

The authors show promising experimental result in four systems of interest.   The method outperforms several baselines, demonstrates some robustness to diffusion coefficient misspecification, and improves performance with high stochasticity and high inter-observation time.

**Weaknesses:**

The paper is lacking in detail to understand the proposed method.    For example,
- the optimal control cost is relegated to Eq. 37 on pg. 24 of the Appendix.
- the interacting particle system from Maoutsa and Opper 2021a for solving the control problem is not explained, as far as I can tell
- it is not clear how to accurately estimate the $q_t(x)$ appearing in the solution in Eq. 7 (unless this is intended as the equation below Eq. 42), or how this was derived.
- "We employ a sample-based approximation of the densities in Eq. 38 ($Q_f$) resulting from the particle sampling of the path measure $Q$" (L1328, pg. 25) requires more detailed explanation.
- to what extent are the boundary constraints enforced by the solutions to the stochastic control problem in step $\beta$?
- the EM algorithm is deferred to Eq. 19-20 (pg 20-21), although the procedure is basic and components are suggested by the main text.   Nevertheless, I feel this simple statement would help ground the explanation of the method.
    - presumably, $\hat{f}$ is obtained using the updated drift estimate in step $\gamma$, but this fact and the procedure for obtaining $\hat{f}$ from Eq 7 is not clear in the main text.

I would greatly appreciate an algorithm box specifying steps of the algorithm and links to Appendix sections and/or Eq. numbers explaining details.    I was able to find a workshop version of this paper online, and while details were still lacking, I appreciated the probabilistic statements and context throughout that version of the work.



 The authors might consider citing and comparing with Kapuśniak et. al (NeurIPS 2024) "Metric flow matching for smooth interpolations on the data manifold".   Generalized Schrodinger Bridge Matching (Liu et. al 2023) and Wasserstein Lagrangian Flow (Neklyudov et. al 2023) would also be relevant baselines (where couplings could be given by same-trajectory samples and the state-cost is using the proposed distance-to-geodesic).

Minor comments;

- Line 244 is cut off by the figure.
- It is unclear to which experimental setting Fig 3 and L356-372 refer
- please be clear about the bold- and non-bold notation for $\mathcal{O}$ (e.g. Eq 2).   Presumably, we want to construct geodesics between non-bold $\mathcal{O}_k$ (i.e. same-trajectory samples from the bolded set)?

**Questions:**

My immediate questions are mostly regarding details of the method above, where lack of clarity in exposition is a primary weakness of the current submission.

---

> ### Author Response · Authors · 2025-11-28
>
> **We thank the reviewer for the overall positive and careful assessment of our work and for their very insightful suggestions for improvement.**
>
>
> ## Summary of revisions
>
> **Briefly we have made the following updates in response to reviewers fNt7 suggestions:**
> - **More methodological detail in the main text.** We have updated the methodological description of our paper to include more details originally deferred in the supplement into the main text.
>  - **Interacting particle control framework.** We updated the supplement with a more detailed description of the particle-based control employed for path augmentation.
>  - **Clarification on treatment of Dirac-delta and particle densities.** We have updated the supplement to include additional details regarding how we deal with the initial conditions of the exact observations that correspond to Dirac delta densities, and details on the particle representation of the densities, and requested additional details regarding the experiment in Fig 3 and L356-372.
>  - **Added a section where we provide the algorithms for the constituents of our framework.**
>  - **Included comparisons with additional baseline methods.** We have additionally compared our framework to the following existing frameworks:
>     - **Metric flow matching** (Kapuśniak, NeurIPS 2024),
>     - **Generalized Schrodinger Bridge Matching** (Liu et al, ICLR 2024), and
>     - **Simulation-free Schroedinger bridges via score and flow matching** ([SF]^2M) (Tong et al., AISTATS 2024).
>
>
>
>
>
>
> **Overall we have revised the manuscript to improve clarity and presentation. As the initial evaluation assigned a low presentation score,
> we would be grateful for any further guidance on aspects that remain unclear.**
>
> **The reviewer has commented that**
> > My immediate questions are mostly regarding details of the method above, where lack of clarity in exposition is a primary weakness of the current submission.
>
>
> **therefore we hope that the updated manuscript addresses their concerns. We hope the additions to the main text and the supplementary information address the reviewer’s concerns and warrant an updated assessment of both the presentation score and the overall rating.**
>
> ---
>
> ---
>
> ## Detailed responses
>
> The reviewer finds that our paper
> > frames the problem well, gives extensive references to temporal and geometric methods, and provides results across a range of settings. Thus, the proposed geometric constraints are well motivated.
>
> ---
> Below we address Reviewer fNt7’s comments point-by-point:
>
> ### **1. Details on the optimal control problem added in the main text**
> > The paper is lacking in detail to understand the proposed method. For example,
> > the optimal control cost is relegated to Eq. 37 on pg. 24 of the Appendix.
>
> We appreciate the reviewer's suggestion to include more methodological details in the main text, as opposed to deferring most details to the supplement. We agree that the main text should contain the essential methodological details. In the revised version of the manuscript, we have now included the definition of the optimal-control objective into the main text and provided a more detailed description of the control problem we solve for each bridge. However, due to space constraints, for a detailed understanding of our approach, one would still have to refer to the supplement.

---

> ### Author Response · Authors · 2025-11-28
>
> ### **2. Added detailed description of the particle system implementation in the Supplement**
> > the interacting particle system from Maoutsa and Opper 2021 for solving the control problem is not explained, as far as I can tell
>
> We thank the reviewer for raising this issue. We have now updated the supplement to contain a description of the implementation of the particle system used for computing the solution of the optimal control problem required for the path augmentation. Since we used a modified version of the implementation of Maoutsa and Opper 2021 for solving the control problem required for each augmented path, we only cited the paper that introduced the framework. However, we understand the necessity of including such details in our manuscript and for that reason we provide now an explanation of the particle system used (Supplement Sec. A.3 - Eq.40-43 and text below), and additionally have added an algorithm box that outlines both this part and the rest of our framework.
>
> Briefly the implementation involves propagating a particle density forward according to filtering Equation (42) starting from initial condition $\mathcal{\boldsymbol{O}}_k$. We implement the particle propagation of the filtering equation as a two-staged deterministic process: at every time step, we propagate the particles following the probability flow ODE with drift $\hat{\mathbf{f}}$ (the current drift estimate) and diffusion constant/matrix $\boldsymbol{\sigma}$ followed by a deterministic rewighting using the ensemble transform particle filter (a deterministic OT reweighting) (Reich, _SIAM Journal on Scientific Computing_, 2013).
> Consequently we perform a time-reversed propagation of a particle representation of the density starting at from initial condition using the score estimation of the filtering density following Eq.(40), and subsequently compute the time-dependent control (additional drift) as the difference of the score functions of the two particle densities. We employ the controlled drift for the augmentation.
>
> ---
>
> ### **3. Updated main text to refer to the supplement for description on how to construct the path augmentation**
> >it is not clear how to accurately estimate the $q_t(x)$
> appearing in the solution in Eq. 7 (unless this is intended as the equation below Eq. 42), or how this was
> derived.
>
>  > "We employ a sample-based approximation of the densities in Eq. 38 ($Q_f$ ) resulting from the particle sampling of the path measure $Q$  (L1328, pg. 25) requires more detailed explanation.
>
> We thank the reviewer for raising this point. As detailed in the Supplement, $q_t(x)$ is precisely the controlled particle density we construct for the path augmentation (_the one we described in the previous question_). Thus the integral below Eq. 7 in the main text, refers to a Monte Carlo integration based on the particle representation of the augmented density. We have now updated the main text to clarify this point. This is precisely the path measure $Q$  mentioned in L1328, pg. 25. We have also updated this part of the supplement.
>
>
> ---
> ### **4. Discussed the treatment of the Dirac delta distribution/boundary condition with a smooth mollifier in the Supplement**
>
> > to what extent are the boundary constraints enforced by the solutions to the stochastic control problem in step $\beta$ ?
>
>
>
> We appreciate bringing up this detail. In our formulation, the boundary constraints are imposed exactly at the continuous level (marginal constraints at the observation times). In practice, because observations are _exact_ (Dirac delta distributions), they pose an issue for the deterministic particle propagation (particle flow ODE) we employ, since the logarithmic gradient (score function) is undefined for that case.
>
>  There are several options to mitigate this issue: \
> (i) to consider the Dirac distribution as a very narrow Gaussian $\mathcal{N}(\mathcal{\boldsymbol{O}}_k, \epsilon \mathbf{I})$ and employ the analytic formulation of the score function of a Gaussian, i.e. $\nabla_x \text{log} p(\mathbf{x}) = - \frac{\mathbf{x}- \mathcal{\boldsymbol{O}}_k  }{\epsilon}$ and start from a very narrow Gaussian of variance $\epsilon \mathbf{I})$, i.e. to use a smooth mollifier,\
>  (ii) to propagate the particles for 1-2 steps with a stochastic update equation.
>
>  In our implementation we employed the first option, but in preliminary work we have explored the other alternative and observed that they do not confer any quantifiable difference in the resulting controlled paths. We have added these details to the supplement.

---

> ### Author Response · Authors · 2025-11-28
>
> ### **5. Added an algorithm outlining the drift estimation described in Eq.38-42**
> > the EM algorithm is deferred to Eq. 19-20 (pg 20-21), although the procedure is basic and components are suggested by the main text. Nevertheless, I feel this simple statement would help ground the explanation of the method.
> ◦ presumably, $\hat{f}$ is obtained using the updated drift estimate in step $\gamma$, but this fact and the procedure for obtaining $\hat{f}$ from Eq 7 is not clear in the main text.
>
> We thank the reviewer for this comment. At the beginning of our algorithm we employ a coarse estimation for the drift based on Gaussian process regression for the state increments, i.e. approximating this conditional expectation $\hat{\mathbf{f}}(\mathbf{x}) = \langle \frac{\text{d} \mathbf{X}_t}{\tau} | \mathbf{X}_t=\mathbf{x} \rangle$. At the subsequent steps $\hat{\mathbf{f}}(\mathbf{x}) $ is the estimate of the drift obtained during the previous iteration.
>
> The estimation of the drift from Eq. 7 is detailed in Eq. 38-42 in the Supplement due to space constraints for adding more details in the main text. We have now added a pointer from the main text to the corresponding text in the supplement and updated the supplement to include additional details as well as an algorithm box describing the implementation.
>
> ---
>
> ### **6. Added an algorithm for all the constituents of our framework in the Supplement**
> > I would greatly appreciate an algorithm box specifying steps of the algorithm and links to Appendix sections and/or Eq. numbers explaining details. I was able
> to find a workshop version of this paper online, and while details were still lacking, I appreciated the probabilistic statements and context throughout that version of the work.
>
> In the updated version of the manuscript we have provided the algorithm for all components of our framework. Upon acceptance, we will additionally release a repository with the python implementation of this work.
>
> Regarding the probabilistic details we have now included more details from the probabilistic formulation of the problem in the added **Setup and Background** section (Section 2). We thank the reviewer for this suggestion.
>
> ---

---

> ### Author Response · Authors · 2025-11-28
>
> ---
> ### **7. Constructed setup to facilitate comparison with Schroedinger bridge sampling frameworks and added comparisons with Metric Flow Matching, Generalized Schrodinger Bridge Matching, and Simulation free score and flow matching**
> > The authors might consider citing and comparing with Kapuśniak et. al (NeurIPS 2024) "Metric flow matching for smooth interpolations on the data manifold".
> Generalized Schrodinger Bridge Matching (Liu et. al 2023) and Wasserstein Lagrangian Flow (Neklyudov et. al 2023) would also be relevant baselines (where couplings could be given by same-trajectory samples and the state-cost is using the proposed distance-to-geodesic).
>
> We thank the reviewer for bringing up these frameworks for comparison. In the updated version of our manuscript we now include in Table 2 the results from experiments performed with these frameworks.
> - **Metric flow matching**  (MFM) (employing both the RBF and LAND metrics) (Kapuśniak, NeurIPS 2024)
> - **Generalized Schrodinger Bridge Matching** (GSBM) (Liu et al, ICLR 2024)
> - **Simulation-free Schroedinger bridges via score and flow matching**([SF]^2M) (Tong et al., AISTATS 2024),
>
>
> Regarding the **Wasserstein Lagrangian Flow** WLF (Neklyudov, ICML 2024) framework. While we have an implementation that worked by the end of the rebuttal period. We did not have enough time and resources to complete the experiments. However upon acceptance of the manuscript, the comparison with this method will be added too. Nevertheless, since this framework is by construction able to identify drifts that are gradients of a potential (conservative) we do not expect its performance to exceed ours or the other competitors.
>
> Briefly, these methods are slightly conceptually differently formulated compared to our method.
> These methods have been introduced to solve the Schroedinger bridge sampling problem, that is identifying the necessary **stochastic dynamics to transport an initial distribution to a final distribution**, assuming access to snapshot data from the same underlying process at multiple time points. By contrast, **our setting considers a single trajectory** (one member of the population) observed at discrete times.
>
>
> However, since we have the invariant density, to enable a comparison, we created the following setup: we form local “snapshots” by grouping states near each observation and pairing them with their subsequent observations, thereby creating two-time marginal constraints suitable for bridge estimation. We then partition the state space into clusters, construct bridges that transport each cluster at time $t$ to its successor at $t+\Delta t$, and estimate a local drift from each bridge according to these frameworks.
>
> To obtain a global drift estimate and compare it with our estimated drift, we compute weights or "responsibilites" for each bridge for each state space region that indicate how relevant each bridge is for approximating the drift at each state space location.
>
> We employ this protocol to compare the drift obtained by these Schroedigner bridge sampling baselines to the drift inferred by our method.
> Overall, we observed that since these approached involve training neural networks, they require much lengthier trajectories to deliver meaningful results, since with the evaluation procedure we described above, we require to have sufficient points at each "neighborhood" in the state space to train the bridge sampler.
>
> We have added implementation details (clustering details, responsibility computation etc.) to the supplement, while we will also release the code we wrote to construct these comparisons.
>
> ---
>
> ### **8. Fixed rendering issue**
>
> > Line 244 is cut off by the figure.
>
> Thank you for noticing this. Indeed the word "them" was occluded by the figure. It is now fixed in the updated version of the manuscript.
>
> ---
>
> ### **9. Clarified experimental setup**
> > It is unclear to which experimental setting Fig 3 and L356-372 refer.
>
> We thank the reviewer for mentioning this. Figure 3 refers to drift inference for a Van-der-Pol oscillator with our framework and with a framework that employs Ornstein-Uhlenbeck bridges (a linear SDE conditioned on the two consecutive observations) for path augmentation proposed in (Batz et al., Physical Review E 2018) for different noise levels and inter-observation intervals. While for small intervals the Ornstein-Uhlenbeck augmentations provide a good approximation of the latent path distribution, as the distance between consecutive observations increases the performance of this framework deteriorates as highlighted in this figure, while the geometric augmentation provides better estimated due to conditioning on the invariant density.

---

> > ### Author Response · Authors · 2025-11-28
> >
> > ### **10. Clarification of notation**
> > > please be clear about the bold- and non-bold notation for $\mathcal{O}$. Presumably we want to construct geodesics between non-bold $\mathcal{O}_k$, i.e. some trajectory samples from the bolded set?
> >
> > Thank you for raising this point. In our manuscript we use $\mathcal{\boldsymbol{O}}_k$ to denote a single $d$-dimensional observation, i.e. the state of the system at some time $t_k$, $\mathcal{{O}}_k^{(d)}$ to
> > denote the $d$-th dimensional component of $\mathcal{\boldsymbol{O}}_k$, and {$\boldsymbol{\mathcal{O}}_k$}$^K_k$    to indicate the set of $K$ observations. While the choice of the calligraphic letter is often used to refer to sets, here we slightly deviate from this notation only for this symbol, to prevent using $\mathbf{O}_k$, which resembles a zero.
> >
> > Thus the geodesics are constructed between boldface $\mathcal{\boldsymbol{O}}_k$, which are $d$-dimensional states.
> >
> > If the reviewers believe that this notation choice is misleading, we are open for suggestions to update the current notation with a non-calligraphic one.

---

### Official Review · Reviewer_uqCb · 2025-11-03

**Soundness:** 2
**Presentation:** 2
**Contribution:** 2
**Rating:** 4
**Confidence:** 3

**Summary:**

The paper introduces a geometry-aware path augmentation framework for learning overdamped Langevin dynamics from sparse snapshots. It first learns a Riemannian metric from the data invariant density, computes geodesics between observations, and then samples diffusion bridges constrained to remain near those geodesics. The authors apply proposed method to model stochastic systems, providing results across synthetic and real-world examples.

**Strengths:**

* **Motivation**: The authors provide a comprehensive overview motivating modeling stochastic processes in the abstract and introduction, makes it easy for reader who is new to the field to understand the theoretical background behind problem formulation
* **Geometric-aware method**: Novel geometric-aware method that is combined with diffusion bridges and control cost
* **Modeling non-conservative dynamics**: Authors specifically target system that exhibit non-conservative forces and show results on a range of stochastic settings

**Weaknesses:**

* **Problem formulation**: The introduction should clearly state problem formulation, summarizing limitations of prior approaches to learning stochastic dynamics (e.g., [1], [2], [3]), geometry-guided methods (e.g., [4]), and then state how the proposed method differs and  in which cases it outperforms these baselines. For example, this would add more value than a description below Figure 1 (which could be included as main text in shorter form and extended in appendix)
* **Existing work**: The work compares results to a limited set of baselines. I would suggest authors to include performance comparisons between proposed methods and baselines that operate by learning metric and/or inferring dynamics from sparse observations. Namely [4] learns Riemannian metrics to construct neural interpolates between end-points, [2] includes control cost and [1] and [3] operate in the stochastic setting. I understand that authors discuss some of these in related work and appendix (namely [2]), however it would be beneficial to include numerical experiments showing differences in empirical performance in the main text. Further it would be good to compare to computational cost across baselines.
* **Methodology and background**: I would suggest expanding theoretical background and methodology in lines 227-290, to improve clarity behind proposed method. Further it would be useful to provide algorithms behind each of the components.

**Questions:**

* Line 134 states that the problem reduces to low-dimensional manifold in case of invariant density. Could you provide some further theoretical clarity behind this?
* How do you construct the bridges in multi marginal setting?
* How do you learn drift control term and drift estimate given the Riemannian set-up?
* Minor comment, but I believe sentence in line 243 is cut off by a figure?

**References **

[1] Tong, Alexander, et al. "Simulation-free schr\" odinger bridges via score and flow matching." arXiv preprint arXiv:2307.03672 (2023).

[2] Liu, Guan-Horng, et al. "Generalized Schr\" odinger Bridge Matching." arXiv preprint arXiv:2310.02233 (2023).

[3] Shen, Yunyi, Renato Berlinghieri, and Tamara Broderick. "Multi-marginal schr\" odinger bridges with iterative reference refinement." arXiv preprint arXiv:2408.06277 (2024).

[4] Kapusniak, Kacper, et al. "Metric flow matching for smooth interpolations on the data manifold." Advances in Neural Information Processing Systems 37 (2024): 135011-135042.

---

> ### Author Response · Authors · 2025-11-29
>
> **We thank the reviewer for the positive assessment of our work and for their very insightful feedback.**
>
> ## Summary of revisions
> **Overall the main updates included in the updated version of the manuscript in response to their suggestions are:**
>  - **Revised introduction.** We transferred part of the background section included in the supplement of the original submission to the main text to clearly introduce the problem we tackle.
>  - **Additional baselines.** We have additionally extended the comparison of our method against the following frameworks that consider as observations snapshots of population dynamics
>       - **Simulation-free Schroedinger bridges via score and flow matching** ([SF]^2M) (Tong et al., AISTATS 2024),
>      - **Generalized Schroedinger Bridge Matching** (GSBM) (Liu et al., ICLR 2024),
>      - **Metric flow matching** (MFM)  using both RBF and LAND metrics (Kapuśniak, NeurIPS 2024).\
> We have additionally created a setup to facilitate a fair comparison between our method and these approaches that are based on Optimal Transport/Schroedinger bridge sampling.
>  - **Updated methodology to include more details in the main text.**
>  - **Will extend the supplementary information to include Section.** We will extend until the camera ready version the supplementary information to include the _Section F: Justification for Riemannian manifold approximation of the invariant density_. The main insights are in the detailed answer below.

---

> ### Author Response · Authors · 2025-12-03
>
> ## Detailed responses
> In the following we address their suggestions mentioned in the weakness section of their review and answer their questions in a point-by-point manner.
>
> ### **1. Revised main text to clarify problem formulation and background**
> > Problem formulation: The introduction should clearly state problem formulation, summarizing limitations of prior approaches to learning stochastic dynamics (e.g., [1], [2], [3]),
> >  geometry-guided methods (e.g., [4]), and then state how the proposed method differs and in which cases it outperforms these baselines.
> >  For example, this would add more value than a description below Figure 1 (which could be included as main text in shorter form and extended in appendix)
>
> We thank the reviewer for their helpful suggestion. In the originally submitted manuscript due to space constraints we had included the discussion and connection with prior work on Schroedinger bridge sampling in the supplement.
> However we agree with the suggestion that the problem formulation should be more clearly expressed in the main text. Thus we have now revised our main text to include an additional section before the Methodology  section, now Section 2.Setup and Background.
>
> Regarding the mentioning of the frameworks devised for Schroedinger bridge sampling in the introduction, we have to acknowledge that we suspect there is a small misunderstanding by the reviewer regarding  the setting of our framework and it compares to the setting considered with the mentioned papers. In our work, we consider that we observe a single trajectory (Stochastic path) over time and collect observations at discrete time intervals. In the mentioned papers, the setting assumes as observations the states of a **population of independent stochastic systems** evolving in parallel according to the same underlying dynamics. Thus at every observation point, in our setting we observe a single state, while in those frameworks the observations are **distributions of states**. In our work we assume a sequence of observations from a single path of the stochastic process, while these papers assume that they observe a population of stochastic processes at discrete time points, i.e. multiple paths of the process with initial conditions drawn from the initial distribution/configuration of states among the population. While the methods mentioned by the reviewer essentially estimate the system through the evolution of the probability density of the population, our setting identifies system dynamics through the evolution of a single trajectory. **Thus our approach employs more limited information at each time point (for longer observation time) compared to the aforementioned methods that consider an ensemble of paths/trajectories evolving in parallel.**
>
> However since both out work and these frameworks infer either explicitly or implicitly stochastic dynamics, in the updated version of our manuscript **we have now developed a setup to compare these frameworks to our work**, by clustering nearby states of the invariant density into "populations" learning the underlying **local** stochastic dynamics by constructing the Schroedinger bridge to match the distribution of states of each cluster with their succeeding one in time.
>
> However, given the differences in the setting considered by our framework and those additional Schroedinger based methods, we found that mentioning them as related work in the opening section of our paper was confusing or misleading for the reader. To that end, while we have added the comparison with the frameworks mentioned above as baselines, we have deferred the discussion on how their approach relates to our in the supplement in the Related work section where

---

> > ### Author Response · Authors · 2025-12-03
> >
> > ### **2 Included additional baselines with Schroedinger bridge matching frameworks**
> > > Existing work: The work compares results to a limited set of baselines. I would suggest authors to include performance comparisons between proposed
> > methods and baselines that operate by learning metric and/or inferring dynamics from sparse observations. Namely [4] learns Riemannian metrics to
> > construct neural interpolates between end-points, [2] includes control cost and [1] and [3] operate in the stochastic setting. I understand that authors
> > discuss some of these in related work and appendix (namely [2]), however it would be beneficial to include numerical experiments showing differences in
> > empirical performance in the main text. Further it would be good to compare to computational cost across baselines.
> >
> >
> > We have now extended the comparison of our work to additionally consider the following frameworks:
> >
> >    - **Metric flow matching** (MFM) (Kapuśniak, NeurIPS 2024) (with both the LAND and the RBF approached for estimating metrics))
> >    -  **Generalized Schrodinger Bridge Matching** (GSBM) (Liu et al, ICLR 2024),
> >    - **Simulation-free Schroedinger bridges via score and flow matching** ([SF]^2M) (Tong et al., AISTATS 2024).
> >
> > ---
> >
> >
> >
> > ### **3. Improved clarity and details on methodology in main text and added section with Algorithm boxes**
> > > Methodology and background: I would suggest expanding theoretical background and methodology in lines 227-290, to improve clarity behind proposed method. Further it would be useful to provide algorithms behind each of the components.
> >
> > We have now updated the methological exposition in our main text and have additionally added Algorithm boxes that detail the algorithmic implementations of all constituents of our work in Section I in the supplement.

---

> > > ### Author Response · Authors · 2025-12-03
> > >
> > > ### **4. Justification of assuming a an insuced low-dimensional structure**
> > > > Line 134 states that the problem reduces to low-dimensional manifold in case of invariant density. Could you provide some further theoretical clarity behind this?
> > >
> > >
> > > We thank the reviewer for pointing out that this statement in our text requires further explanation.
> > > In our manuscript we precisely mention that:\
> > > "The invariant density of the observed system imposes a low-dimensional structure on the state space, within which the observations are confined. We propose that this low-dimensional structure is well approximated by a Riemannian manifold $\mathcal{M}_{\infty} \in \mathcal{R}^{m \leq d}$ in the ambient space, and that the ensemble of observations $\{\boldsymbol{\mathcal{O}}_k\}_{k=1}^{K}$ offers a reliable discrete approximation to $\mathcal{M}_{\infty}$."
> > >
> > > In these sentences we employ the term "low-dimensional structure" as a concise way to refer to the fact that for many dissipative dynamical systems, the invariant measure has support on a subset of the state space with dimension $m \le d$ smaller than the (ambient) state space dimension.
> > > This is well established for deterministic dissipative systems, whose invariant (SRB/physical) measure has a compact support on flow-invariant sets, the system's attractors, with dimensionality typically smaller than the ambient dimension.\
> > > For dissipative dynamical sytems that are additionally perturbed by (weak to moderate) stochastic noise, their invariant measures often concentrate within tubular neighborhoods around these attractors. In our work, we exploit this concentration of the invariant density around the deterministic low-dimensional object, by approximating the invariant measure with a Riemmanian manifold with same dimensionality as the ambient space and a data-dependent metric that captures essentially the (inverse) density of the observations in the state space. We then propose that the most likely path between consecutive observations in the space will respect the geometry of this Riemannian manifold.
> > >
> > >
> > > To further clarify this point, we have will add a new subsection in the supplementary information [**Section Justification for Riemannian manifold approximation of the invariant density**] where we will formalise this argument until the camera ready version of the manuscript.

---

> > > > ### Author Response · Authors · 2025-12-03
> > > >
> > > > ### **5. Clarification on multi-marginal setting**
> > > > > How do you construct the bridges in multi marginal setting?
> > > >
> > > > In our work we consider direct state observations of a single stochastic trajectory. This means that the boundary condition for every bridge is a Dirac delta distribution centered at each observation, thus individual bridges are independent given a drift function estimate and are sampled in parallel.
> > > >
> > > > -----
> > > > ### **6. Clarification on drift estimation in the Riemannian setup**
> > > > > How do you learn drift control term and drift estimate given the Riemannian set-up?
> > > >
> > > > Thank you for this question. **We consider the Riemannian metric only in the construction of the geodesic curves between consecutive observations, not for constructing the augmented path.** This is the advantage of operating in the ambient space for constructing the geodesics.\
> > > > More preciselly, we endow $\mathcal{R}^d$ with the data–adapted Riemannian metric $H(x)$ to construct the more likely path of the system in $\mathcal{R}^d$ given the invariant density. Thus **the underlying set and coordinate axes between the ambient Euclediant space and the Riemannian manifold do not change, what changes is the metric**, i.e. inner products, lengths, and distances.\
> > > > For each interval $[t_k,t_{k+1}]$, we compute a geodesic $\gamma_k$ in $\mathcal{M}=(\mathcal{R}^d,H)$ between consecutive
> > > > observations, and then use this curve (sequence of states) as reference states in the path cost of the control problem. **The stochastic dynamics are still a SDE defined in $\mathcal{R}^d$ with an Eucledian metric**.
> > > > We use the data-dependent metric $H$ only to construct the path cost and do not redefine the SDE in a different space.\
> > > > Thus once we define the sequence of states of the most likely path between consecutive observations according to $\mathcal{M}=(\mathcal{R}^d,H)$, we employ these states as a path constraint in the control problem in the Eucledian space; **no projection or coordinate transformation is required**. We estimate the drift control term in the Eucledian space in a similar manner as described in Maoutsa \& Opper, 2022.\
> > > > We have now included a more detailed description of the control problem in the main text (lines ), and have provided more details also in the appendix (Sec. lines   )
> > > >
> > > > ---
> > > >
> > > >
> > > > ### **7. Rendering issue resolved**
> > > > > Minor comment, but I believe sentence in line 243 is cut off by a figure?
> > > >
> > > > We thank the reviewer for noticing this detail. Indeed the word "_them._" was covered by the figure in question. We have fixed the issue in the updated version of our manuscript.

---

### Author Response · Authors · 2025-12-03
**Overview of revisions**

**We thank again the reviewers for their time and effort to evaluate our work and to propose insightful suggestions to improve our manuscript.** In the following we briefly outline the updates we made in the paper in response to the reviewers' suggestions.
Overall the reviewers evaluated positively our contribution, identifying the **novelty and soundness** of our framework. The reviewers proposed suggestions for improving the presentation of the paper in particular related to deferring key aspects of the method in the theoretical part in the supplement.
In response to the suggestions of the four reviewers we have updated the manuscript with the following changes:


- **[1]** **Included additional baseline comparisons with Schroedinger bridge sampling frameworks (Table 1).**\
 While these frameworks consider  slightly different setting than we do (population (they) vs single state (we) observations), we constructed a setup to compare these framework to our approach by creating populations from nearby regions of the state space of the invariance density. We compared with following Schroedinger bridge sampling frameworks:
   -  **Metric flow matching** (MFM)  (_Kapuśniak, NeurIPS 2024_) (with both the LAND and the RBF approaches for estimating metrics))
   - **Generalized Schrodinger Bridge Matching**  (GSBM)   (_Liu et al, ICLR 2024_),
   -  **Simulation-free Schroedinger bridges via score and flow matching** ([SF]^2M)   (_Tong et al., AISTATS 2024_),

- **[2]** **We extended the Supplement to include an additional Section named Related work (Section E)**\
 where we discuss related work both on inference of stochastic dynamics and for sampling Schroedinger bridges that implicitly results in learning local stochastic dynamics within these frameworks, and position our work in relation to these frameworks.
- **[3]** **Revised the main text to add Section 2. Setup and Background to clarify the problem description and setting**\
 in response to reviewer uqCb.
- **[4]** **Compared with an additional baseline method that considers the LatentSDE together with a pre-trained prior drift. (Table 1)**\
 We also clarified that their intuition of using as a prior some geometric drift would be beneficial, but there is no obvious way to define or infer this geometric prior, and this is the reason we have resorted to the control approach.
- **[5]** **Performed ablations wrt the metric approximation used in our approach (Geomtric_RBF in table 1).**\
 We repeated some of the experiments while employing different metric approximation (radial basis function metric from Arvanitidis et al. 2021.
- **[6]** **Clarified in the main text, supplement and to the reviewer details on the control framework we employ.**
- **[7]** **Added a supplement section where we provide the algorithms for the constituents of our framework. (Section I)**
- **[8]** **Will add an extra theoretical supplementary section: Justification for Riemannian manifold approximation of the invariant density.**.
- **[9]** **Clarified that we do not need to compute the score function on the Riemannian setup**\
 because the Riemannian manifold is only employed to compute the most-probable path in the same coordinate axis where the SDE lives.
- **[10]** **Extended our theoretical exposition of the remainder term of the Euler-Maruyama approximation to add a geometric intuition in the Supplement (Section G.3.)**


**We hope these revisions address the reviewers’ comments and clarify our contributions, and we welcome a reassessment of the scores.**

---

### Meta-Review · Area_Chair_8Wyy · 2026-01-07

**Summary:**

This paper aims at learning overdamped Langevin systems from temporally under-resolved trajectory observations. It does so by first learning a Riemannian metric from the data invariant density, then computing geodesics between observations, and finally sampling diffusion bridges constrained to remain near those geodesics. Reviewers and I agree this is an interesting idea. Meanwhile, reviewers had significant concerns, such as about presentation and empirical results (for example, experiments are on low dimensional synthetic data). I also encourage the authors to quantify the implicit bias of their approach - given under-resolved observations, the dynamics will not be unique, but the proposed algorithm makes a specific prediction - under what circumstances would such prediction be valid, and when would it not be? Overall, I appreciate the idea and the efforts, and hope the authors could take the discussions into consideration and re-submit a revised version.

**Reviewer Concerns:**

Just a guess: they may not be fully convinced.

**Reviewer Scores:**

Just a guess: they probably won't increase by too much.

---

### Decision · Program_Chairs · 2026-01-26

Reject